# Regulators of coastal wetland methane production and responses to simulated global change

Carmella Vizza[1], William E. West[1, 2], Stuart E. Jones[1], Julia A. Hart[1, 3], and Gary A. Lamberti[1]

[1]Department of Biological Sciences, University of Notre Dame, Notre Dame, 46556, USA
[2]Kellogg Biological Station, Michigan State University, Hickory Corners, 49060, USA
[3]Center for Limnology, University of Wisconsin, Madison, 53706, USA

*Correspondence to*: Carmella Vizza (cvizza@nd.edu)

**Abstract.** Wetlands are the largest natural source of methane ($CH_4$) to the atmosphere, but their emissions vary along salinity and productivity gradients. Global change has the potential to reshape these gradients and therefore alter future contributions of wetlands to the global $CH_4$ budget. Our study examined $CH_4$ production along a natural salinity gradient in fully inundated coastal Alaska wetlands. In the laboratory, we incubated natural sediments to compare $CH_4$ production rates between non-tidal freshwater and tidal brackish wetlands, and quantified the abundances of methanogens and sulfate-reducing bacteria in these ecosystems. We also simulated seawater intrusion and enhanced organic matter availability, which we predicted would have contrasting effects on coastal wetland $CH_4$ production. Tidal brackish wetlands produced less $CH_4$ than non-tidal freshwater wetlands probably due to high sulfate availability and generally higher abundances of sulfate-reducing bacteria, whereas non-tidal freshwater wetlands had significantly greater methanogen abundances. Seawater addition experiments with freshwater sediments, however, did not reduce $CH_4$ production, perhaps because the 14-day incubation period was too short to elicit a shift in microbial communities. In contrast, increased organic matter enhanced $CH_4$ production in 75% of the incubations, but this response depended on the macrophyte species added, with half of the species treatments having no significant effect. Our study suggests that $CH_4$ production in coastal wetlands, and therefore their overall contribution to the global $CH_4$ cycle, will be sensitive to increased organic matter availability and potentially seawater intrusion. To better predict future wetland contributions to the global $CH_4$ budget, future studies and modeling efforts should investigate how multiple global change mechanisms will interact to impact $CH_4$ dynamics.

**Keywords:** Methanogenesis, sea-level rise, saltwater incursion, redox, microbial communities

# 1 Introduction

Wetlands contribute about 60% of all natural methane ($CH_4$) emissions to the atmosphere (Kirschke et al., 2013). As global temperatures continue to increase, some models predict that wetland $CH_4$ emissions will double by 2100 (Gedney et al., 2004). Because $CH_4$ is a potent greenhouse gas whose radiative forcing continues even after its oxidation to $CO_2$ (Neubauer and Megonigal 2015), higher wetland emissions could trigger a positive feedback loop that further increases temperatures and $CH_4$ release. Higher future $CO_2$ levels could result in further warming, an extended growing season (Walther et al., 2002), and $CO_2$ fertilization of photosynthetic plants (Matthews, 2007; Ringeval et al., 2011). If the resulting increases in plant productivity provide additional organic matter to fuel additional $CH_4$ production, this effect could shift the wetland greenhouse gas emission baseline. Predicting the response of these ecosystems to global change is challenging because we do not fully understand the sensitivity of the $CH_4$ cycle to enhanced productivity of wetland plants (McGuire et al., 2009; Ringeval et al., 2011).

Any global change element that directly alters the availability of electron donors or electron acceptors could change $CH_4$ production rates and baseline emissions, thereby exacerbating or mitigating the radiative forcing of climate. Methanogens generally use substrates provided by the fermentation of organic matter as electron donors, producing $CH_4$ via two pathways: (1) acetoclastic methanogenesis, where acetate is electron donor and acceptor, and (2) hydrogenotrophic methanogenesis, where $H_2$ and $CO_2$ are the substrates utilized (Conrad, 1999). Acetate is therefore an important substrate that methanogens either directly use (acetoclastic pathway) or indirectly use via the $H_2$ and $CO_2$ resulting from its fermentation and that of other organic matter (hydrogenotrophic pathway). However, assuming competition for the same electron donor, methanogens can be outcompeted for these substrates because carbon as $CH_3$ or $CO_2$ is not an energetically favorable electron acceptor in comparison to those used by other microbes (e.g., $NO_3^-$, $SO_4^{2-}$). The presence of alternative electron acceptors can indicate intense microbial competition for the fermentative substrates that methanogens utilize (Lovley and Klug, 1983; 1986; Lovley and Phillips, 1987). For example, Winfrey and Ward (1983) observed much greater rates of sulfate reduction than $CH_4$ production in intertidal sediments until sulfate became depleted. However, an abundant supply of organic matter can increase substrate availability, act as an electron donor, and allow for depletion of alternative electron acceptors (Achtnich et al., 1995). Both the availability of electron donors and acceptors will therefore play an important role in determining the effects of global change on $CH_4$ production.

Accurately forecasting the effects of sea-level rise and increased organic matter on coastal wetland greenhouse gas budgets requires a process-level understanding of responses to potential changes in electron acceptors and donors (Fig. 1). Laboratory studies and field surveys report increased $CH_4$ production and emissions with warming (Moore and Dalva, 1993; Klinger et al., 1994; Lofton et al., 2014). Additionally, elevated $CO_2$ levels can also lead to higher photosynthesis and $CH_4$ emission rates (Megonigal and Schlesinger, 1997; Vann and Megonigal, 2003). However, despite their potential importance in regulating $CH_4$ emissions from wetlands, especially those at northern latitudes, few studies have attempted to simulate the

effects of seawater intrusion or increased substrate availability on $CH_4$ production. Both of these global change mechanisms are likely to disrupt coastal wetland biogeochemical cycles, especially at northern latitudes where their effects are likely to be stronger and more abrupt.

We studied wetland ecosystems in the Copper River Delta of Alaska, an area vulnerable to global change because of its northern location and proximity to the ocean. Over the past 50 years, average annual temperatures in Alaska have increased 1.9 °C, with winter temperatures rising 3.6 °C (U.S. Global Climate Change Program, 2009), which is extending the growing season. In addition, the projected global sea-level rise of 100 cm by 2100 (Vermeer and Rahmstorf, 2009) will be exacerbated along the southcentral Alaskan coast where tectonic subsidence is prominent (Freymueller et al., 2008). For example, the Copper River Delta, which is subsiding at about 0.85 cm per year (Freymueller et al., 2008), is at risk of a relative sea-level rise of about 170 cm by 2100.

Our study objectives were to (1) compare $CH_4$ production rates and microbial community abundances in sediments from constantly inundated non-tidal freshwater and tidal brackish wetlands on the Copper River Delta, (2) simulate seawater intrusion in freshwater wetlands using a seawater addition experiment, and (3) simulate increased organic matter availability in freshwater wetlands. We hypothesized that (1) tidal brackish wetlands sediments will have lower $CH_4$ production rates than those from the non-tidal freshwater wetlands, (2) tidal brackish wetland sediments will have higher abundance of sulfate-reducing bacteria, but lower numbers of methanogens than non-tidal freshwater wetlands, (3) simulating seawater intrusion in freshwater sediments will decrease $CH_4$ production rates, with sulfate availability largely being responsible for this effect, and (4) increasing the amount of organic matter available will enhance $CH_4$ production, but substrate quality will moderate this effect. Our conceptual model for these interactions is depicted in Fig. 1.

## 2 Materials and Methods

### 2.1 Study area

The Copper River in southcentral Alaska is the eighth largest river in the United States (U.S. Geological Survey, 1990). Draining a large region of the Chugach Mountains and the Wrangell Mountains into the Gulf of Alaska, the Copper River and its sediment deposits have shaped the largest contiguous wetland on the Pacific Coast of North America. The Copper River Delta (CRD) encompasses about 283,000 hectares of wetland habitat and supports extraordinary biodiversity (Bryant, 1991) in a relatively pristine landscape. Wetlands and shallow ponds (0.2 to 2 m in depth) were created and modified by the Great Alaska earthquake in 1964 that elevated the CRD by 1−4 m depending on location (Thilenius, 1995). A natural succession of wetlands thereby emerges from the ocean to the uplands (Fig. 2). Our study focused on the brackish tidal wetlands and non-tidal freshwater wetland/pond habitats. The brackish tidal wetlands we chose to study are at the confluence of a river mouth

and the Gulf of Alaska. Therefore, these wetlands become increasingly brackish and deeper during rising high tide, but are also waterlogged during mean low tide; we consider them appropriately comparable to the fully inundated non-tidal freshwater wetlands. The freshwater wetland habitats currently receive little to no tidal influence, but their surrounding sloughs and rivers are tidally influenced, which could result in future seawater intrusion with sea-level rise. We consider the freshwater wetlands

to be "pond-like" because they have clearly delineated boundaries, whereas the brackish wetlands are more continuous in nature. We chose these two ecosystem types because they are the most prevalent yet distinctive habitats on the CRD with which to contrast $CH_4$ production.

## 2.2 Experimental design

### 2.2.1 Sample collection

Using a handheld bucket auger, sediment samples (~ 250 mL) were collected from nine non-tidal freshwater wetlands and five tidal brackish wetland sites varying in physicochemical parameters (Tables 1 and 2). Because non-tidal freshwater wetland ponds had distinct boundaries and extensive habitat heterogeneity within each wetland (i.e., open water and several different macrophyte zones), we collected at least five sediment samples representative of the different habitats at each wetland (n = 9) along with at least 1 L of hypolimnetic water during each sampling period, so that the average $CH_4$ production rates from each

system could be accurately assessed. In contrast, the tidal brackish wetland complex was continuous, lacking distinct boundaries, and generally exhibited less habitat heterogeneity than the non-tidal freshwater wetlands (i.e., we observed only sites dominated by *Carex* spp.). Because we observed temporal fluctuations in salinity with a YSI Pro Plus multiparameter water quality meter indicative of tidal influence, we collected 1 L of water and one sediment sample at five different sites along a salinity gradient. Although sediment and water from tidal brackish wetland sites were collected in one continuous

wetland complex, they were considered separately in analyses due to large differences in salinity.

### 2.2.2 Non-tidal freshwater and tidal brackish wetland comparison

To assess $CH_4$ production, laboratory incubations were conducted using sediment and water samples collected during two sampling periods (June and August 2014). To capture the greater habitat heterogeneity of the non-tidal freshwater wetlands,

we conducted five $CH_4$ production assays for each wetland (5 sediment samples x 9 wetlands x 2 time periods = 90 total incubations). We therefore characterized the non-tidal freshwater wetlands to a greater spatial extent than the brackish tidal wetlands where we conducted ten total incubations (5 sites along a salinity gradient within the continuous tidal brackish wetlands complex x 2 time periods). To account for this difference in spatial sampling, we then used the average $CH_4$ production rates from each non-tidal freshwater wetland as a replicate in comparing $CH_4$ production rates between non-tidal

freshwater (n = 9) and tidal brackish (n = 5) systems at each sampling period.

### 2.2.3 Seawater addition experiment

To assess the effects of seawater addition on $CH_4$ production, additional sediments were collected in June from a single site in five of the freshwater wetlands (n = 5) and then incubated with tidal brackish water (6.3 mM sulfate). We then compared them to the average $CH_4$ production rates of the five sediment samples incubated with freshwater from that same subset of non-tidal

freshwater wetlands (n = 5) during June 2014.

### 2.2.4 Increased organic matter simulation

To assess the effects of increased organic matter on $CH_4$ production, four sediment samples from different sites were used from five of the non-tidal freshwater wetlands (n = 20). An aliquot of each sediment sample from each wetland was incubated with fresh macrophyte tissue from one of four species (treatment) and then compared to an aliquot that served as a paired

control sediment sample (total pairs = 20; 5 wetlands x 4 treatments). This paired design controlled for "within wetland" sediment heterogeneity to better capture the response of the methanogens to adding organic matter, or $\Delta CH_4$ production (treatment–control). Our four organic matter treatments were based upon the four dominant aquatic macrophyte species on the CRD – buckbean (*Menyanthes trifoliata*), horsetail (*Equisetum variegatum*), lily (*Nuphar polysepalum*), and marestail (*Hippuris vulgaris*). Specifically, we cut aboveground tissue to a standard size per species such that 3.0 g of live biomass could

be added to each incubation resulting in approximately $0.23 \pm 0.02$ mmol C per gram of dry sediment (mean ± sd). In most incubations, this addition of organic matter increased the total amount of carbon already available in the sediment by $45 \pm 15\%$ (Table 2). All vegetation for each species was collected from the same plant individual to ensure minimal difference in quality within each treatment. Differences in substrate quality between these treatments, as described by % C, % N, and % P as well as C:N and C:P, are available from Tiegs et al. (2013): (1) Horsetail had the lowest carbon content at 38%, while the other

three species contain approximately 44–47% C, (2) Lily tissue had the highest % N (2.5) and % P (0.24) followed by marestail with 1.7% N and 0.17% P, and (3) Buckbean had 0.94% N and 0.15% P and horsetail had 1.1% N and 0.11% P.

### 2.3 Laboratory analyses

### 2.3.1 Sediment slurry incubations

For each incubation, approximately 60 mL ($82 \pm 2.5$ g) of wet sediment and 60 mL of water were incubated in a 250-mL serum

bottle in the dark at approximately 14.0 °C. To remove oxygen introduced to the inundated sediments during sample collection and slurry making, each bottle was made anoxic by purging it with $N_2$ gas for five minutes. Since incubation temperature was generally lower than average wetland temperature (June: $17.2 \pm 0.9$ °C, August: $18.4 \pm 1.3$ °C), estimated rates of $CH_4$ production potential were considered conservative. However, we do acknowledge that $CH_4$ production potentials generated by bottle incubations may not exactly reproduce $CH_4$ production rates in these ecosystems. Headspace samples (10 mL) were

removed at 2, 5, 8, 11, and 14 days, injected into a 2-mL serum vial (pre-evacuated with a vacuum pump), sealed with silicone, and stored upside down in water for less than three months until the samples could be analyzed using gas chromatography. To

maintain atmospheric pressure in the slurry incubations, 10 mL of $N_2$ gas was added after each sampling point. $CH_4$ concentrations were measured using an Agilent 6890 gas chromatograph equipped with a flame ionization detector (Agilent Technologies, Santa Clara, CA, USA) as detailed by West et al. (2015). After accounting for headspace dilution due to sampling, $CH_4$ production rates were inferred from the slope of the linear regressions of $CH_4$ concentrations over time and are

reported as nmol $CH_4$ per g of dry sediment per day (nmol $g^{-1}$ $day^{-1}$).

### 2.3.2 Physicochemical measurements

Temperature, pH, dissolved oxygen, specific conductivity, and salinity were measured at each sampling location using a YSI Pro Plus multiparameter water quality meter (YSI, Yellow Springs, OH, USA). Dissolved organic carbon was analyzed using a Shimadzu TOC-VCSH (Shimadzu Scientific Instruments, Kyoto, Japan). All samples, with the exception of five of the tidal

brackish samples, registered above the lowest standard (1 mg/L); the five exceptions registered between the blanks and the lowest standard. Acetate, nitrate, and sulfate concentrations were analyzed using a Dionex ICS-5000 (Thermo Fisher Scientific, Sunnyvale, CA, USA), but only sulfate was detectable in the water column. Detection limits for acetate, nitrate, and sulfate were approximately 10, 2, and 1 µM, respectively. Water chemistry analyses were performed using instrumentation at the University of Notre Dame Center for Environmental Science and Technology.

### 2.3.3 Sediment organic matter and porewater chemistry

To examine starting conditions for each $CH_4$ production assay, a subsample of sediment was frozen at the start of the incubation for later analysis. A portion of each subsample was dried for at least 48 hours at 60 °C, and the dry weight was recorded. Subsequently, the organic matter in the sediment was combusted at 500 °C for four hours, and the sediment was re-wetted and then dried at 60 °C for at least 48 hours before re-weighing (Steinman et al., 2011). Sediment organic matter was estimated as

the percent of sediment material lost during combustion (SOM %) and converted to the total sediment organic carbon (Thomas et al., 2005) available per g of dry sediment (Table 2). To extract porewater from the sediment, another portion (~ 50 mL) was centrifuged for 45 minutes at 4 °C at ~ 4000 RCF. The total volume of supernatant per volume of sediment was recorded, and a subsample of the porewater was also analyzed on the Dionex ICS-5000 for acetate, nitrate, and sulfate. To account for the widely differing porewater volumes we extracted from sediment (0.17 ± 0.09 ml porewater per mL of sediment), porewater

concentrations were converted to the total amount of each anion (nmol) per g of dry sediment (i.e., µM x porewater volume in incubation x porewater volume per mL of sediment x sediment volume in bottle / mass of dry sediment x 1000; Table 2).

### 2.3.4 Microbial analyses

According to the manufacturer's protocol with a PowerSoil DNA isolation kit (Mo Bio, Carlsbad, CA, USA), DNA was extracted from frozen sediments used in other analyses, including multiple June tidal brackish sediments (n = 10), the

freshwater sediments used in the seawater intrusion simulation (n = 5), and a composite of the five sediment samples (1 g sediment per sample was added to make a 5-g composite) from the nine freshwater wetlands for the June time period (n = 9).

We chose to make composites for microbial analyses of the non-tidal freshwater wetlands for the purpose of controlling analytical costs while controlling for the significant spatial heterogeneity in these ecosystems. Extracted DNA served as a template for quantitative PCR (qPCR) targeting of two genes – the alpha subunit of methyl coenzyme reductase (*mcrA*) and the alpha subunit of dissimilatory sulfite reductase (*dsrA*). The *mcrA* gene catalyzes the reduction of a methyl group to $CH_4$

(Thauer, 1998), and is possessed by all known methanogens thereby making it ideal for quantifying methanogen abundance (Luton et al., 2002; Earl et al., 2003; Castro et al., 2004). The *dsrA* gene catalyzes the final step in sulfate respiration, and its ubiquity in sulfate-reducing bacteria makes it powerful at assessing their abundance (Wagner et al., 1998; Klein et al., 2001; Zverlov et al., 2005). Although the number of genes does not necessarily equate with number of cells or gene activity, qPCR of functional genes for particular guilds is a commonly used approach to estimate the abundance of a functional group and

these gene abundances have been correlated with functional processes such as $CH_4$ production (e.g., Morris et al., 2015).

The *mcrA* and *dsrA* genes were amplified using a 20-µL qPCR reaction in a Mastercycler ep realplex[2] gradient S (Eppendorf, Hamburg, Germany), using SYBR Green as the reporter dye. Each reaction contained 1 µL of brackish or freshwater wetland DNA template and was conducted using the PerfeCTa SYBR Green FastMix (Quanta BioSciences). For the *mcrA* qPCR, primer details and thermocycling conditions in West et al. (2012) were replicated except that we employed a

15 fluorescent detection step at 78 ℃ for 20 seconds. For the *dsrA* qPCR primer, details and thermocycling conditions in Kondo et al. (2008) were replicated. Melting curves for both *mcrA* and *dsrA* were run to ensure absence of non-specific amplification. Amplification, fluorescence data collection, and initial data analysis were all performed by the Eppendorf realplex[2] software.

Standard qPCR curves for *mcrA* and *dsrA* were generated by pooling gel-extracted amplicons containing our qPCR primer sites from a subset of our non-tidal freshwater and tidal brackish wetland samples. We amplified *mcrA* using primers

detailed in Luton et al. (2002) and thermocycling conditions in West et al. (2012), and *dsrA* by replicating primer details and thermocycling conditions in Kondo et al. (2008). After amplification, we used gel electrophoresis and an Invitrogen PureLink Quick Gel Extraction Kit (Invitrogen, Carlsbad, CA, USA) to isolate the *mcrA* and *dsrA* amplicons. Following clean-up, we quantified the purified amplicons using Invitrogen's Qubit technology. We then used serial ten-fold dilutions of these genes to generate standard curves for qPCR. Our detection limit for each gene was approximately 1000 copies per g of wet sediment.

Samples below detection were assigned a value of 999 copies per g for further analysis. We ran triplicate analyses of all samples for both the *mcrA* and *dsrA* qPCR, the averages of which were used in summary statistics and analyses.

**2.4 Statistical analyses**

For the non-tidal freshwater (n = 18, 9 sites x 2 time periods) and tidal brackish wetland comparison (n = 10, 5 sites x 2 time periods), we analyzed how four factors influenced log-transformed $CH_4$ production rates using generalized linear models

(GLM) and Akaike Information Criterion (AIC) based model selection. The four factors were: (1) ecosystem type (non-tidal freshwater or tidal brackish), (2) time period (June or August), (3) porewater acetate availability (nmol g$^{-1}$ dry sediment), and

(4) total sulfate present (nmol g$^{-1}$ dry sediment). As nitrate availability was extremely low in these ecosystems in comparison to total sulfate availability (i.e., ~5%), we did not include nitrate as a factor in the GLMs. AIC-based model selection identifies the most likely model given the data while penalizing for model complexity (i.e., the number of parameters). In our analysis, we corrected for small sample sizes (AIC$_c$; Burnham and Anderson 2002). The model with the lowest AICc value is considered

the most likely, and all remaining models are compared relative to the most likely model using delta AIC$_c$ ($\Delta_i$). Models with a $\Delta_i$ less than or equal to 2 are considered to have substantial support, while models having a $\Delta_i$ greater than 7 have little support (Burnham and Anderson 2002). The relative strength of our candidate models was then evaluated with Akaike weights ($\omega_i$), which indicate the probability of a model being the most likely model, given the data and the set of candidate models (Burnham and Anderson 2002). We considered 16 candidate models (all possible additive combinations of the four factors including the

null model) using the methods described above. A subset of those models, excluding the null model (i.e., intercept only) and those with relatively low support ($\Delta_i > 4$), were then used to determine model-averaged parameter estimates and to estimate the relative importance of variables (Burnham and Anderson, 2002). To estimate the relative importance of predictor variable $x$, we used the sum of Akaike weights for models including variable $x$ (the closer the sum is to 1, the more important the variable $x$); we only considered models where $\Delta_i < 4$ for this analysis (Burnham and Anderson, 2002).

To compare the abundance of methanogens and sulfate-reducing bacteria, we first used a chi-squared test for each gene to determine whether the presence/absence of *mcrA* or *dsrA* was independent of ecosystem type. We then used a non-parametric Kruskal-Wallis tests to determine whether the number of copies of *mcrA* or *dsrA* varied by ecosystem type. For all statistical analyses excluding AIC model selection, α was set 0.05.

       For the seawater addition experiment, we conducted a paired *t*-test to determine whether CH$_4$ production rates in non-

tidal freshwater wetland sediments were affected by being incubated anaerobically with brackish tidal water instead of freshwater from their respective wetlands. Pearson correlations were computed (Zar, 2010) to determine whether porewater acetate or total sulfate levels were related to CH$_4$ production rates during this experiment.

       To determine whether adding organic matter affected CH$_4$ production rates, we first used an analysis of variance (ANOVA) with treatment (i.e., macrophyte species) as the factor of interest and non-tidal freshwater wetland as a blocking

variable. Then we analyzed how three factors influenced the response of each sediment, or $\Delta$CH$_4$ production (treatment–control), using additive GLMs. The three factors were: (1) macrophyte species added, (2) total acetate available in the porewater (nmol g$^{-1}$ dry sediment), and (3) total amount of sulfate present (nmol g$^{-1}$ dry sediment). A total of eight candidate models (all possible additive combinations of the three factors including the null model) were compared as described above. To determine whether macrophyte species stoichiometry influenced the response of methanogens to increased organic matter,

linear regressions were computed for % C, % N, % P, C:N, and C:P (from Tiegs et al., 2013) against $\Delta$CH$_4$ production. All

statistical analyses were conducted in the R software environment using the base and MuMIn packages (R Development Core Team, 2016).

## 3 Results

### 3.1 Non-tidal freshwater and tidal brackish wetland comparison

**3.1.1 Water column and porewater chemistry**

Water column and sediment porewater chemistry of the incubations varied by ecosystem type (Tables 1 & 2), and variation by ecosystem type tended to be greater than temporal variation. Total sulfate levels in non-tidal freshwater incubations (June: 84 $\pm$ 65; August: 48 $\pm$ 43 nmol gram$^{-1}$ dry sediment; mean $\pm$ sd) were about two orders of magnitude lower than in tidal brackish incubations (June: 4300 $\pm$ 4300; August: 3500 $\pm$ 3700 nmol gram$^{-1}$ dry sediment) and did not vary between time periods. In

comparison to total sulfate levels, porewater nitrate availability was very low, with non-tidal freshwater wetlands (June: 1.5 $\pm$ 0.9; August: 1.8 $\pm$ 1.8 nmol gram$^{-1}$ dry sediment) having relatively higher nitrate than the tidal brackish wetlands (June: 0.24 $\pm$ 0.51; August: 0.0092 $\pm$ 0.0025 nmol gram$^{-1}$ dry sediment; Table 2). The total amount of acetate available in the non-tidal freshwater wetland incubations was similar in June (28 $\pm$ 22 nmol gram$^{-1}$ dry sediment) and August (30 $\pm$ 17 nmol gram$^{-1}$ dry sediment), while levels in the tidal brackish wetland incubations were generally higher and more variable especially in August

(210 $\pm$ 260 nmol gram$^{-1}$ dry sediment) than in June (130 $\pm$ 80 nmol gram$^{-1}$ dry sediment).

**3.1.2 CH$_4$ production**

CH$_4$ production rates were higher in non-tidal freshwater wetlands than in tidal brackish wetlands and approximately an order of magnitude higher in both ecosystems in August compared to June (Fig. 3). Porewater acetate was positively associated with higher CH$_4$ production rates, while higher total sulfate availability was associated with lower CH$_4$ production rates (Table 3).

The most likely model contained all four factors – ecosystem type, time period, acetate, and total sulfate (Table 3). Based upon model averaging of the top three models (Table 3), all four factors appeared to influence CH$_4$ production with the relative importance of these variables being 1.00 for ecosystem, 1.00 for porewater acetate, 0.87 for total sulfate availability, and 0.74 for time period.

**3.1.3 Functional group abundances**

Tidal brackish sediments tended to have higher abundances of sulfate-reducing bacteria when present, while non-tidal freshwater sediments were characterized by higher numbers of methanogens. In the tidal brackish wetlands, three out of ten samples were below the detection limit for the *dsrA* gene, our proxy for sulfate-reducing bacteria abundance, but we detected this gene in all nine non-tidal freshwater wetland composite samples. The presence or absence of the *dsrA* gene was independent of ecosystem type ($\chi^2 = 3.21$, df = 1, $P = 0.07$). Tidal brackish sediments (n = 10) and non-tidal freshwater wetland

sediments (n = 9) had $3.52 \pm 5.39$ x $10^5$ and $5.20 \pm 5.08$ x $10^4$ copies of *dsrA* per gram of wet sediment, respectively. Due to high variability, the number of copies of *dsrA* did not differ significantly by ecosystem (Kruskal-Wallis: $H = 1.31$, df = 1, $P = 0.25$). In contrast, we detected the *mcrA* gene, our proxy for methanogen abundance, in only two out of ten tidal brackish samples, but in all nine non-tidal freshwater wetland samples. The presence or absence of the *mcrA* gene was dependent on ecosystem type ($\chi^2 = 12.44$, df = 1, $P = 0.0004$). Tidal brackish samples had $2.14 \pm 5.78$ x $10^4$ copies of the *mcrA* per gram of wet sediment, while non-tidal freshwater wetlands had $1.84 \pm 1.25$ x $10^5$ copies of *mcrA* per gram of wet sediment. Methanogen abundance therefore differed significantly between ecosystem types (Kruskal-Wallis: $H = 11.24$, df = 1, $P = 0.0008$)

**3.2 Seawater addition experiment**

Incubating non-tidal freshwater wetland soils with brackish water did not affect $CH_4$ production rates (Fig. 4). Even though total sulfate levels increased from $63 \pm 37$ to $5400 \pm 400$ nmol gram$^{-1}$ dry sediment with the addition of tidal brackish water, $CH_4$ production rates did not differ between treatment and control incubations (paired *t*-test: $t = 0.44$, df = 4, $P = 0.68$). However, $CH_4$ production rates were significantly correlated with porewater acetate levels ($r = 0.88$, $t = 5.18$, df = 8, $P = 0.0008$), but not with total sulfate levels ($r = 0.09$, $t = 0.24$, df = 8, $P = 0.81$). The non-tidal freshwater wetland sediments used in this seawater addition experiment (n = 5) had about an order of magnitude higher number of copies of *mcrA* ($3.12 \pm 4.40$ x $10^5$) than *dsrA* ($5.32 \pm 6.33$ x $10^4$) per gram of wet sediment.

**3.3 Increased organic matter simulation**

The organic matter treatments significantly influenced $CH_4$ production rates ($F_{4, 16} = 4.48$, $P = 0.01$), but this effect varied with macrophyte species (Fig. 5). Adding buckbean and marestail had little effect on $CH_4$ production, while the lily and horsetail treatments generally increased methanogen activity (Fig. 5). The most likely model for predicting $\Delta CH_4$ production (treatment – control) included acetate availability, which had a negative effect on the response (Table 4). The next best models included porewater acetate and species (Model 2) or porewater acetate and total sulfate availability (Model 3), which had a positive effect on the response (Table 4). Models 1–4 (Table 4) were averaged to determine parameter estimates with the relative importance of the variables being 0.88 for porewater acetate, 0.33 for macrophyte species, and 0.15 for total sulfate availability. Using the model-averaged parameters, our predictions of the response of $CH_4$ production rates to increased substrate availability closely followed the observed results (Fig. 6). Finally, macrophyte species stoichiometry (i.e., % C, % N, % P, C:N, and C:P) had no effect on $\Delta CH_4$ production ($r^2 < 0.08$, $P > 0.24$ for all regressions).

# 4 Discussion

To begin to understand likely responses of wetlands to global change processes, we conducted a space-for-time substitution of how seawater intrusion might affect $CH_4$ production in freshwater wetlands by comparing them to brackish systems. We found that $CH_4$ production was lower in tidal brackish than in non-tidal freshwater wetlands, likely due to differences in availability of alternative electron acceptors (i.e., higher sulfate levels in the tidal brackish) and in microbial communities (i.e., lower methanogen abundances in the tidal brackish). Experimental addition of seawater in non-tidal freshwater sediments (~14 days), however, did not influence $CH_4$ production rates. In contrast, higher organic matter availability enhanced $CH_4$ production rates in 75% of incubations, but this response depended on the amount of substrate already available and the macrophyte species added, with half of the species treatments having no significant effect. Because acetate and sulfate availability had contrasting effects depending on the experiment (i.e., freshwater/brackish comparison vs. increased organic matter), these results indicate that we do not have a sufficient mechanistic understanding of how changes in electron donors and electron acceptors will interact to ultimately influence $CH_4$ production. Future studies should consider the possible interaction of global change mechanisms, such as sea-level rise and $CO_2$ fertilization/longer growing seasons, which will likely alter the availability of electron acceptors and electron donors, thereby influencing $CH_4$ production (Fig. 1).

## 4.1 Non-tidal freshwater and tidal brackish wetland comparison

$CH_4$ production rates in tidal brackish wetlands were substantially lower than those of non-tidal freshwater wetlands, as predicted. Many studies have attributed the decrease in wetland $CH_4$ emissions along increasing salinity and sulfate concentrations to sulfate-reducing bacteria outcompeting methanogens for substrates (DeLaune et al., 1983; Bartlett et al., 1987; Magenheimer et al., 1996; Poffenbarger et al., 2011), but none of these studies directly assessed whether lower $CH_4$ emissions resulted from reduced $CH_4$ production or higher $CH_4$ oxidation. Two recent studies documented lower $CH_4$ production with elevated salinity (Chambers et al., 2013; Neubauer et al., 2013), and attempted to link C mineralization rates to extracellular enzymes, but microbial communities were not quantified. In comparison, our study quantified $CH_4$ production along a similar spatial gradient and directly linked lower $CH_4$ production to higher sulfate availability and indirectly to relative abundance of functional microbial guilds. The presence of alternative electron acceptors such as sulfate likely indicates that methanogens have to compete for organic substrates with sulfate-reducing bacteria (Oremland and Polcin, 1982; Lovley and Klug, 1986; Achtnich et al., 1995). Our study also demonstrates that tidal brackish sediments tended to have generally higher sulfate-reducing bacteria (*dsrA*) abundances when present, but significantly lower levels of methanogens (*mcrA*) than non-tidal freshwater sediments. Although we did not include microbial data in the model selection due to sample size limitations, we hypothesize that microbial community differences could help to explain why ecosystem type (freshwater vs. brackish) was an important factor during model selection. Collectively, these results along with higher sulfate availability in tidal brackish

wetlands (and sulfate's importance in our model selection analysis) suggest that shifts in the relative abundance of functional microbial guilds between tidal brackish and non-tidal freshwater wetlands contribute to differences in $CH_4$ production between these ecosystems.

The difference between brackish and freshwater wetland $CH_4$ production could also be shaped by other ecosystem factors such as salinity and salinity-induced cation exchange. Because salinity and sulfate availability are often correlated, it can be difficult to disentangle these two factors; Chambers et al. (2011) isolated their effects in a laboratory manipulation and found that seawater (sulfate) had a more dramatic and longer lasting effect on $CH_4$ production than saltwater (NaCl). Nevertheless, salinity often places additional stress on organisms, such that saltwater intrusion alters microbial and plant communities (Herbert et al., 2015). Additionally, saltwater intrusion can influence cation exchange in the sediments, such that calcium is mobilized, which can co-precipitate with phosphate, and ammonium is released, all of which can shift a wetland towards P rather than N limitation (Herbert et al., 2015; van Dijk et al., 2015). Although we did not directly measure these effects of salinity and therefore cannot rule them out, we hypothesize that sulfate availability and differences in functional microbial guilds are primarily responsible for differences in $CH_4$ production rather than salinity and salinity-induced cation exchange. Our hypothesis relies on three observations: (1) N and P availability were extremely low in both freshwater and brackish ecosystems (DIN: $< 25 \mu g$ N $L^{-1}$, SRP: $< 15 \mu g$ P $L^{-1}$) and therefore different sediment cation exchange capacities were unlikely to change the N and P limitation of these wetlands, (2) salinity tended to be consistently low in freshwater wetlands, but $CH_4$ production was still negatively correlated with sulfate availability, and (3) sulfate availability was an important factor in ecosystem comparison model selection, and was the only factor where a direct mechanistic link can be made to the differences in $CH_4$ production between freshwater and brackish ecosystems (i.e., acetate availability was higher in brackish wetlands and therefore one might expect higher $CH_4$ production).

In addition to the influences of microbial communities and alternative electron acceptors on $CH_4$ production, acetate availability appeared to be an important factor. Substrate availability regulates $CH_4$ production (Whalen, 2005), and acetate is one of the major precursors for methanogenesis (Conrad, 1999) as it can be a direct (acetoclastic) or an indirect (hydrogentrophic) substrate for methanogens after further fermentation. Although the importance of acetate as a factor in our experiments suggests that acetoclastic methanogenesis may be prevalent in the CRD, we cannot rule out the potential of hydrogenotrophic methanogenesis, which is thought to be the primary pathway in other Alaskan wetlands (Hines et al., 2001). According to Hines et al. (2008), acetate tended to accumulate in Alaskan peat rather than be converted to $CH_4$ possibly due to homoacetogenic bacteria (i.e., those that make acetate) being able to outcompete methanogens for $CO_2$ and $H_2$ in colder temperatures and the general lack of acetoclastic methanogens. In contrast, $CH_4$ production in CRD wetlands was tightly related to acetate availability in the ecosystem comparison as well as in both simulations. Despite the differences between these Alaskan wetlands (CRD sediment is more similar to clay than to peat; see SOM % in Table 2), CRD freshwater wetlands

exhibited similar $CH_4$ production rates to those conducted during August 2001 by Hines et al. (2008), which ranged from about 10 to 500 nmol $g^{-1}$ dry peat $day^{-1}$. Because $CH_4$ production rates in Alaskan peat tended to increase with higher proportions of vascular plant cover (Hines et al., 2008) and the fermentation of this plant matter facilitates the production of acetate, it is possible that the role of the acetoclastic pathway may grow more important in northern wetlands in the future as vascular plant growth increases (Klady et al., 2011).

$CH_4$ production rates often vary seasonally as a function of temperature, but we observed August rates that were an order of magnitude higher than those conducted in June despite these incubations being conducted at the same temperature. Other factors affecting $CH_4$ production that could vary seasonally include (1) availability of organic matter such as acetate for $CH_4$ production (Whiting and Chanton, 1993; Walter et al., 2001), (2) availability of alternative electron acceptors including sulfate (Sinke et al., 1992), (3) microbial population densities (Yannarell and Triplett, 2005), or (4) the pathway by which $CH_4$ is produced (Avery et al., 1999). In our study, we did not observe large seasonal differences in porewater acetate or sulfate availability in CRD wetlands, but we did not assess seasonal variation in the abundances of methanogens and sulfate-reducing bacteria, their per-cell activity rates, or availability of $H_2$ or methanogenic substrates other than acetate. Therefore, it is possible that the observed seasonal differences in $CH_4$ production rates were the result of microbial community shifts, decreased per-cell activity of methanogens in June, greater $CH_4$ produced from the hydrogenotrophic pathway during August as acetate levels did not change, or some combination of these potential explanations. Additionally, we acknowledge that the porewater acetate level we measured is an indicator of the balance between acetogenesis and acetate consumption, so it is possible that acetogenesis rates increased during August and the acetoclastic pathway of methanogenesis correspondingly increased such that acetate availability appeared to be similar during these two months. Although we did not collect the data that satisfactorily explain these intriguing seasonal differences, we hypothesize that $CH_4$ production rates vary in accordance with macrophyte phenology in these ecosystems, which clearly affects both the availability of electron donors and microbial processing rates (e.g., Eviner and Chapin, 2003). We think the following seasonal trajectory is possible: (1) In early growing season, $CH_4$ production is low, but steeply increases at peak growing season as more labile plant exudates are produced, and (2) The end of the growing season results in plant senescence, increased organic matter availability as plants decompose, and reduced oxygen levels, which then results in higher $CH_4$ production until colder temperatures start to decelerate microbial processing. All of these conditions could lead to seasonal succession in microbial communities and their activity rates. Future studies should seek to explain the mechanism behind seasonal differences in $CH_4$ production that are independent of temperature.

**4.2 Seawater addition experiment**

Despite our finding that $CH_4$ production rates were significantly lower in tidal brackish wetlands sites, adding seawater to non-tidal freshwater sediments surprisingly did not affect $CH_4$ production rates. We acknowledge that our experiment simulated

short-term consequences of seawater intrusion such as increased sulfate availability and the addition of other marine nutrients and microbial communities, but we were not simulating longer term changes such as differences in plant communities and production that may result from increased salinity (Neubauer 2013; Hopfensperger et al., 2014; Herbert et al., 2015). Nevertheless, many other short-term studies conducting similar seawater addition experiments have observed a decrease in

$CH_4$ production rates with elevated salinity (DeLaune et al., 1983; Chambers et al., 2011; Marton et al., 2012; Chambers et al., 2013; Neubauer et al., 2013; van Dijk et al., 2015). In many of these studies, however, sulfate availability was much higher. For example, DeLaune et al. (1983) found that $CH_4$ production was inhibited with the addition of ~10 mM sulfate, which is higher than the sulfate concentration (~6 mM) used in this study. Chambers et al. (2011) observed a reduction in the treatments where sulfate concentrations were about 130 and 320 µmol per $g^{-1}$ of dry sediment, which is over one order of magnitude

larger than our seawater addition experiment (5 µmol $g^{-1}$ of dry sediment). Additionally, the majority of all these experiments were conducted at 25–30 °C, or almost double the temperature used in this study (14 °C), which could increase the rates at which microbial communities and their activities respond. It is therefore likely that the external environmental conditions imposed, such as the temperature, salinity, or sulfate availability used in a seawater addition experiment, can influence the results.

In addition to environmental conditions, initial factors such as soil characteristics or site properties may mediate how methanogens respond to seawater addition experiments (Neubauer et al., 2013). For example, van Dijk et al. (2015) found that elevated salinity decreases $CH_4$ production in peat but not in clay, and the sediment of the CRD wetlands is claylike in nature. Additionally, in some of these experiments, the sediments prior to incubation had been exposed to higher levels of sulfate (e.g., brackish sediments used by DeLaune et al. 1983), and microbial communities therefore could have been more likely to respond

with higher rates of sulfate reduction, thereby increase competition for organic substrates. In contrast, the freshwater sediments used in this simulation had lower sulfate availability, and the sulfate-reducing bacteria abundances were an order of magnitude lower than methanogens. In some cases, however, sulfate reduction can increase without a corresponding decrease in $CH_4$ production (Hopfensperger et al., 2014), especially if seawater intrusion increases both sulfate and organic matter availability (Weston et al., 2011).

Seawater intrusion could therefore affect both availability of alternative electron acceptors and organic matter, but their contrasting effects on $CH_4$ production are mediated by microbial communities and processes. Although the presence of sulfate-reducing bacteria was detectable in the sediments used in this simulation, we do not know whether these taxa were active or dormant. In fact, dormant taxa can account for almost 40% of taxon richness in nutrient-poor systems (Jones and Lennon, 2010), such as the CRD freshwater wetlands. Additionally, we conducted 14-day incubations, which may have been

too short to allow for shifts in the relative abundance of sediment microbial populations (Hoehler and Jørgensen, 2013). For example, Edmonds et al. (2009) found no changes in microbial community composition of bacteria or archaea after sediment

cores had been exposed to seawater for 35 days. We therefore hypothesize that the reason that $CH_4$ production in freshwater sediments did not respond to the seawater addition experiment is a combination of environmental conditions, initial sediment factors, and a lag in response time from the microbial communities.

### 4.3 Increased organic matter simulation

Higher availability of organic matter generally increased $CH_4$ production rates, but this effect varied with the species of macrophyte added to the incubations. Differences in litter quality is known to influence methanogen communities and $CH_4$ production (Yavitt et al., 1990; 2000; Valentine et al., 1994). For example, West et al. (2012) found that adding algal carbon significantly enhanced $CH_4$ production relative to terrestrial carbon. Although aquatic macrophyte carbon may be of lower quality than that of algae, aquatic macrophytes are generally more labile than terrestrial plants (Schlickeisen et al., 2003). For

example, Tiegs et al. (2013) found that terrestrial plants decomposed more slowly than aquatic macrophytes in CRD wetlands. Additionally, Tiegs et al. (2013) conducted a decomposition assay of all the macrophyte species used in this study, as a way of assessing litter quality, and found that buckbean and lily leaves decomposed at about the same rate, but both were faster than marestail and horsetail. The rate of decomposition of different plant species was correlated with phosphorus content, and therefore indicative of litter quality differences (Tiegs et al., 2013). However, our $CH_4$ production response did not follow the

decomposition pattern documented by Tiegs et al. (2013); we observed higher $CH_4$ production for the lily and horsetail treatment relative to the control, but not for buckbean and marestail. We also did not find that the $CH_4$ production response to organic matter treatment varied by % C, % N, % P, C:N, C:P, or any other measure of litter quality assessed by Tiegs et al. (2013).

Other measures of litter quality beyond elemental composition could explain differences in the methanogen response.

West et al. (2015), for example, found that higher lipid content of phytoplankton enhanced $CH_4$ production rates. Alternatively, certain properties may influence the fermentative microbial communities associated with vegetation during decomposition (Boon et al., 1996), which are responsible for providing methanogenic substrates. For example, in a survey of 209 plants, Bishop and MacDonald (1951) reported that buckbean was one of the 10 most active species for antibacterial substances, while horsetail did not possess such properties. Specifically, buckbean extracts include aucubin, a defensive compound that can

inhibit many strains of anaerobic bacteria (Weckesser et al., 2007). Marestail also contains aucubin as well as a verbascoside, another antimicrobial compound (Damtoft et al., 1994). In contrast, the only part of lily linked to potential antimicrobial properties is the rhizomes, which have been used in folk medicine (Padgett, 2007) and are more likely to require defensive compounds because of competition with the sediment microbial community than the floating leaves we used for this experiment. Therefore, we hypothesize that $CH_4$ production varied as a function of a different measure of litter quality than

previously put forward (e.g., C:N:P, percent lignin, or lipid content), whereby the negative effects of the antimicrobial

properties of buckbean and marestail on the fermentative bacteria superseded the positive effect of increasing the amount of organic matter. We suggest that this hypothesis is worthy of further examination.

Many other studies have documented that $CH_4$ production is enhanced by the addition of direct substrates such as acetate and $H_2$ (Williams and Crawford, 1984; Bachoon and Jones, 1992; Amaral and Knowles, 1994; Coles and Yavitt, 2002;

Yavitt and Seidman-Zager, 2006), or the addition of indirect substrates such as dextrose and glucose (DeLaune et al., 1983; Williams and Crawford, 1984; Coles and Yavitt, 2002), which would need to be broken down by fermentative bacteria before methanogens could utilize them. Fewer studies have examined the effects of more biologically realistic, indirect substrates such as plant or algal matter on $CH_4$ production incubations (but see Valentine et al., 1994; West et al., 2012; 2015). However, two studies involving larger scale plots with elevated $CO_2$ levels exhibited greater photosynthetic rates and greater $CH_4$

emissions (Megonigal and Schlesinger, 1997; Vann and Megonigal, 2003). Although Vann and Megonigal (2003) observed enhanced plant biomass that was strongly correlated with $CH_4$ emissions, Megonigal and Schlesinger (1997) did not see increased biomass and therefore hypothesized that lower transpiration rates, not increased substrate availability, led to higher $CH_4$ emissions by increasing flooding duration and stimulating anaerobic processes. In our study, increased substrate availability is likely the mechanism behind increased $CH_4$ production because our smaller scale simulation did not alter

flooding duration, anaerobic conditions, or the physical structures by which plants can act as conduits for gas exchange (i.e., aerenchyma). Interestingly, the amount of acetate already available in the sediment appeared to moderate the methanogen response to enhanced substrate availability. The negative relationship between $\Delta CH_4$ production and porewater acetate concentration suggests that methanogenic substrate concentrations can become saturated, which is expected from traditional Michaelis-Menten enzyme kinetics.

Another indication of substrate limitation is the positive relationship between the methanogenic response to added organic matter and the total amount of sulfate available in the incubation. This alternative electron acceptor provides more energy than either methanogenic pathway (acetoclastic or hydrogenotrophic) when coupled to the oxidation of organic matter (Stumm and Morgan, 1996; Schlesinger and Bernhardt, 2013). For example, Westermann and Ahring (1987) found that inhibiting sulfate reduction stimulated $CH_4$ production in an alder swamp, suggesting that methanogens and sulfate-reducing

bacteria compete for common substrates. Sulfate availability, therefore, may signal strength of competition for electron donors (organic matter) that methanogens must overcome to produce $CH_4$. The higher the competition, the more likely that methanogens respond positively to the addition of organic matter. The response of methanogens to increased substrate availability, therefore, is likely regulated by the quality of the substrate (e.g., C:P, lipid content, or antimicrobial compounds), strength of competition for substrate (e.g., availability of alternative electron acceptors, microbial community assemblages, or

30  per-cell activity rates), and whether substrate availability is limiting or saturated in the environment. Although total sulfate

availability played a less significant role than acetate and macrophyte species, the model using averaged estimates from all three parameters allowed us to accurately predict the response in $CH_4$ production for this experiment.

## 5 Conclusions

Our study demonstrates that potential interactions between elements of global change, specifically seawater intrusion and increased organic matter from longer growing seasons and $CO_2$ fertilization, could have competing effects on $CH_4$ production from coastal wetlands (Fig. 1). Determining the timescale required for processes at the microbial scale to shift towards sulfate reduction is challenging, and the magnitude of seawater intrusion needed to induce this shift is currently unclear. Microbial community shifts can occur over longer timescales than several months, and $CH_4$ production can be more affected by long-term salinization (~ 3.5 years) than 2-day salinity pulses (Neubauer et al., 2013). As others have noted, the global carbon cycle is inextricably linked to other elemental cycles (i.e., sulfur) by processes taking place at the microbial scale (Schimel, 2004; Burgin et al., 2011). In addition, the potential effects of seawater intrusion are not limited to $CH_4$ production alone. Salinization also reduces aerobic and anaerobic methane oxidation, with aerobic organisms being particularly sensitive to salinity (Dalal et al., 2008; Herbert et al., 2015). Furthermore, the effects of sulfate availability on the $CH_4$ cycle extend beyond sea-level rise to other aspects of global change such as road salts and agricultural land use (Helton et al., 2014; Herbert et al., 2015).

In contrast to sea-level rise and increased sulfate availability, longer growing seasons and $CO_2$ fertilization will likely enhance carbon substrate supply and in turn $CH_4$ production. Our study demonstrates that the effect of increased organic matter depends on plant species, the availability of other methanogenic substrates, and the presence of alternative electron acceptors. It is possible that longer growing seasons and $CO_2$ fertilization could reduce competition between methanogens and other microbial communities by providing more substrates, as we saw in freshwater wetlands with higher sulfate concentrations, thereby superseding the effect of seawater intrusion. Additionally, the $CO_2$ fertilization effect could increase organic matter accretion of marsh plants, which could physically counteract sea-level rise by raising marsh elevation (Langley et al., 2009). Future studies should consider how the interaction of sea-level rise, increased organic matter, and warming will affect both the microbial and ecosystem processes of the global methane cycle. This intersection of global change processes will be particularly important for projecting the future $CH_4$ budgets of coastal wetland ecosystems.

## 6 Data availability

The data will be freely accessible through the international repository, Knowledge Network for Biocomplexity (KNB) at: https://knb.ecoinformatics.org/#view/doi:10.5063/F1028PF8.

*Author contributions.* CV designed the study as sparked from discussions with SEJ. CV and JAH conducted the fieldwork and laboratory analyses. WEW played a key role in methodology and analyzing methane samples with the GC. SEJ and GAL played advisory roles in shaping this research. CV prepared the manuscript with contributions from all co-authors.

*Competing interests.* The authors declare that they have no conflict of interest.

*Acknowledgments.* We thank the Cordova Ranger District of the USDA Forest Service for providing field and logistical support, particularly Deyna Kuntzsch, Andrew Morin, Sean Meade, Luca Adelfio, and Ken Hodges, without whom this work on the Copper River Delta would not have been possible. We also thank Gordie Reeves of the Pacific Northwest Research Station for his leadership and direction in the extensive research being conducted on the Copper River Delta. Mike Brueseke, Melanie Runkle, and Josephine Chau assisted with DOC and SOM analyses. The Center for Environmental Science and Technology at UND provided instrumentation and analytical assistance for the chemical analyses. Funding was provided by the USDA Forest Service, the Pacific Northwest Research Station, the National Fish and Wildlife Foundation, the University of Notre Dame, and the National Science Foundation Graduate Research Fellowship Program. We also thank members of the Jones laboratory and the Lamberti laboratory at UND for their feedback on the manuscript. We acknowledge Drs. Mary Scranton, Scott Neubauer, and Marcelo Ardón, as well as an anonymous reviewer who put tremendous effort into reviewing this manuscript and whose comments and suggestions greatly improved its quality.

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

**Table 1.** Water column physical and chemical characteristics (mean ± sd) of the wetlands sampled in the Copper River Delta including elevation, depth, temperature, pH, specific conductivity (SpC), salinity, dissolved organic carbon (DOC), and sulfate concentrations. For the freshwater wetlands, physicochemical parameters are from spot measurements of the hypolimnion conducted throughout summer 2014 (n = 4 per freshwater wetland), whereas DOC and sulfate were measured in the surface layer in June and August 2014 (5 sites x 2 time periods, n =10 per wetland). For the brackish wetlands, physicochemical parameters are from one spot measurement of the surface layer. Tidal brackish wetlands A–E were sampled in June and F–J were sampled in August.

| Wetland | Elevation (m) | Depth (m) | Temp (°C) | pH | SpC ($\mu s\ cm^{-1}$) | Salinity (ppt) | DOC ($mg\ L^{-1}$) | Sulfate ($\mu M$) |
|---|---|---|---|---|---|---|---|---|
| Eyak N | 5.2 | 0.60 ± 0.09 | 15.3 ± 0.9 | 5.5 ± 0.4 | 13 ± 3 | 0.01 ± 0.01 | 6.5 ± 1.9 | 1.6 ± 0.3 |
| Eyak S | 5.5 | 0.61 ± 0.03 | 16.1 ± 1.3 | 7.0 ± 1.1 | 11 ± 2 | 0.00 ± 0.01 | 5.6 ± 0.5 | 2.0 ± 0.2 |
| Lily | 8.2 | 0.65 ± 0.04 | 13.1 ± 0.8 | 5.9 ± 0.2 | 60 ± 19 | 0.03 ± 0.01 | 3.5 ± 1.0 | 6.0 ± 2.2 |
| Rich Hate Me | 18.3 | 0.57 ± 0.15 | 11.6 ± 2.9 | 6.1 ± 0.4 | 56 ± 7 | 0.03 ± 0.01 | 2.1 ± 0.5 | 24 ± 5 |
| Scott S | 13.4 | 0.81 ± 0.07 | 14.2 ± 0.9 | 6.3 ± 0.3 | 61 ± 37 | 0.03 ± 0.02 | 2.1 ± 0.6 | 54 ± 17 |
| Storey N | 4.6 | 0.56 ± 0.04 | 16.8 ± 1.0 | 6.9 ± 0.4 | 74 ± 11 | 0.04 ± 0.01 | 11 ± 0.6 | 4.5 ± 0.7 |
| Storey S | 2.1 | 0.60 ± 0.04 | 16.6 ± 2.4 | 7.3 ± 0.7 | 70 ± 6 | 0.03 ± 0.00 | 4.2 ± 0.3 | 7.9 ± 0.5 |
| Tiedeman N | 5.5 | 0.66 ± 0.03 | 16.6 ± 1.1 | 6.0 ± 0.5 | 13 ± 3 | 0.01 ± 0.01 | 6.7 ± 0.7 | 1.8 ± 0.2 |
| Tiedeman S | 5.5 | 0.73 ± 0.03 | 15.4 ± 1.4 | 6.7 ± 0.6 | 8.8 ± 1.7 | 0.00 ± 0.00 | 5.2 ± 0.6 | 1.7 ± 0.3 |
| Brackish A | 2.4 | 0.22 | 14.9 | 6.9 | 190 | 0.09 | 0.79 | 130 |
| Brackish B | 2.7 | 0.30 | 17.6 | 7.6 | 19000 | 11 | 0.97 | 8100 |
| Brackish C | 0 | 0.38 | 17.4 | 7.8 | 20000 | 12 | 1.2 | 8900 |
| Brackish D | 0 | 0.49 | 16.0 | 7.6 | 570 | 0.28 | 0.68 | 37 |
| Brackish E | 0 | 0.35 | 15.3 | 7.6 | 2100 | 1.1 | 0.67 | 1100 |
| Brackish F | 0.6 | 0.73 | 11.3 | 6.4 | 1500 | 0.78 | 0.88 | 250 |
| Brackish G | 2.4 | 0.78 | 12.2 | 6.8 | 160 | 0.07 | 0.78 | 11 |
| Brackish H | 2.7 | 0.65 | 12.9 | 7.5 | 13000 | 7.7 | 1.7 | 6000 |
| Brackish I | 0.3 | 0.80 | 12.2 | 7.5 | 24000 | 15 | 2.2 | 8300 |
| Brackish J | 2.7 | 0.89 | 13.1 | 7.6 | 3800 | 2.0 | 21 | 930 |

**Table 2.** Sediment chemical characteristics (mean ± sd) of the wetlands in the Copper River Delta including sediment organic matter (SOM %), total sediment carbon, and porewater (PW) concentrations of acetate, nitrate, and sulfate as well as total sulfate availability in the slurry incubations. For the freshwater wetlands, measurements were conducted in June and August 2014 (5 sites x 2 time periods, n =10 per wetland). For the brackish wetlands, sediment chemistry is from one measurement

5    only. Tidal brackish wetlands A–E were sampled in June and F–J were sampled in August. All chemistry parameters were converted to the total amount of anion per gram of dry sediment (nmol g$^{-1}$) for analyses, but standard porewater concentrations (µM) are also reported for comparison with other studies. Porewater nitrate concentrations were extremely low with many below what we considered our detection limit (i.e., < 2 µM).

| Wetland | SOM (%) | Total Sediment C (mmol g$^{-1}$) | PW Acetate (nmol g$^{-1}$) / (µM) | | PW Nitrate (nmol g$^{-1}$) / (µM) | | PW Sulfate (nmol g$^{-1}$) / (µM) | | Total Sulfate (nmol g$^{-1}$) |
|---|---|---|---|---|---|---|---|---|---|
| Eyak N | 2.0 ± 0.5 | 0.81 ± 0.22 | 57 ± 57 | 360 ± 350 | 1.2 ± 0.7 | 9.6 ± 9.9 | 150 ± 160 | 970 ± 950 | 160 ± 160 |
| Eyak S | 1.8 ± 0.5 | 0.71 ± 0.21 | 18 ± 13 | 120 ± 82 | 1.4 ± 1.0 | 4.3 ± 3.1 | 65 ± 48 | 500 ± 350 | 67 ± 48 |
| Lily | 2.1 ± 0.5 | 0.86 ± 0.21 | 58 ± 43 | 620 ± 560 | 0.86 ± 0.26 | 0.4 ± 0.2 | 5.1 ± 3.2 | 49 ± 27 | 11 ± 3 |
| Rich Hate Me | 3.1 ± 3.9 | 1.3 ± 1.6 | 29 ± 44 | 110 ± 140 | 3.7 ± 4.9 | 2.2 ± 4.2 | 60 ± 130 | 96 ± 110 | 84 ± 140 |
| Scott S | 1.5 ± 2.2 | 0.60 ± 0.89 | 31 ± 34 | 300 ± 340 | 2.2 ± 2.4 | 0.8 ± 0.8 | 11 ± 11 | 120 ± 110 | 51 ± 14 |
| Storey N | 1.8 ± 0.2 | 0.73 ± 0.09 | 10 ± 5 | 120 ± 64 | 0.92 ± 0.39 | 1.2 ± 0.8 | 18 ± 11 | 210 ± 160 | 22 ± 12 |
| Storey S | 1.9 ± 3.1 | 0.76 ± 1.2 | 15 ± 16 | 160 ± 120 | 0.58 ± 0.39 | 2.8 ± 3.1 | 39 ± 44 | 450 ± 460 | 46 ± 45 |
| Tiedeman N | 2.8 ± 2.9 | 1.1 ± 1.2 | 25 ± 14 | 190 ± 95 | 1.7 ± 1.2 | 3.7 ± 2.8 | 56 ± 43 | 420 ± 320 | 93 ± 81 |
| Tiedeman S | 2.3 ± 0.8 | 0.93 ± 0.32 | 17 ± 7 | 120 ± 45 | 2.1 ± 1.9 | 4.3 ± 3.4 | 66 ± 53 | 440 ± 320 | 67 ± 53 |
| Brackish A | 14 | 5.5 | 240 | 2500 | 1.2 | 12 | 660 | 6900 | 770 |
| Brackish B | 1.7 | 0.70 | 72 | 230 | 0.03 | 0.10 | 3000 | 9500 | 8900 |
| Brackish C | 10 | 4.1 | 180 | 2400 | 0.01 | 0.10 | 1200 | 15000 | 9000 |
| Brackish D | 2.3 | 0.94 | 61 | 950 | 0.01 | 0.10 | 630 | 9900 | 660 |
| Brackish E | 1.9 | 0.75 | 110 | 1000 | 0.01 | 0.10 | 1100 | 10000 | 2100 |
| Brackish F | 7.3 | 2.9 | 140 | 1500 | 0.01 | 0.10 | 540 | 5900 | 740 |
| Brackish G | 3.3 | 1.3 | 660 | 6400 | 0.01 | 0.10 | 430 | 4200 | 440 |
| Brackish H | 9.0 | 3.6 | 200 | 1500 | 0.01 | 0.10 | 2500 | 19000 | 7600 |
| Brackish I | 13 | 5.3 | 38 | 550 | 0.01 | 0.10 | 1200 | 17000 | 7500 |
| Brackish J | 1.3 | 0.54 | 25 | 380 | 0.01 | 0.10 | 670 | 10000 | 1400 |

**Table 3.** Generalized linear models (GLMs) wherein log-transformed $CH_4$ production rate is the response variable and ecosystem type (non-tidal freshwater or tidal brackish), time period (June or August), porewater acetate level, and total sulfate availability are potential factors. Positive ($\uparrow$) or negative effects ($\downarrow$) of continuous factors are indicated. Models are ranked in order of the lowest Akaike information criterion corrected for low samples sizes ($AIC_c$) along with delta $AIC_c$ ($\Delta_i$) and Akaike weights ($\omega_i$) before and after model averaging (MA). Models with a $\Delta_i$ larger than 4 were not included in the model averaging. The three models with a larger $AIC_c$ than the null model (intercept only) are not presented.

| Model # | GLM | $AIC_c$ | $\Delta_i$ | $\omega_i$ | $\omega_{i\,(MA)}$ |
|---|---|---|---|---|---|
| 1 | ecosystem + time period + acetate ($\uparrow$) + sulfate ($\downarrow$) | 125.3 | 0.0 | 0.571 | 0.61 |
| 2 | ecosystem + acetate ($\uparrow$) + sulfate ($\downarrow$) | 127.0 | 1.7 | 0.244 | 0.26 |
| 3 | ecosystem + time period + acetate ($\uparrow$) | 128.4 | 3.1 | 0.120 | 0.13 |
| 4 | ecosystem + acetate ($\uparrow$) | 129.7 | 4.4 | 0.062 | - |
| 5 | ecosystem + time period + sulfate ($\downarrow$) | 137.7 | 12.4 | 0.001 | - |
| 6 | ecosystem + sulfate ($\downarrow$) | 139.2 | 13.9 | 0.001 | - |
| 7 | time period + sulfate ($\downarrow$) | 140.2 | 14.9 | 0 | - |
| 8 | sulfate ($\downarrow$) | 140.9 | 15.6 | 0 | - |
| 9 | time period + acetate ($\uparrow$) + sulfate ($\downarrow$) | 141.4 | 16.2 | 0 | - |
| 10 | acetate ($\uparrow$) + sulfate ($\downarrow$) | 141.4 | 16.2 | 0 | - |
| 11 | ecosystem + time period | 142.9 | 17.6 | 0 | - |
| 12 | ecosystem | 144.2 | 19.0 | 0 | - |
| 13 | null | 158.2 | 33.9 | 0 | - |

**Table 4.** Generalized linear models (GLMs) wherein $\Delta CH_4$ production rate (treatment minus control) is the response variable and the macrophyte species added (buckbean, horsetail, lily, or marestail), porewater acetate availability, and total sulfate availability are potential factors. Positive ($\uparrow$) or negative effects ($\downarrow$) of continuous factors are indicated. Models are ranked in order of the lowest Akaike information criterion corrected for low samples sizes ($AIC_c$) along with delta $AIC_c$ ($\Delta_i$) and Akaike weights ($\omega_i$) before and after model averaging (MA). The null model (intercept only) was not included in the model averaging, and the three models with a larger $AIC_c$ than the null model are not presented.

| Model # | GLM | $AIC_c$ | $\Delta_i$ | $\omega_i$ | $\omega_{i\ (MA)}$ |
|---|---|---|---|---|---|
| 1 | acetate ($\downarrow$) | 286.4 | 0.0 | 0.429 | 0.52 |
| 2 | acetate ($\downarrow$) + species | 288.2 | 1.8 | 0.178 | 0.22 |
| 3 | acetate ($\downarrow$) + sulfate ($\uparrow$) | 288.9 | 2.5 | 0.121 | 0.15 |
| 4 | species | 289.4 | 3.0 | 0.096 | 0.12 |
| 5 | null | 290.0 | 3.6 | 0.072 | - |

**Figure Captions**

**Figure 1.** Conceptual diagram illustrating the potential effects of warming, sea-level rise, and increased organic matter (OM) availability on $CH_4$ production in coastal wetlands. These three global change mechanisms are all indirect consequences of rising $CO_2$ levels.

**Figure 2.** Aerial photo of the Copper River Delta taken by the USDA Forest Service depicting the major wetland ecosystem types extending from glaciers to ocean.

**Figure 3.** Mean $CH_4$ production rates (nmol $g^{-1}$ of dry sediment $day^{-1}$) from Copper River Delta non-tidal freshwater (n = 9) and tidal brackish (n = 5) wetlands during A) June and B) August, 2014. Error bars represent standard errors.

**Figure 4.** Mean $CH_4$ production rates (nmol $g^{-1}$ of dry sediment $day^{-1}$) from non-tidal freshwater wetland sediments incubated with freshwater (FW/FW; n = 5) and other sediments from the same freshwater wetlands incubated with brackish water from tidal brackish wetlands (FW/BR; n = 5). Error bars represent standard errors. This seawater addition experiment was conducted over a 14-day period in June 2014.

**Figure 5.** Mean $CH_4$ production rates (nmol $g^{-1}$ of dry sediment $day^{-1}$) from organic matter treatments (CTL = control, BB = buckbean *Menyanthes trifoliata*, HT = horsetail *Equisetum variegatum*, LI = lily *Nuphar polysepalum*, and MT = marestail *Hippuris vulgaris*) replicated in five non-tidal freshwater wetlands during August 2014. Error bars represent standard error.

**Figure 6.** Actual response of $\Delta CH_4$ production (treatment–control; nmol $g^{-1}$ of dry sediment $day^{-1}$) plotted against the predicted response from model-averaged parameter estimates of the macrophyte species added (BB= buckbean *Menyanthes trifoliata*, HT = horsetail *Equisetum variegatum*, LI = lily *Nuphar polysepalum*, and MT = marestail *Hippuris vulgaris*), porewater acetate availability, and total sulfate availability. The dashed black line depicts the 1:1 line, and above the gray dotted line marks the point at which adding organic matter increased $CH_4$ production (or $\Delta CH_4$ production > 0). The solid black line is the best-fit line between the actual and the predicted responses (y = 0.95x – 86; $r^2$ of 0.59), which demonstrates that although the model did a decent job of predicting relative changes in the response, it tended to underestimate $\Delta CH_4$ production.

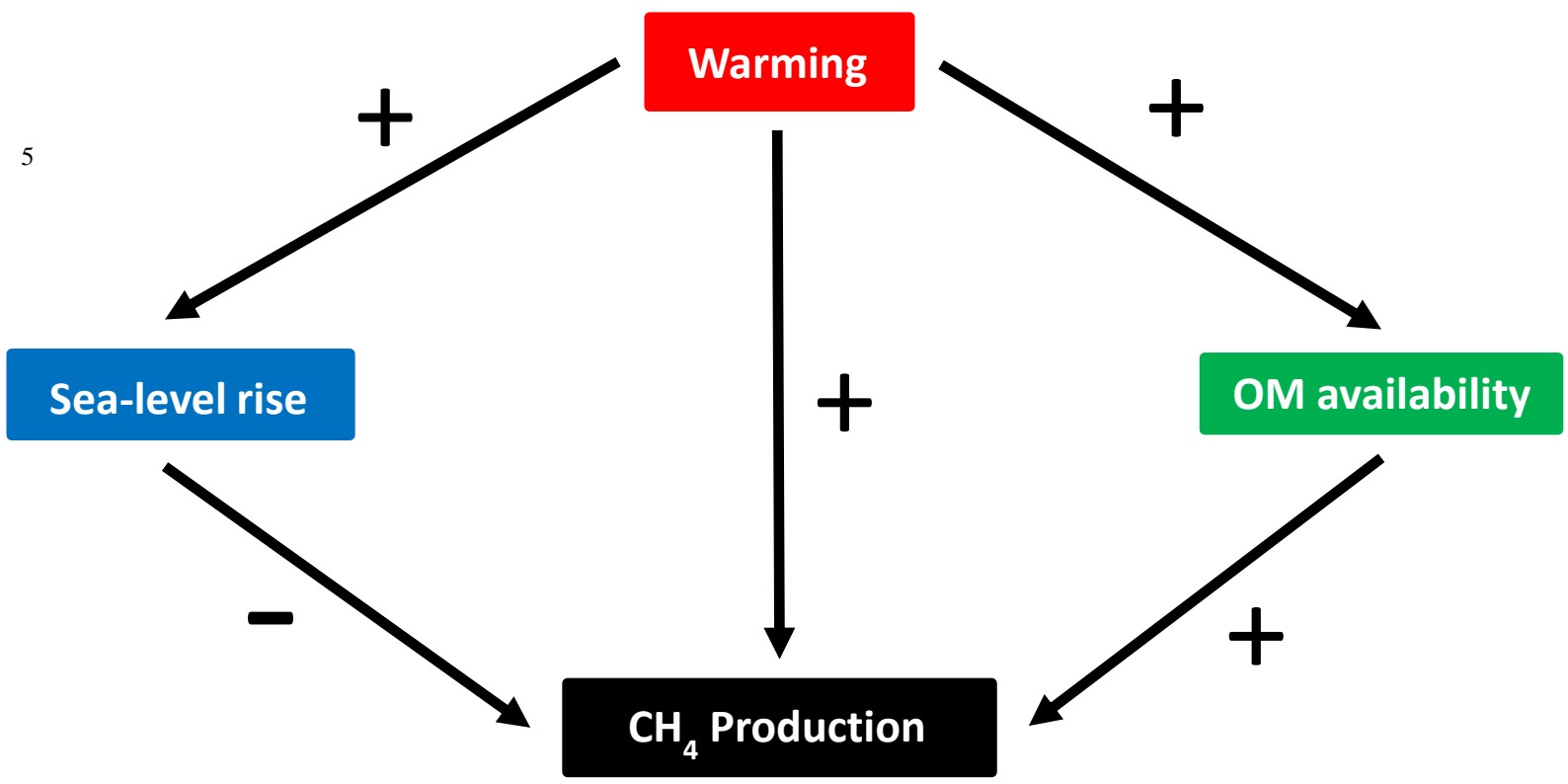

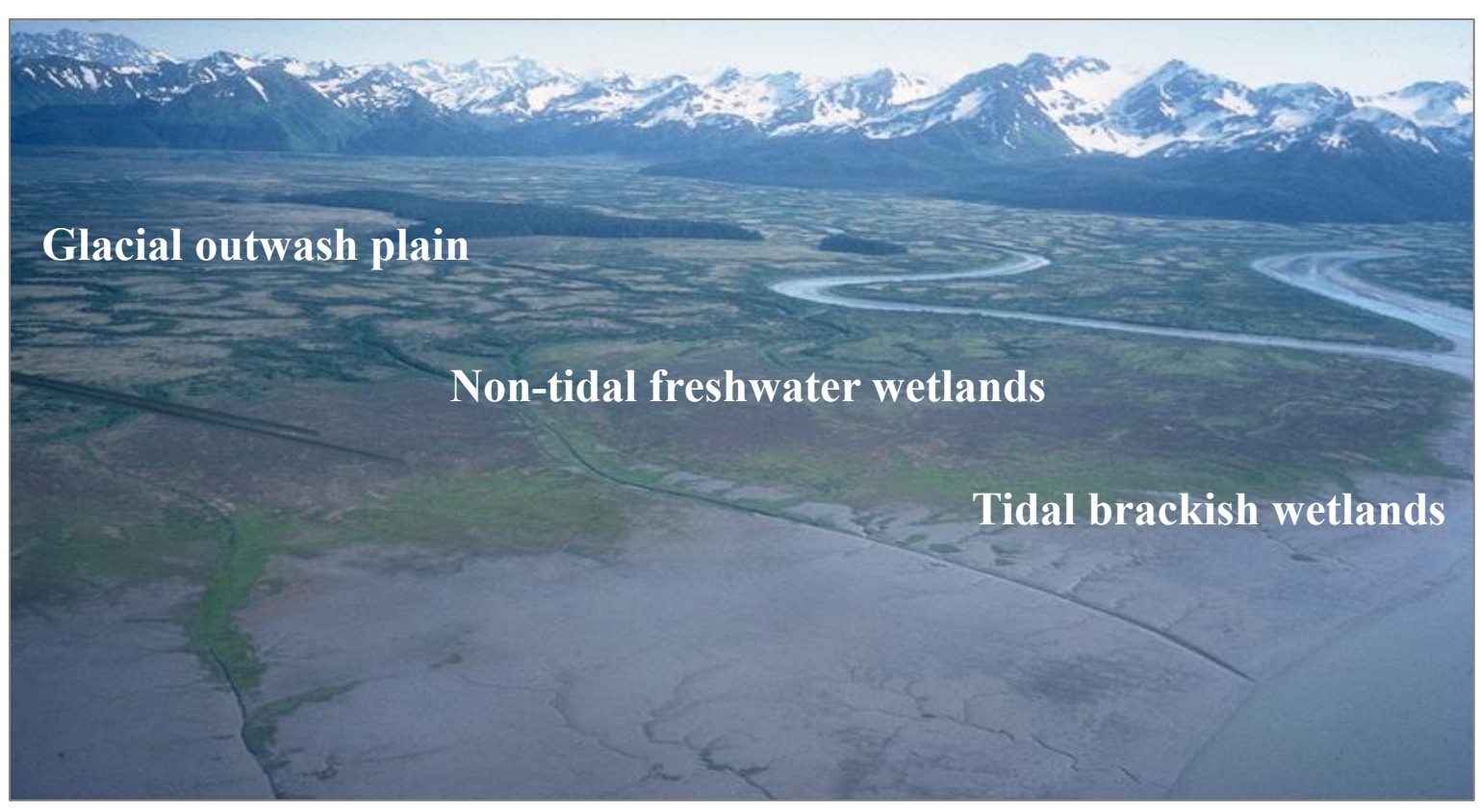

Glacial outwash plain

Non-tidal freshwater wetlands

Tidal brackish wetlands

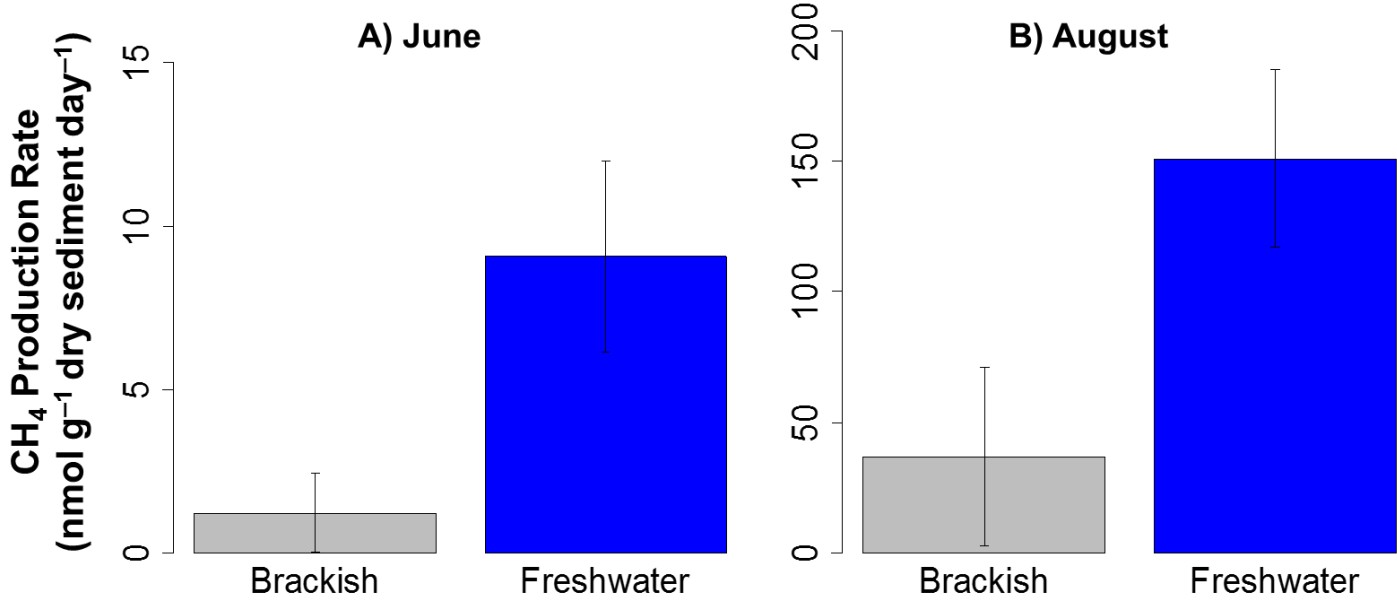

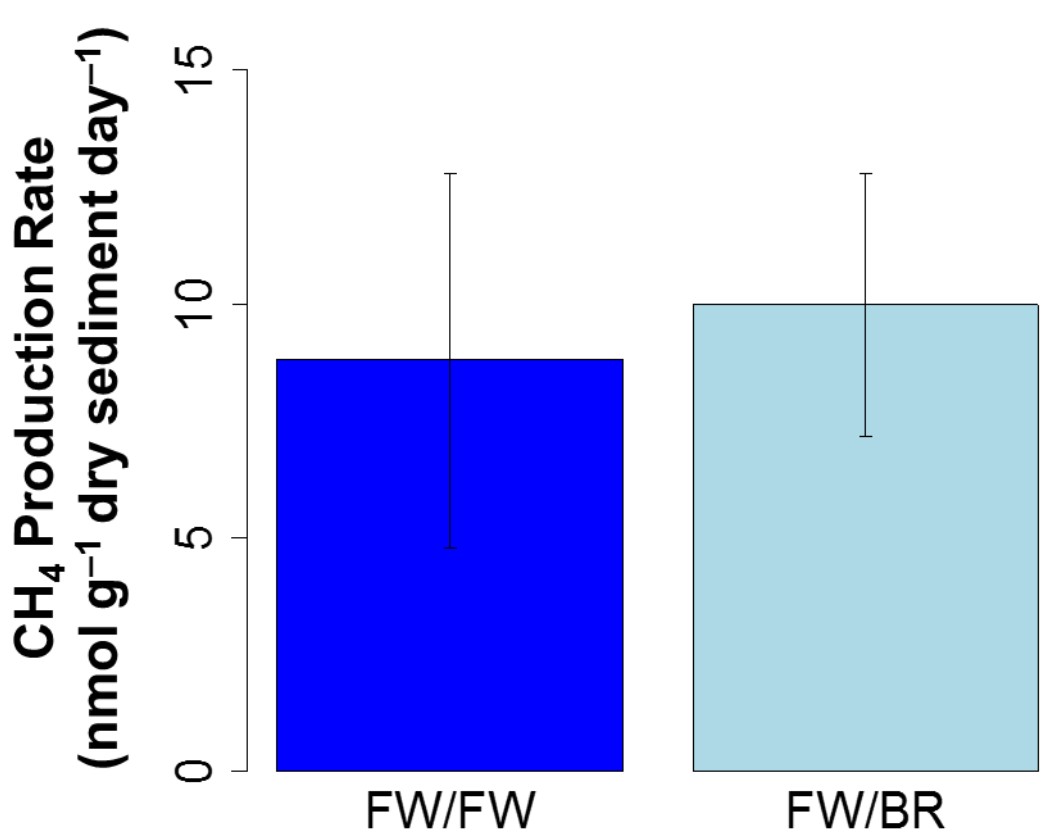

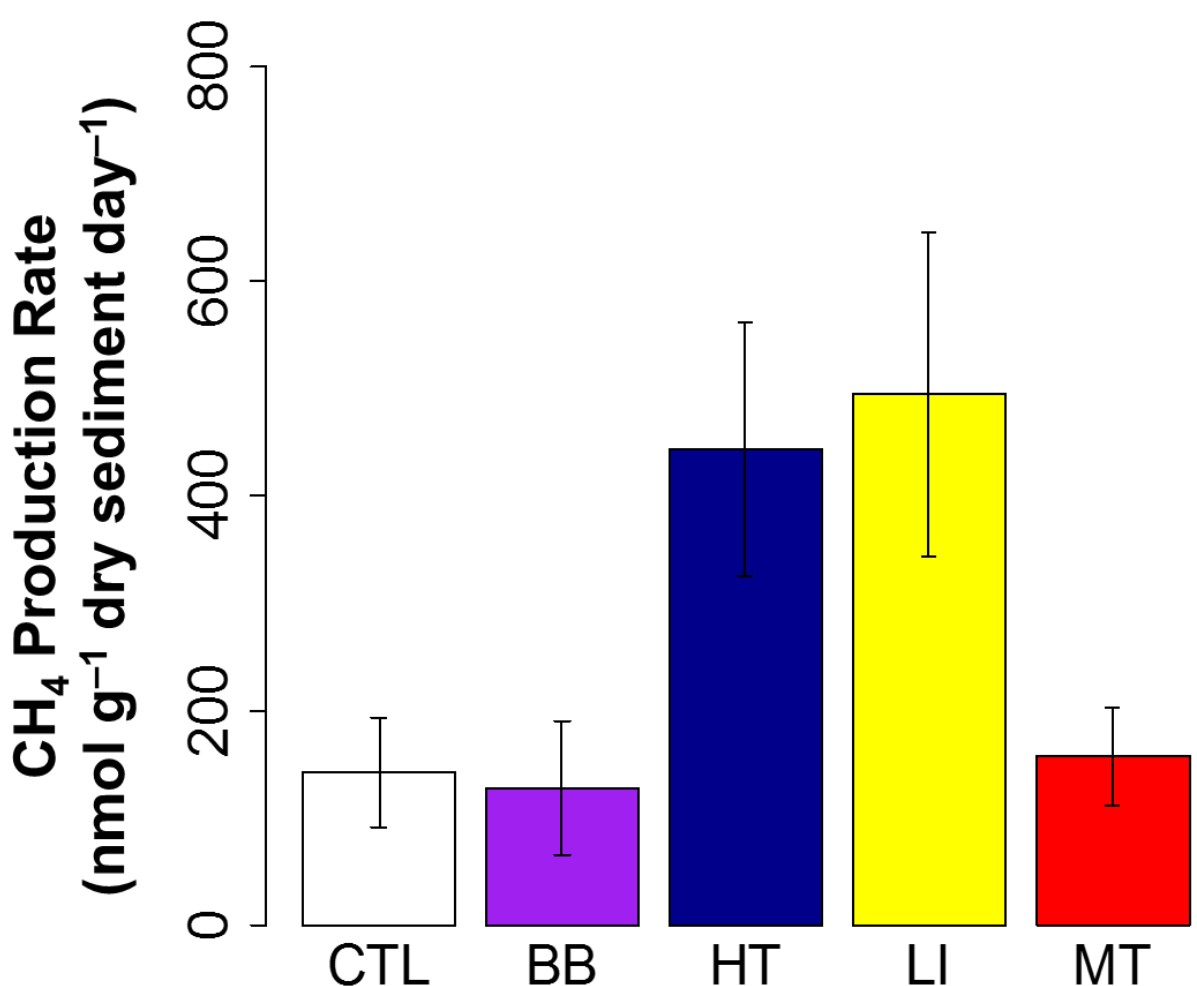

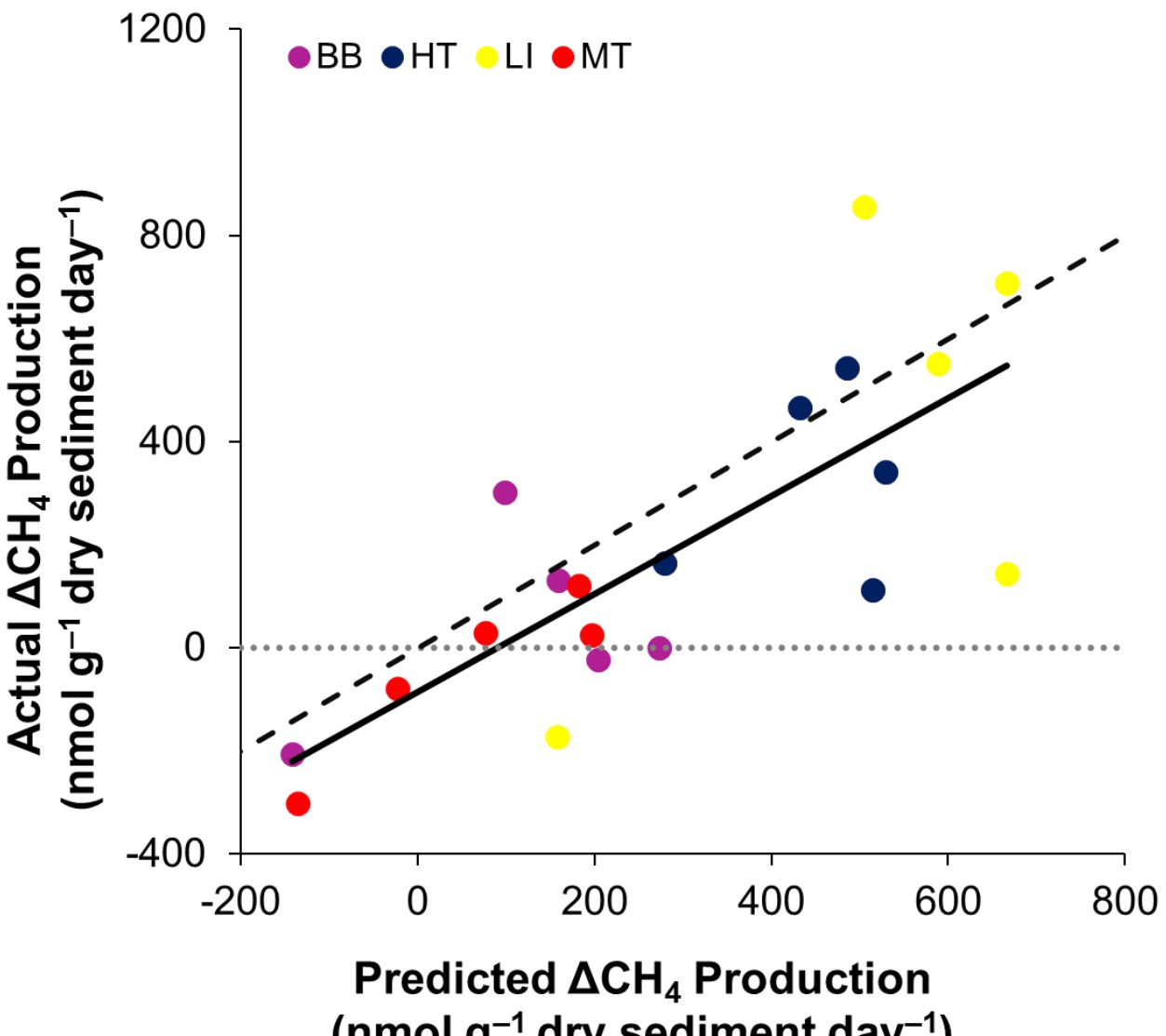