# Peer review of "Regulators of coastal wetland methane production and responses to simulated global change"

_Biogeosciences, 2016_

## Short Comment (SC1) · 18 Aug 2016

Vizza et al: "Regulators of coastal wetland methane production and responses to simulated global change" Biogeosciences Discussions doi:10.5194/bg-2016-314

This paper describes a number of measurements and experiments made on Copper River Delta sediment with the goal of determining whether methane production from these sediments is enhanced by factors which could be influenced by climate change (in particular sea level rise and increased input of presumably labile organic matter). Unfortunately I do not think the authors make a good case that their results add much to the discussion.

The basic premise on which the authors base their study is that methane production depends on substrates produced during fermentation of organic matter and that

methane production and sulfate reduction usually do not occur in the same sediment, presumably because of competition for these substrates. The authors mimic sea level rise by adding brackish water to their samples (which ignores the possible importance of increased water logging and decreased oxygen in the sediments). All incubations are done after vials are flushed with nitrogen although the natural sediments apparently all had some level of oxygen present in situ. Values for nitrate (another potential substrate for carbon remineralizing organisms) and for acetate (the putative important substrate for methanogens) were measured but data were apparently quite variable and are not reported. No measurements were made of H2, another potential substrate for methanogens and methanogens were not examined to see whether they were actually acetoclastic or hydrogenoclastic. The data for the dsrA and mcrA genes are again not presented and in the results section appear to contract the conclusions made about their abundance. The authors also ignore the fact that numbers of genes do not directly relate either to number of cells (a cell can have more than one copy of a gene) or to gene activity. Finally the authors use natural organic matter to enrich their incubations but do not indicate the amount of carbon added or the relative lability of that carbon. (Clearly a gram of sucrose and a gram of twigs would not be expected to stimulate microbial activity to the same extent).

Below I provide a number of specific comments

P2 Line 6ff: I do not like the term green house compensation point. It is not widely used and does not directly relate to what you are measuring (as you say nothing about carbon sequestration in this system). You do not mention carbon fertilization again, and there is no indication in the text of whether any increases in carbon production in the CRD system would be due to warming or to carbon fertilization. I think you hurt yourself by trying to draw connections to too many issues. I would recommend drastic simplification of this section and sticking to the facts.

P2 Line 16: redox conditions are only indirectly related to climate change. You seem to imply the mere presence of nitrate or sulfate changes redox conditions but this is not

true, especially if oxygen is present. In fact there can be a lot of nitrate and sulfate in oxic surface sediments. Did you ever have sulfide in your samples? Does oxygen penetration vary with ecosystem? How far below the surface were the samples collected?

P2 Line 23: I don't think an "abundant supply of organic matter can reduce competition for methanogens by increasing substrate availability". Try instead "abundant supply of organic matter can increase substrate availability"

P2 Line 28: replace "are likely results of" with "may be influenced by"

P3 lines 7 and 10. Use same unit for sealevel rise (100 and 170 cm)

P3 line 15: Numbers of methanogens not as important as whether or not the methanogens are active.

P4 line4: The range of physicochemical parameters in table 1 are actually pretty small for most measurements. Perhaps more important is whether the intertidal sediments are exposed to the atmosphere at low tide (tidal range?). How long are they submerged? What is the water content? Again it matters how far below the surface these sediments were collected. From the table it must be shallow since there was more O2 in these sediments than the freshwater wetlands.

P4 line 26: Were no replicates run for sediments from a single site? How can you tell if observed variability is just typical of replicate samples? I would also think you MUST indicate how much macrophyte tissue you added (probably in terms of gC/g sediment or something like that) to even know if these treatments were similar since the lability of the carbon is likely not the same.

P5 Line 6: You mean incubation temperature not ambient temperature?

P5 line 8: Purging with nitrogen will likely have a bigger effect than incubating at a few degrees cooler than the actual sediment. I would expect stimulation by this as you allow more anaerobic processes to occur. Were any blanks or controls run?

P5 line 12: "flame ionization" not "flame ionizing" detector

P5 line 20: You report that only sulfate was detectable in the water column. Did you present any water column data? Do they mean anything? What are detection limits for pore waters? Present nitrate and acetate data for sediments as well as sulfate.

P5 line 27: the method you describe is typically called "loss on ignition" and represents a loss of organic matter. Did you convert to organic C? Is this data reported anywhere in the paper? Was DOC measured the same way? Blanks? Detection limit?

P6 line 2: I would use the word "converted" rather than "scaled"

P6 line 5: I couldn't tell what this composite sediment was used for. Were incubations for each site or was all sediment made into a composite? Why do this? It seems that then the averages for the genes refer to different samples than the physiochemical properties. Can you properly do statistics comparing the combined samples in one parameter to the individual samples in one parameter?

P7 line 2: move "log transformed" to immediately follow "four factors"

P7 line 26ff: You use a lot of significant figures for something that is so variable. Maybe you are justified in two significant figures but not 3 or more. Again blanks and detection limits need to be mentioned as your averages are pretty close to zero (or at least almost include zero). I would like to see these data associated with specific samples.

P8 line 14: This paragraph seemed very odd. You seem to contradict yourself a lot. For intertidal samples, three had no dsrA but this was found in all freshwater AND the dsrA was independent of ecosystem type. This seems to contradict your hypothesis which you nevertheless cling to. Remember the presence of a gene doesn't mean the organisms are doing anything at the moment and the number of genes does not necessarily indicate the number of cells of a particular organisms.

P8 line 27ff: I really don't like your equating sealevel rise purely with sulfate concentration. It is ok to say that sealevel rise will flood current intertidal areas, but the vegetation

will change and the water will be permanently water logged rather than periodically exposed to air. You have no data on estuarine wetlands to compare to the freshwater wetlands (although others have done this comparison in other systems). I would guess the reason you see no effect here is that you are mimicking the process the wrong way. You are really looking at increased sulfate levels, not sealevel rise.

P10, line 2: The sentence beginning "our study also demonstrates..." seems to directly contradict your own data as you said before that dsrA did not correlate with ecosystem type

P10 line 13ff and next paragraph: These two sections are wild speculation with absolutely no data behind them. There are a lot of other factors that might be important that you haven't included.

P11 line14: This whole section is hurt by the fact that you don't characterize the macrophytes at all in terms of potential lability. Did you add a constant amount of C or organic N? Or just dry mass? Or just a "chunk" of leaves? Again much of this section is speculation without more facts

P12 line19: You have no idea whether there are hydrogen utilizing methanogens around or how much hydrogen there might be. You can reiterate what other people have said in other papers but I don't see you connecting these to your system with any facts.

Modeling: I didn't really understand the models. If you are making linear equations including various parameters and then constant factors for each, I would like to see more detail on how this worked. If you have a lot of variables, you can fit most data but figure 6 is completely mysterious and doesn't convince me of the value of your model. I also didn't understand the columns in tables 2 and 3. I have never heard of an AIC before, for example. Explain the statistics a bit to an audience that may include non-biologists.

---

## Author Comment (AC1) · 25 Aug 2016

Dear Dr. Scranton,

Thank you for taking the time to read and comment on our manuscript. We appreciate your observations and questions. Below is our response to your comments.

**Comment:** "The basic premise on which the authors base their study is that methane production depends on substrates produced during fermentation of organic matter and that methane production and sulfate reduction usually do not occur in the same sediment, presumably because of competition for these substrates." **Response:** In this comment, you seem to imply that methane production and sulfate reduction do not occur in the same sediment. Methanogens and sulfate-reducing bacteria do directly compete for organic substrates such that when sulfate or other more energetically profitable alternative electron acceptors are available, some methane is still produced, but it is at lower rates (please see manuscript citations: Achtnich et al. 1995; Lovley and Klug 1983, 1986).

**Comment:** "The authors mimic sea level rise by adding brackish water to their samples (which ignores the possible importance of increased water logging and decreased oxygen in the sediments)." **Response:** These sediments came from freshwater wetland ponds (0.4 – 1.2 m total water depth) that were already waterlogged and therefore had decreased oxygen in the sediments.  We have no reason to suspect that the physical act of sea-level rise would change the saturation status of the sediments.

**Comment:** "All incubations are done after vials are flushed with nitrogen although the natural sediments apparently all had some level of oxygen present in situ." **Response:** In this comment and throughout your review, you seem to imply that purging vials with nitrogen gas is problematic. It is widespread practice in methane production bottle assays to purge with nitrogen gas in order to remove the oxygen that was introduced by removing sediment from below the water's surface and exposing it to the air in the field and in the lab (Lofton et al. 2014; Sinke et al. 1992; West et al. 2012, 2015).  Of course, it is possible that there are microsites where oxygen is present in the freshwater wetland sediments, but we believe that the same may be true in the bottle experiments.

**Comment:** "Values for nitrate (another potential substrate for carbon remineralizing organisms) and for acetate (the putative important substrate for methanogens) were measured but data were apparently quite variable and are not reported." **Response:** Summary porewater acetate levels are reported in Results section 3.1.1. You are correct in that we did not report actual values for nitrate, and we will add a short line in the results to indicate that nitrate was usually 2+ orders of magnitude lower than total sulfate levels. We measured acetate, nitrate, and sulfate fairly extensively in both the water column and the porewater as well as other parameters.  We did not present all the data we collected because we did not want to overwhelm the reader, but these data are all available in the archived data.  If reviewers and editors feel that these data would enhance the manuscript, we will add them in tabular or figure form to the manuscript or as supplemental information.

**Comment:** "No measurements were made of H2, another potential substrate for methanogens and methanogens were not examined to see whether they were actually acetoclastic or hydrogenoclastic." **Response:** We did not measure $H_2$ production as we do not currently have an instrument available to measure this nor did we classify the type of methanogens.  However, acetate was the predominant factor explaining methane production patterns in almost all experiments, which suggests that a large proportion of the methanogenesis is acetoclastic (e.g., pg. 8 lines 29-30).  However, we do not deny the

potential of hydrogenotrophic methanogenesis occurring in our ecosystems (please see last paragraph on page 10).

**Comment:** "The data for the dsrA and mcrA genes are again not presented and in the results section appear to contract the conclusions made about their abundance. The authors also ignore the fact that numbers of genes do not directly relate either to number of cells (a cell can have more than one copy of a gene) or to gene activity." **Response:** The *dsrA* and *mcrA* summary data are presented in the Results section 3.1.3, and individual gene data per sample are also archived in the data set (see microbial tab). Although we acknowledge that the number of genes does not equate with number of cells or gene activity, qPCR of functional genes for particular guilds is a commonly used approach to estimate the abundance of a functional group. We also acknowledge in the manuscript that it is possible that some of the genes we detect are from dormant microbial communities (please see pg. 11, lines 8-12). The number of genes corresponding to abundance is always going to be a bit of an assumption, but in a broader survey of our wetlands, we have unpublished data from our system that *mcrA* abundance relative to *dsrA* abundance is correlated with methane production rates. Also, in *Letters in Applied Microbiology* (Volume 62, Issue 2, Pages 111-118), Morris et al. 2016 found that hydrogenotrophic methane production rates corresponded to *mcrA* abundance. Although observation of functional gene transcripts or even the actual enzymes catalyzing the reactions would clearly be preferred, logistics associated with our remote field sites precluded work with these more labile macromolecules. Despite challenges associated with interpreting DNA copy number of functional genes, we believe that our use of qPCR substantially adds to the literature as there are few studies that simultaneously measure methane production and microbial functional group abundances. For example, the studies listed on pg. 9, lines 22-26, have hypothesized that microbial community processes are behind methane patterns along a salinity gradient, but none of these studies actually tested this hypothesis by measuring microbial communities.

**Comment:** "Finally the authors use natural organic matter to enrich their incubations but do not indicate the amount of carbon added or the relative lability of that carbon. (Clearly a gram of sucrose and a gram of twigs would not be expected to stimulate microbial activity to the same extent)." **Response:** We added 3 g of live tissue per macrophyte species, which translated to 0.10-0.12 mol of C added to each incubation, and we will add these values to the methods of the manuscript. Dr. Scott Tiegs of Oakland University has measured the % C, % N, and % P of the four litter species we added: Marestail (45% C, 1.7% N, 0.17% P), Buckbean (44% C, 0.94% N, 0.15% P), Lily (45% C, 1.7% N, 0.17% P), and Horsetail (47% C, 2.5% N, 0.24% P), but the patterns we observed did not follow %C, %N, %P, C:N or C:P, which are considered standard indices of litter quality. We discuss the phosphorus in detail because Tiegs et al. 2013 found that litter decomposed more rapidly when P was higher (see pg. 11, lines 14-24). You are correct in that we did not specifically measure the lability of the carbon from each macrophyte species, but we are leveraging information from Tiegs et al. 2013, whose study had already assessed this by measuring decomposition rates of these litter types in Copper River Delta ponds. We will clarify this in the discussion.

**Comment:** "P2 Line 6: I do not like the term green house compensation point. It is not widely used and does not directly relate to what you are measuring (as you say nothing about carbon sequestration in this system)." **Response:** We believe that the term "greenhouse compensation point" accurately describes wetlands' dual roles in carbon sequestration and emissions, and we believe that because northern wetlands are on the edge of this compensation point that it is so important to study how

global change may alter the methane cycle. Greenhouse compensation point is a good way to set up the broader context of this study.

**Comment:** "You do not mention carbon fertilization again, and there is no indication in the text of whether any increases in carbon production in the CRD system would be due to warming or to carbon fertilization. I think you hurt yourself by trying to draw connections to too many issues. I would recommend drastic simplification of this section and sticking to the facts." **Response:** Increased $CO_2$ levels lead to warming as well as $CO_2$ fertilization, both of which could affect the amount of substrate available. We are not precisely sure whether increased organic matter will result from warming, or longer growing seasons, directly or from $CO_2$ fertilization and therefore it seems prudent to mention both potential mechanisms. Regardless, we will reevaluate our discussion of these points based on your recommendation and consider ways to focus our comments.

**Comment:** "P2 Line 16: redox conditions are only indirectly related to climate change. You seem to imply the mere presence of nitrate or sulfate changes redox conditions but this is not true, especially if oxygen is present. In fact there can be a lot of nitrate and sulfate in oxic surface sediments. Did you ever have sulfide in your samples? Does oxygen penetration vary with ecosystem? How far below the surface were the samples collected?" **Response:** True, redox conditions are a potential indirect effect of climate change. Sea-level rise could increase sulfate levels just as higher decomposition rates due to warming could deplete oxygen levels, both of which affect redox conditions. The sediment came from the top 20 cm, and although some oxygen is likely present in small amounts, it is also likely depleted quickly. For example, the layer of water directly above the sediments often has low DO levels of around 1 mg/L, particularly in the evening, and it is not uncommon to have anoxic groundwater upwelling in these systems. We also observe that water column DO levels drop throughout the season as vegetation begins to senesce. As for hydrogen sulfide, we did not directly measure that, but sediment characteristics (black coloration and pungent odor) suggest the presence of sulfide. Usually sediments become anaerobic within the first few cm of freshwater ponds, which is another reason that we make the incubations anaerobic by purging with nitrogen gas. Of course oxygen matters, but the amount of nitrate in particular is orders of magnitude lower than agricultural and other human impacted systems. Lastly, if significant levels of oxygen were present in these sediments, it would kill the methanogens because they are extremely sensitive to $O_2$ (please see manuscript citation: Whalen 2005) and we would therefore see very little methane production in our experiments.

**Comment:** "P2 Line 23: I don't think an "abundant supply of organic matter can reduce competition for methanogens by increasing substrate availability". Try instead "abundant supply of organic matter can increase substrate availability"" **Response:** We appreciate the wording suggestion and will make the change.

**Comment:** "P2 Line 28: replace "are likely results of" with "may be influenced by" " **Response:** We appreciate the wording suggestion and will make the change.

**Comment:** "P3 lines 7 and 10. Use same unit for sealevel rise (100 and 170 cm)" **Response:** We appreciate the wording suggestion and will make the change.

**Comment:** "P3 line 15: Numbers of methanogens not as important as whether or not the methanogens are active." **Response:** Please see our response to your major comment above about qPCR and the feasibility of RNA work.

**Comment;** "P4 line4: The range of physicochemical parameters in table 1 are actually pretty small for most measurements. Perhaps more important is whether the intertidal sediments are exposed to the atmosphere at low tide (tidal range?). How long are they submerged? What is the water content? Again it matters how far below the surface these sediments were collected. From the table it must be shallow since there was more O2 in these sediments than the freshwater wetlands." **Response:** The intertidal sediments were collected from the top 20 cm just as in the freshwater wetlands, and they are covered with freshwater during low tide and increasingly brackish water during high tide. So again these sediments are waterlogged with depleted oxygen levels. In Table 1, the DO data are actually from the water column, as we do not have oxygen data for the sediment. In the freshwater ponds where limnological profiles could be conducted, we reported DO levels from the bottom of the water column, but in intertidal marsh, DO levels came from the surface layer. This is detailed in the legend for Table 1. We will remove the water column DO data altogether since they seem more misleading than helpful.

**Comment:** "P4 line 26: Were no replicates run for sediments from a single site? How can you tell if observed variability is just typical of replicate samples? I would also think you MUST indicate how much macrophyte tissue you added (probably in terms of gC/g sediment or something like that) to even know if these treatments were similar since the lability of the carbon is likely not the same." **Response**: We ran five control replicates without added substrate at each freshwater wetland (n = 5), which we then averaged to form the basis of the delta CH4 production metric. Each macrophyte treatment was replicated in 5 different ponds. In the increased organic matter simulation, we used pond as the replicate because we were more interested in capturing how wetlands differing in biogeochemistry along a glacial to oceanic gradient would respond to organic matter addition rather than how much variability there exists within a single wetland's response.

**Comment:** "P5 Line 6: You mean incubation temperature not ambient temperature?" **Response"** Yes, we do. We appreciate this good suggestion for a wording change.

**Comment:** "P5 line 8: Purging with nitrogen will likely have a bigger effect than incubating at a few degrees cooler than the actual sediment. I would expect stimulation by this as you allow more anaerobic processes to occur. Were any blanks or controls run?" **Response:** Please see our response to your major comment above.

**Comment:** "P5 line 12: "flame ionization" not "flame ionizing" detector" **Response:** We will make this change.

**Comment:** "P5 line 20: You report that only sulfate was detectable in the water column. Did you present any water column data? Do they mean anything? What are detection limits for pore waters? Present nitrate and acetate data for sediments as well as sulfate." **Response:** We do have extensive water column data for these wetlands, some of which are in the archived data. If reviewers feel these data add to the manuscript, we can add them in tabular form or in supplemental information. Some (i.e., DOC and sulfate) are also already presented in Table 1, but acetate and nitrate were not detectible in the water column (we will add concentration detection limits to the manuscript which were about 10 and 5 uM, respectively).

**Comment:** "P5 line 27: the method you describe is typically called "loss on ignition" and represents a loss of organic matter. Did you convert to organic C? Is this data reported anywhere in the paper? Was DOC measured the same way? Blanks? Detection limit?" **Response:** Yes, we took the sediment organic

matter data (in % of OM) from loss on ignition and converted it to organic C (pg. 5, lines 25-29).  We will add in the following citation about conversion to organic C: Thomas et al. 2005; Aquat. Sci. 67:424–433. The data were not reported in the paper as to not overwhelm the reader, especially since the amount of carbon did not affect the results, but the data are archived and available online.  DOC was measured using a Shimadzu TOC-V (pg. 5, lines 18-19), and we did run blanks.  All samples had detectible DOC, but our lowest standard was 1 mg/L and all samples registered above this level with the exception of five of the intertidal samples, which fell in between the blank and the lowest standard.  If reviewers would like more detail on these analyses in the methods, we will add them to the manuscript.

**Comment:** "P6 line 2: I would use the word "converted" rather than "scaled" " **Response:** Thank you for the wording suggestion.

**Comment:** "P6 line 5: I couldn't tell what this composite sediment was used for. Were incubations for each site or was all sediment made into a composite? Why do this? It seems that then the averages for the genes refer to different samples than the physiochemical properties. Can you properly do statistics comparing the combined samples in one parameter to the individual samples in one parameter?" **Response:** The composite sediment was a combination of the five sediment samples from different sites for each freshwater wetland, but this was only done for the microbial analyses.  Making a composite is highly practiced in soil microbial ecology for the purpose of controlling analytical costs while still capturing significant spatial heterogeneity.  We made a composite for each freshwater wetland which were then compared against ten sediment samples from intertidal marsh.  So yes, it is true that we could not directly link methane production of a single sample or its physicochemical properties to the functional group abundances, but overall we were able to characterize methanogen and sulfate-reducer abundances across ecosystems and relate these to average methane production rates.

**Comment:** "P7 line 2: move "log transformed" to immediately follow "four factors"" **Response"** Methane production rates were log-transformed, not the factors. We cannot make this change because it would misrepresent our statistics.

**Comment:** "P7 line 26ff: You use a lot of significant figures for something that is so variable. Maybe you are justified in two significant figures but not 3 or more. Again blanks and detection limits need to be mentioned as your averages are pretty close to zero (or at least almost include zero). I would like to see these data associated with specific samples" **Response:** We will reduce the number of significant figures and make the needed detection statements in the methods.  We only reported parameters if they were detectible.  The reason that these averages are close to zero in this particular line is that we converted actual concentrations to the total amount in the incubation bottle (i.e., µM * PW volume in incubation * (PW volume per ml of sediment) * sediment volume in bottle).

**Comment:** "P8 line 14: This paragraph seemed very odd. You seem to contradict yourself a lot. For intertidal samples, three had no dsrA but this was found in all freshwater AND the dsrA was independent of ecosystem type. This seems to contradict your hypothesis which you nevertheless cling to. Remember the presence of a gene doesn't mean the organisms are doing anything at the moment and the number of genes does not necessarily indicate the number of cells of a particular organisms." **Response:** Indeed, we did not detect *dsrA* in 3 of the 10 intertidal samples.  Nevertheless, *dsrA* abundance tended to be higher in intertidal ecosystems, although not significantly so.  In contrast, the *mcrA* presence and abundance did vary significantly by ecosystem, and methanogens are the group that

we tend to use to explain our methane production results directly, while we use sulfate and sulfate-reducer abundance as supporting evidence (please see pg. 9, lines 14-16).

**Comment:** "P8 line 27ff: I really don't like your equating sealevel rise purely with sulfate concentration. It is ok to say that sea level rise will flood current intertidal areas, but the vegetation will change and the water will be permanently water logged rather than periodically exposed to air. You have no data on estuarine wetlands to compare to the freshwater wetlands (although others have done this comparison in other systems). I would guess the reason you see no effect here is that you are mimicking the process the wrong way. You are really looking at increased sulfate levels, not sealevel rise." **Response:** In these systems, which are waterlogged throughout the year, we expect sulfate concentration to be a major biogeochemical change. Our simulation not only adds sulfate, but it also adds other nutrients and the microorganisms brought in with the saltwater. You are correct that over the long term vegetation and other characteristics of these ecosystems will also change. We will add comments at pg. 3., line 12, and pg. 4 line 20 to reflect our biogeochemical focus on the effects of sea-level rise. Our data on intertidal ecosystems for comparison with freshwater ecosystems can be found in Results section 3.1.2 and Figure 3.

**Comment:** "P10,line2: The sentence beginning "our study also demonstrates..."seems to directly contradict your own data as you said before that dsrA did not correlate with ecosystem type" **Response:** Just because *dsrA* presence and abundance did not significantly correlate with ecosystem type does not mean the presence or abundance of sulfate-reducing bacteria does not affect methane production in intertidal ecosystems. Even though these bacteria are also present in freshwater ecosystems, they could be limited by sulfate availability. Also *dsrA* tended to be higher on average relative to freshwater ecosystems, just not significantly so. We will change the wording to "tended to have generally higher sulfate-reducer abundances when present" to address this concern.

**Comment:** "P10 line 13ff and next paragraph: These two sections are wild speculation with absolutely no data behind them. There are a lot of other factors that might be important that you haven't included." **Response:** We provide potential hypotheses and explanations for our data that were grounded in the literature, which is not an uncommon practice in the discussion. Specifically, we discuss the acetoclastic pathway of methane production and how we might expect that to change in the future as well as possible reasons for why methane production rates varied by an order of magnitude throughout the season. We could omit these sections, but felt that it was important to say something about the results we observed. Regardless, we will reevaluate the nature and extent of our explanations.

**Comment:** "P11line14: This whole section is hurt by the fact that you don't characterize the macrophytes at all in terms of potential lability. Did you add a constant amount of C or organic N? Or just dry mass? Or just a "chunk" of leaves? Again much of this section is speculation without more facts" **Response:** Please see our response above to your major comment about lability of macrophytes. We evaluated many different possibilities for explaining the results of the different macrophyte treatments including % C, % N and % P, but the quality measurement that seemed to matter the most was antimicrobial properties. None of these potential indicators of lability appeared to have an effect on methane production, which we will clarify in the Discussion pg 11, lines 14-30, and pg 12, lines 1-9.

**Comment:** "P12 line19: You have no idea whether there are hydrogen utilizing methanogens around or how much hydrogen there might be. You can reiterate what other people have said in other papers but I

don't see you connecting these to your system with any facts." **Response:** We will try to improve the clarity of the manuscript, by acknowledging that methanogens in general utilize acetate, directly (acetoclastic) or indirectly (hydrogenotrophic). We will also add that $CO_2$ and $H_2$ are compounds that also come from fermentation of organic matter such as acetate. Hence, this is why acetate is important and was measured in this study.

**Comment:** "Modeling: I didn't really understand the models. If you are making linear equations including various parameters and then constant factors for each, I would like to see more detail on how this worked. If you have a lot of variables, you can fit most data but figure 6 is completely mysterious and doesn't convince me of the value of your model. I also didn't understand the columns in tables 2 and 3. I have never heard of an AIC before, for example. Explain the statistics a bit to an audience that may include non-biologists." **Response:** Akaike Information Criterion (AIC) is an increasingly utilized form of model selection that generates estimates of the model being the best representation given the data and the set of models being explored.  It also penalizes for the number of parameters in the model.  We will add more description of this in the methods. Basically, general linearized models are created, each of which is associated with an AIC value, which is then used to rank the models. We also corrected for small sample sizes ($AIC_c$). The model with the lowest $AIC_c$ value is considered the best, and all remaining models are compared relative to the best approximating model using delta $AIC_c$ ($\Delta_i$).  Models with a $\Delta_i$ less than or equal to 2 are considered to have substantial support, while models having a $\Delta_i$ greater than 7 have little support (Burnham and Anderson 2002).  The relative strength of the models was then evaluated with Akaike weights ($\omega_i$), which indicate the probability of a model being the best model, given the data and the set of candidate models (Burnham and Anderson 2002).

Respectfully,

Carmella Vizza, Will West, Stuart Jones, Julia Hart, and Gary Lamberti

---

## Short Comment (SC2) · 30 Aug 2016

My sense from reading your responses to my comments is that there is in part a "language" issue, perhaps related to the difference between someone used to looking at methane cycling in the marine environment rather than the freshwater environment. For example although methanogens and sulfate reducers may directly compete for substrates, typically marine scientists assume that sulfate reducers will win this competition to the extent that methane production is VERY low in the presence of sulfate reducers. So technically, yes, they do coexist, but the whole idea is that one can effectively ignore methanogens until sulfate is depleted (and this ignores the possible presence of consortia where the sulfate reducers are acting as methane oxidizers). Similarly, nitrate reduction is assumed to go to completion before sulfate reduction kicks in. This is true even if nitrate levels are orders of magnitude lower than sulfate

levels (commonly true in marine systems). Your data do not really permit assessing this paradigm since there are no profiles and samples are homogenized bulk samples. Pooled concentration data really can't be used in situations where gradients are large and microenvironments exist as the microorganisms respond to the exact chemistry of their specific environment, not to pooled values. Similarly the use of pooled samples for genetic work confuses things, as the various subsamples being pooled could have different chemistries. (I also think it is confusing in the paper whether you are reporting data from sediments or overlying water column. Concentrations of all redox species will change rapidly below the sediment/water interface and be strongly effected by periodic exposure to air.)

I know that it is common practice to purge bottles with nitrogen to get methane production rates, but that does not mean that rates measured in purged bottles give you in situ rates.

To me sea level rise implies higher water levels implying inundation of wetlands, raising both water levels and sulfate concentrations. It was unclear from your paper the extent to which the wetlands you sampled were continuously submerged or whether tidal influences exposed the wetlands at low tide. I assumed from your figure that the wetlands were exposed meaning that at least some of the surface sediments periodically had oxygen added. How far into currently purely freshwater wetlands will the sea advance with sealevel change? This is likely to have a bigger effect (and totally change the ecology of the plants as well).

I don't think it makes sense to say that because acetate correlated with your results that hydrogen is necessarily unimportant. Perhaps if you had included hydrogen data these modelling results would have been different? In this, and a number of other places, more careful language would at least convince the reader that you had considered an issue.

I apologize but I don't have the time to go through the remainder of your responses in

detail, and in any case it is hard to tell if you have appropriately modified the text without actually seeing it. I still feel that the paper does NOT tell us a lot about the effect of sealevel rise or climate change (actually temperature increase), but does contribute to an understanding of how increasing sulfate concentrations might change methane production (but not methane or carbon dioxide flux to the atmosphere)

---

## Referee Comment (RC1) · S. Neubauer (Referee) · 31 Aug 2016

General comments:

In this manuscript, Vizza and colleagues were interested in investigating CH4 production in freshwater and brackish wetlands of the Copper River delta in Alaska, USA. They made measurements of CH4 production of sediment slurries and also measured a range of physicochemical parameters in order to assess what factors were most important in controlling CH4 production. The authors also conducted two separate experiments to look at how various changes expected due to climate change – namely, increased salinity and increased organic matter inputs to the sediment – would affect CH4 production. The authors concluded that CH4 production was higher in freshwater vs. brackish marshes, a finding that has been reported elsewhere, and that sulfate

concentrations and the composition of the microbial community played important roles in those ecosystem-related differences in CH4 production. When salinity and sulfate levels were experimentally raised, there was (unexpectedly) no change in CH4 production rates, which was attributed to a slow response of the microbial community. After adding organic matter from four different plant species to sediment slurries, the authors reported that CH4 production rates increased for two of the species but did not change for the other two.

I was very interested in the topics covered by this manuscript and think that the measurements and experiments can provide some insight into questions about controls of CH4 production and how global changes will affect CH4 dynamics. However, I was a bit disappointed in the analysis and interpretation of the data, especially in how the authors tied their findings to existing knowledge. For example: One of the global changes studied in this manuscript was the movement of saline water into freshwater wetlands, as should occur due to sea level rise. Although the authors mentioned several studies that have examined CH4 emissions along existing estuarine salinity gradients, they did not include any papers from the (steadily-growing) body of literature that has used focused experiments in the field and lab to determine how carbon cycling (including rates of CH4 production) responds to saltwater intrusion. As a starting point, you should look at recent work (last ∼5 years) by Lisa Chambers, Nathaniel Weston, John Marton, Gijs van Dijk, Ashley Helton, and myself. A 2015 wetland salinization review paper in Ecosphere (Ellen Herbert = primary author) is another good place to consult. These papers will help the authors place their findings in the context of what is already known about the effects of saltwater moving into freshwater wetlands. Similarly, I felt that the authors could have done a better job exploring the literature on how plants and organic matter inputs affect CH4 production.

Beyond those issues of interpretation, I had several comments and questions about the experimental design and the statistical analyses that were conducted. Those comments, plus some others, are listed below.

Specific comments: 1) p. 1, line 13; p. 3, line 12; and throughout manuscript, "freshwater and intertidal wetlands". As someone who studies tidal freshwater wetlands, I take issue with the way you are classifying wetlands as either freshwater *or* intertidal. Wetlands can be both! Indeed, there are over 19,000 ha of tidal *and* freshwater wetlands within the Copper River Delta (see Hall's chapter in the 2009 "Tidal Freshwater Wetlands" book; Barendregt, Whigham, Baldwin (eds). Backhuys Publishers). A better way of characterizing your two groups of sites would be "brackish intertidal" and "non-tidal freshwater" (unless, of course, your freshwater sites were also intertidal).

2) p. 1, lines 15-16: Your data clearly show that rates of CH4 production and porewater sulfate were each higher in the brackish sites. But, how did you determine that the high sulfate was the cause of the lower CH4 production? You also reported differences in porewater nitrate and acetate (top of p. 8) between ecosystem types, and presumably there were differences in salinity as well. How did you conclude that sulfate was the driving factor? Sulfate wasn't even the most important factor from your modeling exercise (p. 8, lines 11-12).

3) p. 1, line 19, "...increased organic matter generally enhanced CH4 production rates." This statement is too strong for your data set. You used four different organic matter amendments (= 4 plant species). The CH4 production rates increased for only two of the species you tested (Fig. 5); the other two species had no effect. So, based on your data, I don't see how you can justify saying that the amendments "generally enhanced CH4 production." If something happening 50% of the time means that it is a "general" occurrence, you could also say that the amendments generally had no effect on CH4 production.

4) p. 2, line 4, "...21 times more effectively..." This value of the global warming potential for CH4 over a 100-year time period is quite old and has been updated in each of the two IPCC reports that have been published after the Whalen paper you cited. The bigger point of this comment is that the global warming potential may not be the best way to compare CO2 and CH4 when one is talking about ecosystem processes, where

gases are emitted or sequestered year after year. I discussed this in a paper that was published last year (Neubauer and Megonigal. 2015. Moving beyond global warming potentials to quantify the climatic role of ecosystems. Ecosystems. 18:1000-1013).

5) p. 2, lines 5-7, "Currently, wetlands at northern latitudes..." Other studies that have taken a longer temporal perspective have concluded that many northern wetlands have had a net cooling effect for the last 8,000-11,000 years (Frolking and Roulet 2007. Global Change Biology. 13:1079-1088). Because CH4 is broken down in the atmosphere – and therefore the warming due to CH4 emitted in any given year is transient – but the cooling due to C sequestration lasts "forever," a wetland that is old enough can have a lifetime net cooling effect, even if its radiative balance over a shorter period implies net warming. So, a single wetland could have a warming or cooling effect, depending on what time scale you consider.

6) p. 3, line 28: Why is the word "wetlands" in quotes? Are you suggesting that your sites aren't actually wetlands? Also, why are you comparing brackish intertidal marshes with (unvegetated?) freshwater ponds? My understanding is that you are trying to compare sites that differ in salinity and sulfate due to their effects on CH4 production. Why not compare vegetated brackish marsh with vegetated freshwater marsh? Or, brackish ponds with freshwater ponds?

7) p. 4, lines 10-11 and 17: I'm a bit confused by your sampling design. You collected a single sample from five sites along the salinity gradient. Elsewhere (e.g., p. 3, lines 16-17), you explained that you expect that the availability of sulfate will be an important driver of rates of CH4 production. Given that, why would you combine all the sites along the salinity gradient into a single "brackish intertidal" value?

8) You have quite a wide range of sulfate values (Table 1); was there a significant relationship between porewater sulfate and CH4 production? This would be another way of getting at your hypothesis about the effects of sulfate on methanogenesis.

9) p. 4, line 20: Does this sulfate concentration indicate the sulfate concentration

in the water that was added when making the slurry or does it indicate the sulfate concentration in the slurry itself? Can you also report the salinity of the water added for the slurries (or the final salinity of the slurry, whatever is consistent with the sulfate concentration)?

10) p. 4, "Increased organic matter simulation" section: How much organic matter did you add to each bottle? Did you characterize the organic matter (e.g., C, N, P contents? lignin content?)? Did you use aboveground or belowground tissues? Were the tissues first cut to a standard size (e.g., passed through a grinder) before added to the bottles? Details such as those should be added to this section.

11) p. 4, line 30: The genus is Menyanthes, not Menanythes. The same genus name is misspelled in some of the figure legends.

12) p. 5, lines 5 and 9, and elsewhere in the manuscript. Generally, you include a space between a number and its units (e.g., "60 mL" on line 5) but other times you don't (e.g., "250mL" on line 5). I also remember seeing some places where you said that your experiment lasted for "14d" (instead of "14 d"). When editing, check throughout the manuscript to see that you include a space between a number and its units.

13) p. 5, line 12: It is a flame ionization detector, not a flame ionizing detector.

14) In the context of comparing between treatments within your study, it is fine to report your CH4 production rates as $\mu$mol per bottle per time. However, this really limits your ability to compare your results with those of others. Note, for example, that you would have gotten different rates (on a per bottle basis) if you had used a different volume of sediment, even if everything else was identical. At a minimum, you should tell the reader the weight of sediment in each bottle (e.g., "The 60 ml of added sediment was equivalent to 80-90 g of dry sediment."). It would be even better if you reported your rates as $\mu$mol CH4 per gram sediment per time.

15) p. 6, lines 1-2, "Porewater concentrations were scaled. . ." I don't understand what

this means. Are you saying that you multiplied the anion concentrations by the volume of porewater in order to determine the total amount of each anion in the bottle? Based on your nutrient/anion results (bottom of p. 7 to top of p. 8), I think this is what you did. But, why did you do this? As with the CH4 production rates (previous comment), reporting things on a per-bottle basis makes your numbers completely dependent on the amount of sediment (and its water content) that you ran through the centrifuge. It seems more straightforward to report your nutrients using molar units (e.g., mM) because those numbers are independent of the volume of sediment that was processed and can be easily compared with other studies.

16) p. 7, line 4, "total sulfate and nitrate" Is this fourth factor the sum of sulfate and nitrate? If so, why did you add them together? I recognize that both sulfate and nitrate are electron acceptors. Adding them together makes the implicit assumption that one mole of nitrate is "the same" as one mole of sulfate. However, the thermodynamics (i.e., energy yield) and stoichiometry (i.e., moles of sulfate or nitrate reduced per mol of carbon oxidized) differ for sulfate reduction and nitrate reduction. So, in terms of competing with methanogens for electron donors, one mole of nitrate is not equivalent to one mole of sulfate.

17) p. 7, line 7: What is delta i?

18) GLMs: I am approaching this manuscript as someone who is interested in the questions you are addressing but is getting lost when trying to understand and interpret the GLMs. Admittedly, this is because I have not ever received formal (or informal) training in GLMs. I am not expecting the revised manuscript or your "Response to reviewers" document to provide a tutorial in how GLMs work, but I hope that you will be able to make some modifications to the manuscript text that will make it easier for someone who isn't familiar with GLMs (such as myself) to follow the analyses that you did. Here are some of the things that are causing me grief:

a. In a GLM, what is a parameter estimate (e.g., as mentioned on p. 7, line 7)? I

am familiar with multiple linear regression analyses, where each explanatory variable has its own parameter (or, "slope"). But, multiple linear regressions use continuous explanatory variables. In contrast, you have some nominal variables (e.g., ecosystem type, time period, macrophyte species). I can't begin to translate from my experience with multiple linear regressions to guess how you would come up with a parameter estimate for a nominal variable, or what such a parameter would even mean.

b. How do you estimate the importance of different variables? And, what does "relative importance" mean? I first thought that relative importance would be where everything is expressed as a fraction of the importance of the most important variable (so, the relative importance of the most important variable would be 1 and everything else would have a lower relative importance). But, that must not be the case since none of the variables for the GLM from the organic matter addition experiment have a relative importance of 1 (p. 9, lines 10-11).

c. Given Figure 6, it is apparent that you can use GLMs to generate predictive equations. Would it be worthwhile to include your best predictive equations in the manuscript for the reader to see?

d. I have no idea how to interpret the "Akaike weights" numbers in Tables 2 and 3.

e. What is a null model?

19) p. 7, Statistical Analyses: What are you using as your level of statistical significance – 0.01? 0.05? 0.10?

20) p. 7-8, Water column and porewater chemistry results: A student of mine did an experiment where he measured porewater concentrations of 130 $\mu$M sulfate, 5 $\mu$M nitrate, and 4 $\mu$M acetate. How do those numbers compare with yours? I don't actually want you to make that comparison but I want to make the point (again) that it is impossible for the reader to make that kind of comparison. I only know that you processed "~50 ml" of sediment but I have no idea of the water content of that sediment. Therefore, I cannot convert your values of the stock of sulfate or nitrate or acetate in a ~50 ml chunk of soil to a concentration.

21) p. 8, line 9 and Tables 2 & 3, "total sulfate/nitrate" Earlier, I thought that you calculated sulfate + nitrate. Did you actually use the ratio of these anions in your analyses? What is the rationale for doing that?

22) p. 9, lines 18-20 and p. 13, lines 4-5: I am unclear how you concluded that it is important to consider the interactions of multiple global change mechanisms. I totally agree with that idea, but do not see how your results demonstrate the importance of studying interactions. After all, you only looked at individual factors, not at interactions. If you had done an experiment where you manipulated salinity and organic matter availability individually and together and had found that the interacting factors gave results that were unexpected based on single factor experiments, then you would have support for the idea that it is important to consider the interactions of multiple global change mechanisms.

23) p. 9, line 22-26, "Many studies. . ." Weston et al. (2014; Biogeochemistry. 120:163-189) is an example of a study that measured CH4 fluxes along a salinity gradient *and* measured rates of methanogenesis. Neubauer et al. (2013; Biogeosciences. 10:8171-8183) reported CH4 fluxes and CH4 production for a wetland that experienced >3 years of experimental saltwater intrusion.

24) p. 9, line 27, "directly linked lower CH4 production to higher sulfate and nitrate concentrations" This is odd phrasing since you saw higher CH4 production where you had higher nitrate; your sentence suggests the opposite pattern. I think the confusion here is related to the way you did the GLMs with sulfate+nitrate (or perhaps sulfate/nitrate) as a model factor. You chemically analyzed these anions separately, but then combined them in some way for the statistical analyses. As noted in earlier comments, I do not understand how/why you combined these anions for statistical analysis.

25) In your Statistical Analysis section (p. 7), you say that you used "general linear

models." In the legends for Tables 2 and 3, you say that you used "general linearized models." A quick Google search for the exact phrase "general linearized models" did not reveal any statistics-related results in the first two pages of search results (I didn't look any deeper than that). A search for the same term (without the quotes) suggested Wikipedia pages for "general linear model" and "generalized linear model" as the top two search results. These are not the same thing. So, this comment is a long-winded request that you clarify whether you used "general linear models," "generalized linear models," or "general linearized models" (whatever they are) and to make sure that you use the correct terminology throughout your manuscript.

26) p. 10, lines 11-15: You cannot directly compare your CH4 production rates with those reported by Hines et al. (2008). Most importantly, Hines used 50 ml of slurry with 1 part soil to 3 parts total slurry volume; in each of your bottles, you used 120 ml of slurry that was 1 part wet sediment to 1 part water. All else being equal, we would expect higher CH4 production in your study simply because you had more sediment in your bottles. Expressing the rates per gram of soil/sediment, as I suggested earlier, would go a long way toward making your results comparable with those from other studies. [Helpfully, Hines et al. reported the typical weight of dry soil per milliliter of their slurry so you can get a rough idea of what their rates would be if expressed per gram of soil. Your data repository file lists sediment weights (wet or dry?) for each bottle.]

27) p. 10, last paragraph: Ultimately, this paragraph is unsatisfying. After seeing the huge June vs. August difference in rates of CH4 production, I was really hoping that you would be able to provide some strong insight into the cause(s) of that difference. I guess you are limited by data availability. Still, I wonder if others working in similar systems have reported order-of-magnitude changes in rates over such a short time period. I don't know anything about your system except what is in the manuscript but I'm wondering if the pattern could be related to the timing of soil thaw in the early growing season or perhaps the phenology of plant growth. Finally, I'll note that the measured

concentration of acetate reflects the balance between rates of acetate production and acetate consumption. So, if higher acetate production in August was balanced by higher acetoclastic methanogenesis, you would see high rates of CH4 production without correspondingly high acetate concentrations.

28) p. 5, line 20 and p. 11, line 8: Change detectible to detectable.

29) p. 12, lines 5-8: The possible role of antimicrobial compounds is an interesting hypothesis and you presented some information to support it. However, I did not see where you tested for the effects of litter quality (e.g., C:N:P, percent lignin, lipid content) on CH4 production rates. Without having run those analyses (either the chemical analyses or the statistical analyses), why are you discounting the possible influence of those factors that have previously been shown to be important?

30) p. 12, line 13, "Fewer studies have examined. . ." There have been studies looking at CH4 emissions when wetland plants are grown in an elevated CO2 environment. Although there are important differences between CH4 emissions and CH4 production, the elevated CO2 studies generally find that CH4 emissions increase with elevated CO2, with this increase often being attributed to higher plant production (see, for example, Vann and Megonigal 2003. Biogeochemistry. 63:117-134).

31) Table 1: The legend says that you made 4-10 spot measurements per site. Given that, why aren't there any standard deviations or other estimates of error/variability for pond depth, temperature, pH, and salinity?

32) Data repository: I took a look at the data that you made available at the knd.informatics.org site and had a question about your CH4 production rates on the "All CH4 data" Excel worksheet. In column R, you reported "areal CH4 production (umol/m2/d)." How did you determine areal rates? What does an areal rate even mean in the context of a soil slurry in a bottle? Is the "area" the same as the cross-sectional area of the bottle? If so, that seems meaningless to me since a cylindrical bottle is going to have the same cross-sectional area whether the bottle is $\frac{1}{4}$ full, $\frac{1}{2}$ full, or $\frac{3}{4}$ full but,

presumably, would have different rates of CH4 production due to the different amounts of soil in the bottle. Given that you didn't report the areal rates in your manuscript, this whole things would be a question of curiosity...except that you used these areal rates to calculate the per-bottle rates (column S of the Excel file) that *are* reported in your manuscript. So, I need to know more about these areal rates before I can judge the validity of the per-bottle rates.

33) Data repository: I do not understand the formula that you used to go from areal rates (umol/m2/d) to per-bottle rates (umol/d): per-bottle rate = areal rate * 0.2 * sediment volume in liters. In order for the units to work out, the 0.2 factor must have units of m2/L. Those are odd units. Where does 0.2 m2/L come from and what does that conversion factor represent?

END OF REVIEW

---

## Author Comment (AC2) · 6 Oct 2016

Dear Dr. Scranton,

We very much appreciate the time and effort you put into reviewing this manuscript. We have synthesized your comments and questions along with the second reviewer's into a single response in the form of an author's comment on the discussion site. In that joint response, we have also attached the revised manuscript as a supplement, where the changes inspired by your comments are marked with blue text and those inspired by the other reviewer are highlighted in yellow. Again, we appreciate the feedback and recognize that your efforts have resulted in a greatly improved manuscript.

Respectfully,

[Figure]

Carmella Vizza, Will West, Stuart Jones, Julia Hart, and Gary Lamberti

---

## Author Comment (AC3) · 6 Oct 2016

Dear Dr. Neubauer,

We very much appreciate the time and effort you put into reviewing this manuscript. We have synthesized your comments and questions along with the first reviewer's into a single response in the form of an author's comment on the discussion site. In that joint response, we have also attached the revised manuscript as a supplement, where the changes inspired by your comments are highlighted in yellow and those inspired by the other reviewer are marked in blue. Again, we appreciate the feedback and recognize that your efforts have resulted in a greatly improved manuscript.

Respectfully,

[Figure]

Carmella Vizza, Will West, Stuart Jones, Julia Hart, and Gary Lamberti

---

## Author Comment (AC4) · 6 Oct 2016

Dear Drs. Scranton and Neubauer,

We very much appreciate the time and effort you put into reviewing this manuscript. We have synthesized your comments and questions into a single response in the form of an author's comment on the discussion site, but we kept different reviews separate. We have also attached the revised manuscript as a supplement, where the changes inspired by Dr. Scranton's comments are marked with blue text and those inspired by Dr. Neubauer's are highlighted in yellow. Again, we appreciate the feedback and recognize that your efforts have resulted in a greatly improved manuscript.

Respectfully,

Carmella Vizza, Will West, Stuart Jones, Julia Hart, and Gary Lamberti

**RESPONSE TO REVIEWER 1, DR. MARY SCRANTON:**

**Response to 2nd set of comments (SC2):**

Dear Dr. Scranton,

Thank you again for taking the time to review our response to your comments. We appreciate the friendly tone of the conversation, and we acknowledge that many of the misunderstandings have resulted from differences between marine and freshwater fields. Having your perspective as a marine geochemist has helped us to hone our language and hopefully the manuscript will now be more easily understood across fields. Although we know that you disagree with our specific sea-level rise simulation, we appreciate your comment that the study "does contribute to an understanding of how increasing sulfate concentrations might change methane production." In the hope of keeping the second response succinct, we will briefly address some of the major points in your second and first set of comments in that sequence. We hope that our actions will speak louder than our words in the supplement that we have attached, which is a revised version of the manuscript with blue text for the changes that have been made in light of your comments and suggestions.

First of all, we agree that there may be differences in terminology between scientists studying marine and freshwater ecosystems, so we have tried to make our terminology clearer throughout the manuscript by describing our wetlands as fully inundated (e.g., **please see pg. 4, lines 6-12**), and we also separated water column parameters from sediment parameters (**please see Tables 1 and 2**) and added data on porewater acetate, nitrate, and sulfate concentrations. Regarding the pooling of samples, if we had wanted to assess the "electron tower" paradigm in freshwater wetlands, we agree that it would be not be appropriate to make composite sediments for microbial samples or even make homogenous 20-cm sediment slurries for methane production without doing more explicit surveys of how porewater chemistry and microbial communities vary by depth of the sediment. However, we believe that assessing the validity of this paradigm is beyond the scope of this paper. While we call on it to inform our results, the main purpose of the first portion of the paper was to assess how average methane production varies across freshwater and brackish wetlands and how microbial communities and normative chemical conditions may influence this process. Our goal was thus to characterize methane production over a large spatial extent, which inevitably involves some homogenizing and compositing of sediments for study feasibility.

Additionally, we explicitly present how much carbon we added in the organic matter simulation (**please see pg. 5, lines 14-17**) and appreciate you detecting this oversight. We also formalized our analyses of litter quality, discussed litter quality more in depth, and tried to be explicit about what Tiegs et al. 2013 previously measured in the Methods, Results, and Discussion (**please see pg. 5, lines 17-18, pg. 8, lines 28-29, p. 10, lines 20-21, and pg. 15, lines 2-9**). We also added more details in the Methods about our chemistry and statistical analyses (**see Sections 2.3.2 and 2.3.3 and pg. 8, lines 1-8**). Regarding your concern with the sea-level rise simulation, we now refer to it throughout the manuscript as a "biogeochemical sea-level rise simulation" (e.g., **please see pg. 1, line 14, pg. 3, lines 19-23, pg. 13, lines 20-25**). Lastly, we agree with you that hydrogenotrophic methanogenesis is potentially important and that we cannot rule it out in our ecosystems (**please see pg. 12, lines 14-18, and pg. 12, last paragraph**).

Again, we appreciate the time and effort you have taken to make this manuscript a better one. Below you will find your set of comments point-by-point to which we have already responded and posted online, but we have added in the exact changes that we have made in the **revised manuscript in bold** for your reference.

Respectfully,

Carmella Vizza, Will West, Stuart Jones, Julia Hart, and Gary Lamberti

**Response to 1st set of comments (SC1):**

Dear Dr. Scranton,

Thank you for taking the time to read and comment on our manuscript. We appreciate your observations and questions. Below is our response to your comments.

**Comment:** "The basic premise on which the authors base their study is that methane production depends on substrates produced during fermentation of organic matter and that methane production and sulfate reduction usually do not occur in the same sediment, presumably because of competition for these substrates." **Response:** In this comment, you seem to imply that methane production and sulfate reduction do not occur in the same sediment. Methanogens and sulfate-reducing bacteria do directly compete for organic substrates such that when sulfate or other more energetically profitable alternative electron acceptors are available, some methane is still produced, but it is at lower rates (please see manuscript citations: Achtnich et al. 1995; Lovley and Klug 1983, 1986).

**Comment:** "The authors mimic sea level rise by adding brackish water to their samples (which ignores the possible importance of increased water logging and decreased oxygen in the sediments)." **Response:** These sediments came from freshwater wetland ponds (0.4 – 1.2 m total water depth) that were already waterlogged and therefore had decreased oxygen in the sediments. We have no reason to suspect that the physical act of sea-level rise would change the saturation status of the sediments. **Please see pg. 4, lines 2-8.**

**Comment:** "All incubations are done after vials are flushed with nitrogen although the natural sediments apparently all had some level of oxygen present in situ." **Response:** In this comment and throughout your review, you seem to imply that purging vials with nitrogen gas is problematic. It is widespread practice in methane production bottle assays to purge with nitrogen gas in order to remove the oxygen

that was introduced by removing sediment from below the water's surface and exposing it to the air in the field and in the lab (Lofton et al. 2014; Sinke et al. 1992; West et al. 2012, 2015).  Of course, it is possible that there are microsites where oxygen is present in the freshwater wetland sediments, but we believe that the same may be true in the bottle experiments. **Please see pg. 5, lines 23-27.**

**Comment:** "Values for nitrate (another potential substrate for carbon remineralizing organisms) and for acetate (the putative important substrate for methanogens) were measured but data were apparently quite variable and are not reported." **Response:** Summary porewater acetate levels are reported in Results section 3.1.1. You are correct in that we did not report actual values for nitrate, and we will add a short line in the results to indicate that nitrate was usually 2+ orders of magnitude lower than total sulfate levels. We measured acetate, nitrate, and sulfate fairly extensively in both the water column and the porewater as well as other parameters.  We did not present all the data we collected because we did not want to overwhelm the reader, but these data are all available in the archived data.  If reviewers and editors feel that these data would enhance the manuscript, we will add them in tabular or figure form to the manuscript or as supplemental information. **Please see Table 2.**

**Comment:** "No measurements were made of $H_2$, another potential substrate for methanogens and methanogens were not examined to see whether they were actually acetoclastic or hydrogenoclastic." **Response:** We did not measure $H_2$ production as we do not currently have an instrument available to measure this nor did we classify the type of methanogens.  However, acetate was the predominant factor explaining methane production patterns in almost all experiments, which suggests that a large proportion of the methanogenesis is acetoclastic (e.g., pg. 8 lines 29-30).  However, we do not deny the potential of hydrogenotrophic methanogenesis occurring in our ecosystems (please see last paragraph on page 10). **Please see revised manuscript, pg. 12, lines 14-18, and pg. 12, last paragraph.**

**Comment:** "The data for the dsrA and mcrA genes are again not presented and in the results section appear to contract the conclusions made about their abundance. The authors also ignore the fact that numbers of genes do not directly relate either to number of cells (a cell can have more than one copy of a gene) or to gene activity." **Response:** The *dsrA* and *mcrA* summary data are presented in the Results section 3.1.3, and individual gene data per sample are also archived in the data set (see microbial tab). Although we acknowledge that the number of genes does not equate with number of cells or gene activity, qPCR of functional genes for particular guilds is a commonly used approach to estimate the abundance of a functional group.  We also acknowledge in the manuscript that it is possible that some of the genes we detect are from dormant microbial communities (please see pg. 11, lines 8-12).  The number of genes corresponding to abundance is always going to be a bit of an assumption, but in a broader survey of our wetlands, we have unpublished data from our system that *mcrA* abundance relative to *dsrA* abundance is correlated with methane production rates.  Also, in *Letters in Applied Microbiology* (Volume 62, Issue 2, Pages 111-118), Morris et al. 2016 found that hydrogenotrophic methane production rates corresponded to *mcrA* abundance.  Although observation of functional gene transcripts or even the actual enzymes catalyzing the reactions would clearly be preferred, logistics associated with our remote field sites precluded work with these more labile macromolecules.  Despite challenges associated with interpreting DNA copy number of functional genes, we believe that our use of qPCR substantially adds to the literature as there are few studies that simultaneously measure methane production and microbial functional group abundances.  For example, the studies listed on pg. 9, lines 22-26, have hypothesized that microbial community processes are behind methane patterns

along a salinity gradient, but none of these studies actually tested this hypothesis by measuring microbial communities. **Please see revised manuscript, pg. 7, lines 7-9 and pg. 11, lines 12-25.**

**Comment:** "Finally the authors use natural organic matter to enrich their incubations but do not indicate the amount of carbon added or the relative lability of that carbon. (Clearly a gram of sucrose and a gram of twigs would not be expected to stimulate microbial activity to the same extent)." **Response:** We added 3 g of live tissue per macrophyte species, which translated to 0.10-0.12 mol of C added to each incubation, and we will add these values to the methods of the manuscript. Dr. Scott Tiegs of Oakland University has measured the % C, % N, and % P of the four litter species we added: Marestail (45% C, 1.7% N, 0.17% P), Buckbean (44% C, 0.94% N, 0.15% P), Lily (45% C, 1.7% N, 0.17% P), and Horsetail (47% C, 2.5% N, 0.24% P), but the patterns we observed did not follow %C, %N, %P, C:N or C:P, which are considered standard indices of litter quality. We discuss the phosphorus in detail because Tiegs et al. 2013 found that litter decomposed more rapidly when P was higher (see pg. 11, lines 14-24). You are correct in that we did not specifically measure the lability of the carbon from each macrophyte species, but we are leveraging information from Tiegs et al. 2013, whose study had already assessed this by measuring decomposition rates of these litter types in Copper River Delta ponds. We will clarify this in the discussion. **Please see revised manuscript pg. 5, lines 14-18, pg. 8, lines 28-29, pg. 10, lines 21-22, and pg. 15, lines 2-9.**

**Comment:** "P2 Line 6: I do not like the term green house compensation point. It is not widely used and does not directly relate to what you are measuring (as you say nothing about carbon sequestration in this system)." **Response:** We believe that the term "greenhouse compensation point" accurately describes wetlands' dual roles in carbon sequestration and emissions, and we believe that because northern wetlands are on the edge of this compensation point that it is so important to study how global change may alter the methane cycle. Greenhouse compensation point is a good way to set up the broader context of this study. **Please see pg. 2, lines 6-8.**

**Comment:** "You do not mention carbon fertilization again, and there is no indication in the text of whether any increases in carbon production in the CRD system would be due to warming or to carbon fertilization. I think you hurt yourself by trying to draw connections to too many issues. I would recommend drastic simplification of this section and sticking to the facts." **Response:** Increased $CO_2$ levels lead to warming as well as $CO_2$ fertilization, both of which could affect the amount of substrate available. We are not precisely sure whether increased organic matter will result from warming, from longer growing seasons directly, or from $CO_2$ fertilization; therefore, it seems prudent to mention both potential mechanisms. Regardless, we will reevaluate our discussion of these points based on your recommendation and consider ways to focus our comments. **Please see pg. 2, lines 9-12 and pg. 3, lines 4-5.**

**Comment:** "P2 Line 16: redox conditions are only indirectly related to climate change. You seem to imply the mere presence of nitrate or sulfate changes redox conditions but this is not true, especially if oxygen is present. In fact there can be a lot of nitrate and sulfate in oxic surface sediments. Did you ever have sulfide in your samples? Does oxygen penetration vary with ecosystem? How far below the surface were the samples collected?" **Response:** True, redox conditions are a potential indirect effect of climate change. Sea-level rise could increase sulfate levels just as higher decomposition rates due to warming could deplete oxygen levels, both of which affect redox conditions. The sediment came from the top 20 cm, and although some oxygen is likely present in small amounts, it is also likely depleted quickly. For

example, the layer of water directly above the sediments often has low DO levels of around 1 mg/L, particularly in the evening, and it is not uncommon to have anoxic groundwater upwelling in these systems. We also observe that water column DO levels drop throughout the season as vegetation begins to senesce. As for hydrogen sulfide, we did not directly measure that, but sediment characteristics (black coloration and pungent odor) suggest the presence of sulfide. Usually sediments become anaerobic within the first few cm of freshwater ponds, which is another reason that we make the incubations anaerobic by purging with nitrogen gas. Of course oxygen matters, but the amount of nitrate in particular is orders of magnitude lower than agricultural and other human impacted systems. Lastly, if significant levels of oxygen were present in these sediments, it would kill the methanogens because they are extremely sensitive to $O_2$ (please see manuscript citation: Whalen 2005) and we would therefore see very little methane production in our experiments. **Please see pg. 2, line 18.**

**Comment:** "P2 Line 23: I don't think an "abundant supply of organic matter can reduce competition for methanogens by increasing substrate availability". Try instead "abundant supply of organic matter can increase substrate availability"" **Response:** We appreciate the wording suggestion and will make the change. **Please see pg. 2, lines 28-29.**

**Comment:** "P2 Line 28: replace "are likely results of" with "may be influenced by" " **Response:** We appreciate the wording suggestion and will make the change. **Please see pg. 3, line 2.**

**Comment:** "P3 lines 7 and 10. Use same unit for sealevel rise (100 and 170 cm)" **Response:** We appreciate the wording suggestion and will make the change. **Please see pg. 3, lines 14-16.**

**Comment:** "P3 line 15: Numbers of methanogens not as important as whether or not the methanogens are active." **Response:** Please see our response to your major comment above about qPCR and the feasibility of RNA work. **Please see revised manuscript, pg. 7, lines 7-9.**

**Comment;** "P4 line4: The range of physicochemical parameters in table 1 are actually pretty small for most measurements. Perhaps more important is whether the intertidal sediments are exposed to the atmosphere at low tide (tidal range?). How long are they submerged? What is the water content? Again it matters how far below the surface these sediments were collected. From the table it must be shallow since there was more O2 in these sediments than the freshwater wetlands." **Response:** The intertidal sediments were collected from the top 20 cm just as in the freshwater wetlands, and they are covered with freshwater during low tide and increasingly brackish water during high tide. So again these sediments are waterlogged with depleted oxygen levels. In Table 1, the DO data are actually from the water column, as we do not have oxygen data for the sediment. In the freshwater ponds where limnological profiles could be conducted, we reported DO levels from the bottom of the water column, but in intertidal marsh, DO levels came from the surface layer. This is detailed in the legend for Table 1. We will remove the water column DO data altogether since they seem more misleading than helpful. **Please see Table 1.**

**Comment:** "P4 line 26: Were no replicates run for sediments from a single site? How can you tell if observed variability is just typical of replicate samples? I would also think you MUST indicate how much macrophyte tissue you added (probably in terms of gC/g sediment or something like that) to even know if these treatments were similar since the lability of the carbon is likely not the same." **Response**: We ran five control replicates without added substrate at each freshwater wetland (n = 5), which we then averaged to form the basis of the delta CH4 production metric. Each macrophyte treatment was

replicated in 5 different ponds. In the increased organic matter simulation, we used pond as the replicate because we were more interested in capturing how wetlands differing in biogeochemistry along a glacial to oceanic gradient would respond to organic matter addition rather than how much variability there exists within a single wetland's response. **Please see pg. 5, lines 7-12.**

**Comment:** "P5 Line 6: You mean incubation temperature not ambient temperature?" **Response"** Yes, we do. We appreciate this good suggestion for a wording change. **Please see pg. 5, line 24.**

**Comment:** "P5 line 8: Purging with nitrogen will likely have a bigger effect than incubating at a few degrees cooler than the actual sediment. I would expect stimulation by this as you allow more anaerobic processes to occur. Were any blanks or controls run?" **Response:** Please see our response to your major comment above. **Please see pg. 5, lines 23-27.**

**Comment:** "P5 line 12: "flame ionization" not "flame ionizing" detector" **Response:** We will make this change. **Please see pg. 5, lines 31.**

**Comment:** "P5 line 20: You report that only sulfate was detectable in the water column. Did you present any water column data? Do they mean anything? What are detection limits for pore waters? Present nitrate and acetate data for sediments as well as sulfate." **Response:** We do have extensive water column data for these wetlands, some of which are in the archived data. If reviewers feel these data add to the manuscript, we can add them in tabular form or in supplemental information. Some (i.e., DOC and sulfate) are also already presented in Table 1, but acetate and nitrate were not detectible in the water column (we will add concentration detection limits to the manuscript which were about 10 and 5 µM, respectively). **Please see Table 1 for water column data and Table 2 for sediment porewater data.**

**Comment:** "P5 line 27: the method you describe is typically called "loss on ignition" and represents a loss of organic matter. Did you convert to organic C? Is this data reported anywhere in the paper? Was DOC measured the same way? Blanks? Detection limit?" **Response:** Yes, we took the sediment organic matter data (in % of OM) from loss on ignition and converted it to organic C (pg. 5, lines 25-29). We will add in the following citation about conversion to organic C: Thomas et al. 2005; Aquat. Sci. 67:424–433. The data were not reported in the paper as to not overwhelm the reader, especially since the amount of carbon did not affect the results, but the data are archived and available online. DOC was measured using a Shimadzu TOC-V (pg. 5, lines 18-19), and we did run blanks. All samples had detectible DOC, but our lowest standard was 1 mg/L and all samples registered above this level with the exception of five of the intertidal samples, which fell in between the blank and the lowest standard. If reviewers would like more detail on these analyses in the methods, we will add them to the manuscript. **Please see sections 2.3.2 and 2.3.3.**

**Comment:** "P6 line 2: I would use the word "converted" rather than "scaled" " **Response:** Thank you for the wording suggestion. **Please see pg. 6, lines 18 and 23.**

**Comment:** "P6 line 5: I couldn't tell what this composite sediment was used for. Were incubations for each site or was all sediment made into a composite? Why do this? It seems that then the averages for the genes refer to different samples than the physiochemical properties. Can you properly do statistics comparing the combined samples in one parameter to the individual samples in one parameter?" **Response:** The composite sediment was a combination of the five sediment samples from different sites

for each freshwater wetland, but this was only done for the microbial analyses.  Making a composite is highly practiced in soil microbial ecology for the purpose of controlling analytical costs while still capturing significant spatial heterogeneity.  We made a composite for each freshwater wetland which were then compared against ten sediment samples from intertidal marsh.  So yes, it is true that we could not directly link methane production of a single sample or its physicochemical properties to the functional group abundances, but overall we were able to characterize methanogen and sulfate-reducer abundances across ecosystems and relate these to average methane production rates. **Please see pg. 6, lines 30-31 and pg. 7, line 1.**

**Comment:** "P7 line 2: move "log transformed" to immediately follow "four factors"" **Response"** Methane production rates were log-transformed, not the factors. We cannot make this change because it would misrepresent our statistics. **Please see pg. 7, line 27-29 where we rearranged the wording for clarity.**

**Comment:** "P7 line 26ff: You use a lot of significant figures for something that is so variable. Maybe you are justified in two significant figures but not 3 or more. Again blanks and detection limits need to be mentioned as your averages are pretty close to zero (or at least almost include zero). I would like to see these data associated with specific samples" **Response:** We will reduce the number of significant figures and make the needed detection statements in the methods.  We only reported parameters if they were detectible.  The reason that these averages are close to zero in this particular line is that we converted actual concentrations to the total amount in the incubation bottle (i.e., µM * PW volume in incubation * (PW volume per ml of sediment) * sediment volume in bottle).  **Please see sections 2.3.2, 2.3.3, 3.1.1., 3.2 and Tables 1 and 2.**

**Comment:** "P8 line 14: This paragraph seemed very odd. You seem to contradict yourself a lot. For intertidal samples, three had no dsrA but this was found in all freshwater AND the dsrA was independent of ecosystem type. This seems to contradict your hypothesis which you nevertheless cling to. Remember the presence of a gene doesn't mean the organisms are doing anything at the moment and the number of genes does not necessarily indicate the number of cells of a particular organisms." **Response:** Indeed, we did not detect *dsrA* in 3 of the 10 intertidal samples.  Nevertheless, *dsrA* abundance tended to be higher in intertidal ecosystems, although not significantly so.  In contrast, the *mcrA* presence and abundance did vary significantly by ecosystem, and methanogens are the group that we tend to use to explain our methane production results directly, while we use sulfate and sulfate-reducer abundance as supporting evidence (please see pg. 9, lines 14-16). **Please see revised manuscript pg. 9, line 22, and pg. 11, lines 18-25.**

 **Comment:** "P8 line 27ff: I really don't like your equating sealevel rise purely with sulfate concentration. It is ok to say that sea level rise will flood current intertidal areas, but the vegetation will change and the water will be permanently water logged rather than periodically exposed to air. You have no data on estuarine wetlands to compare to the freshwater wetlands (although others have done this comparison in other systems). I would guess the reason you see no effect here is that you are mimicking the process the wrong way. You are really looking at increased sulfate levels, not sealevel rise." **Response:** In these systems, which are waterlogged throughout the year, we expect sulfate concentration to be a major biogeochemical change.  Our simulation not only adds sulfate, but it also adds other nutrients and the microorganisms brought in with the saltwater. You are correct that over the long term vegetation and other characteristics of these ecosystems will also change. We will add comments at pg. 3., line 12, and pg. 4 line 20 to reflect our biogeochemical focus on the effects of sea-level rise. Our data on intertidal

ecosystems for comparison with freshwater ecosystems can be found in Results section 3.1.2 and Figure 3. **Please see revised manuscript, pg. 1, line 14, pg. 4, lines 6-8, pg. 13, lines 20-25.**

**Comment:** "P10,line2: The sentence beginning "our study also demonstrates..."seems to directly contradict your own data as you said before that dsrA did not correlate with ecosystem type" **Response:** Just because *dsrA* presence and abundance did not significantly correlate with ecosystem type does not mean the presence or abundance of sulfate-reducing bacteria does not affect methane production in intertidal ecosystems. Even though these bacteria are also present in freshwater ecosystems, they could be limited by sulfate availability. Also *dsrA* tended to be higher on average relative to freshwater ecosystems, just not significantly so. We will change the wording to "tended to have generally higher sulfate-reducer abundances when present" to address this concern. **Please see revised manuscript pg. 9, line 22, and pg. 11, lines 18-25.**

**Comment:** "P10 line 13ff and next paragraph: These two sections are wild speculation with absolutely no data behind them. There are a lot of other factors that might be important that you haven't included." **Response:** We provide potential hypotheses and explanations for our data that were grounded in the literature, which is not an uncommon practice in the discussion. Specifically, we discuss the acetoclastic pathway of methane production and how we might expect that to change in the future as well as possible reasons for why methane production rates varied by an order of magnitude throughout the season. We could omit these sections, but felt that it was important to say something about the results we observed. Regardless, we will reevaluate the nature and extent of our explanations. **Please see pg. 12, line 13, where we have extensively re-worked this paragraph. Also, we acknowledge that the last paragraph on pg. 12 involves hypothesizing about seasonal differences in $CH_4$ production because the data we collected fall short of explaining them, but the discussion is backed up by literature and we put forward a more formal hypothesis stating that future study is needed.**

**Comment:** "P11line14: This whole section is hurt by the fact that you don't characterize the macrophytes at all in terms of potential lability. Did you add a constant amount of C or organic N? Or just dry mass? Or just a "chunk" of leaves? Again much of this section is speculation without more facts" **Response:** Please see our response above to your major comment about lability of macrophytes. We evaluated many different possibilities for explaining the results of the different macrophyte treatments including % C, % N and % P, but the quality measurement that seemed to matter the most was antimicrobial properties. None of these potential indicators of lability appeared to have an effect on methane production, which we will clarify in the Discussion pg 11, lines 14-30, and pg 12, lines 1-9. **Please see revised manuscript pg. 5, lines 14-18, pg. 8, lines 28-29, pg. 10, lines 21-22, and pg. 15, lines 2-9.**

**Comment:** "P12 line19: You have no idea whether there are hydrogen utilizing methanogens around or how much hydrogen there might be. You can reiterate what other people have said in other papers but I don't see you connecting these to your system with any facts." **Response:** We will try to improve the clarity of the manuscript, by acknowledging that methanogens in general utilize acetate, directly (acetoclastic) or indirectly (hydrogenotrophic). We will also add that $CO_2$ and $H_2$ are compounds that also come from fermentation of organic matter such as acetate. Hence, this is why acetate is important and was measured in this study. **Please see pg. 12, lines 14-18.**

**Comment:** "Modeling: I didn't really understand the models. If you are making linear equations including various parameters and then constant factors for each, I would like to see more detail on how this worked. If you have a lot of variables, you can fit most data but figure 6 is completely mysterious and doesn't convince me of the value of your model. I also didn't understand the columns in tables 2 and 3. I have never heard of an AIC before, for example. Explain the statistics a bit to an audience that may include non-biologists." **Response:** Akaike Information Criterion (AIC) is an increasingly utilized form of model selection that generates estimates of the model being the best representation given the data and the set of models being explored.  It also penalizes for the number of parameters in the model.  We will add more description of this in the methods. Basically, general linearized models are created, each of which is associated with an AIC value, which is then used to rank the models. We also corrected for small sample sizes ($AIC_c$). The model with the lowest $AIC_c$ value is considered the best, and all remaining models are compared relative to the best approximating model using delta $AIC_c$ ($\Delta_i$).  Models with a $\Delta_i$ less than or equal to 2 are considered to have substantial support, while models having a $\Delta_i$ greater than 7 have little support (Burnham and Anderson 2002).  The relative strength of the models was then evaluated with Akaike weights ($\omega_i$), which indicate the probability of a model being the best model, given the data and the set of candidate models (Burnham and Anderson 2002). **Please see revised manuscript pg. 8, lines 1-13.**

Respectfully,

Carmella Vizza, Will West, Stuart Jones, Julia Hart, and Gary Lamberti

**RESPONSE TO REVIEWER 2, DR. SCOTT NEUBAUER**

**Response to 1st set of reviewer comments (R1):**

Dr. Neubauer,

We greatly appreciate the extensive time and effort you put into reviewing our manuscript.  Your comments were insightful and extremely thorough, and we believe that your suggestions have greatly improved the manuscript.  Below please find our point-by-point responses to your suggestions and comments with the location of changes we made to the revised manuscript in bold.  The attached supplement is the revised manuscript, which also has the changes your comments inspired highlighted in yellow.

**Comment:** "I was very interested in the topics covered by this manuscript and think that the measurements and experiments can provide some insight into questions about controls of CH4 production and how global changes will affect CH4 dynamics. However, I was a bit disappointed in the analysis and interpretation of the data, especially in how the authors tied their findings to existing knowledge. Similarly, I felt that the authors could have done a better job exploring the literature on how plants and organic matter inputs affect CH4 production." **Response:** We are pleased that you found our study interesting, and we thank you for pointing us towards several studies we had not found in our original literature search. We hope you find our discussion greatly expanded in light of the literature you recommended as well as a few other papers we came across in the process. **Please see pg., 11 lines 25-30, and pgs. 12-14.** Also, we hope that the expanded information on the AIC model selection as well as the fleshing out of some of our hypotheses in the discussion will aid the reader in understanding our data analysis and interpretation.

**Specific comment 1:** p. 1, line 13; p. 3, line 12; and throughout manuscript, "freshwater and intertidal wetlands". As someone who studies tidal freshwater wetlands, I take issue with the way you are classifying wetlands as either freshwater *or* intertidal. Wetlands can be both! Indeed, there are over 19,000 ha of tidal *and* freshwater wetlands within the Copper River Delta (see Hall's chapter in the 2009 "Tidal Freshwater Wetlands" book; Barendregt, Whigham, Baldwin (eds). Backhuys Publishers). A better way of characterizing your two groups of sites would be "brackish intertidal" and "non-tidal freshwater" (unless, of course, your freshwater sites were also intertidal). **Response:** Certainly, wetlands are difficult to classify and the terminology is something that we should clarify because it appears to be a source of confusion. In light of Dr. Scranton's comment, we have added that the 'freshwater wetlands' are constantly inundated with fresh water so as to distinguish them from our 'intertidal wetlands', which are covered in freshwater at low tide and covered in increasingly brackish water at higher tides (rarely exposed to air at low tide). Although our freshwater wetlands currently do not receive any tidal influence, their surrounding sloughs sometimes do, which is why they are at risk of seawater intrusion. We have concluded that the best terms to call our wetlands in light of yours and Dr. Scranton's comments are "tidal brackish wetlands" and "non-tidal freshwater wetlands." **Please see pg. 1, line 13 in the Abstract and the terms were changed throughout the rest of the manuscript (we only highlighted the first instance in the revised manuscript so as not to distract from other changes) as well as pg. 4, lines 9-12.**

**Specific comment 2:** p. 1, lines 15-16: Your data clearly show that rates of CH4 production and porewater sulfate were each higher in the brackish sites. But, how did you determine that the high sulfate was the cause of the lower CH4 production? You also reported differences in porewater nitrate and acetate (top of p. 8) between ecosystem types, and presumably there were differences in salinity as well. How did you conclude that sulfate was the driving factor? Sulfate wasn't even the most important factor from your modeling exercise (p. 8, lines 11-12). **Response:** Table 3 shows that the top model (lowest $AIC_c$ score) included all factors (ecosystem type, acetate, sulfate, and time period) and the second best model included ecosystem type, acetate and sulfate. Therefore, for explaining methane production rates in the non-tidal freshwater wetlands and tidal brackish wetlands, the most important factors are ecosystem type, acetate, and sulfate. First, nitrate availability was negligible in comparison to the total sulfate availability so we decided to remove that from the analysis (also based on your comment #16). Second, we concluded that acetate was not responsible for lower methane production rates in the brackish sites because acetate was actually higher than in the freshwater sites, and higher acetate availability should theoretically increase methane production. We believe that the variable 'ecosystem type' captures factors other than sulfate such as microbial communities and perhaps even salinity. It is difficult to disentangle the effects of sulfate from salinity since those two variables are highly correlated **(please see our expanded discussion of this on pg. 11, lines 18-30 and pg. 12, lines 1-12)**. However, mechanistically from a redox perspective, we believe that it is most plausible that lower methane production rates in brackish sites resulted from sulfate-reducing bacteria (which tended to be higher when present in brackish versus freshwater sites) that outcompete methanogens (which were significantly lower in brackish versus freshwater sites). One could also argue that stress from salinity might also lower methanogenic activity, but if this were the case, this should be a more immediate effect that would have decreased methane production in the sea-level rise simulation. Conversely, the turnover of microbial communities in response to redox conditions may be a less immediate effect. You are correct in pointing out that our hypothesis, although supported by our data, does require some

assumptions and deduction on our part. We have changed the wording to "probably due to higher sulfate availability" on **pg. 1, lines 15-16.**

**Specific comment 3:** p. 1, line 19, ". . .increased organic matter generally enhanced CH4 production rates." This statement is too strong for your data set. You used four different organic matter amendments (= 4 plant species). The CH4 production rates increased for only two of the species you tested (Fig. 5); the other two species had no effect. So, based on your data, I don't see how you can justify saying that the amendments "generally enhanced CH4 production." If something happening 50% of the time means that it is a "general" occurrence, you could also say that the amendments generally had no effect on CH4 production. **Response:** In calculating the difference in methane production between the paired treatment and control, this metric was positive in 15 out of 20 cases and so we believe that 75% of the cases is enough to say "generally." However, we agree with your comment in light of the particular species and will change the wording to say "enhanced $CH_4$ production in 75% of the incubations, but this response depended on the macrophyte species added with half of the species treatments having no significant effect," **please see pg. 1, lines 20-21.**

**Specific comment 4:** p. 2, line 4, ". . .21 times more effectively. . ." This value of the global warming potential for CH4 over a 100-year time period is quite old and has been updated in each of the two IPCC reports that have been published after the Whalen paper you cited. The bigger point of this comment is that the global warming potential may not be the best way to compare CO2 and CH4 when one is talking about ecosystem processes, where C3 gases are emitted or sequestered year after year. I discussed this in a paper that was published last year (Neubauer and Megonigal. 2015. Moving beyond global warming potentials to quantify the climatic role of ecosystems. Ecosystems. 18:1000-1013). **Response:** We appreciate your comment and acknowledge that using GWP may not be the best or most updated way of addressing why methane is an important greenhouse gas. We changed the wording to better reflect this, **please see pg. 2, lines 4-5**.

**Specific comment 5:** p. 2, lines 5-7, "Currently, wetlands at northern latitudes. . ." Other studies that have taken a longer temporal perspective have concluded that many northern wetlands have had a net cooling effect for the last 8,000-11,000 years (Frolking and Roulet 2007. Global Change Biology. 13:1079-1088). Because CH4 is broken down in the atmosphere – and therefore the warming due to CH4 emitted in any given year is transient – but the cooling due to C sequestration lasts "forever," a wetland that is old enough can have a lifetime net cooling effect, even if its radiative balance over a shorter period implies net warming. So, a single wetland could have a warming or cooling effect, depending on what time scale you consider. **Response:** Thank you for your insightful comment. We agree with you that a wetland's role in carbon sequestration and emissions is highly dependent on temporal perspective. **Please see pg. 2, lines 6-12** where we attempt to better reflect this point.

**Specific comment 6:** p. 3, line 28: Why is the word "wetlands" in quotes? Are you suggesting that your sites aren't actually wetlands? Also, why are you comparing brackish intertidal marshes with (unvegetated?) freshwater ponds? My understanding is that you are trying to compare sites that differ in salinity and sulfate due to their effects on CH4 production. Why not compare vegetated brackish marsh with vegetated freshwater marsh? Or, brackish ponds with freshwater ponds? **Response:** We were attempting to name a term that we would use to refer to them throughout the manuscript and therefore should have put both CRD and wetlands in quotes. For clarification, we are comparing vegetated freshwater non-tidal wetlands or "ponds" or with vegetated brackish tidal wetlands. We

consider the freshwater wetlands to be "pond-like" because they have more clearly delineated boundaries, whereas the brackish tidal wetlands are continuous.  Unfortunately, there are no brackish water ponds that we know of in the Copper River Delta to which to compare our freshwater systems. We also wanted our sites to be comparable in depth.  The freshwater sites with depths greater than 0.4 m all tend to be "pond-like" due to the clay-like sediments in the area that allow very little drainage, whereas the brackish sites tend to have a depth of up to 2 m during high tide and 1 m during low tide. In general, the sites we sampled in the brackish tidal wetlands tended to be a bit shallower than the average depth for logistical reasons (i.e., using a handheld bucket auger from a jet boat). We have attempted to make this clearer in this paragraph, **please see pg. 4, lines 5-12**.

**Specific comment 7:** p. 4, lines 10-11 and 17: I'm a bit confused by your sampling design. You collected a single sample from five sites along the salinity gradient. Elsewhere (e.g., p. 3, lines 16-17), you explained that you expect that the availability of sulfate will be an important driver of rates of CH4 production. Given that, why would you combine all the sites along the salinity gradient into a single "brackish intertidal" value? **Response**: We combined all the tidal brackish sites into one value for the tables in the manuscript because this ecosystem was rather continuous and lacked clearly distinct boundaries like the non-tidal freshwater wetlands, **please see pg. 4, lines 15-18**. They are also combined in the figures as to contrast two distinctive ecosystem types.  However, when it comes to data analysis through AIC model selection, each tidal brackish site is considered on its own such that widely varying sulfate levels are appropriately taken into account, **please see pg. 7, lines 27-29.**

**Specific comment 8:** You have quite a wide range of sulfate values (Table 1); was there a significant relationship between porewater sulfate and CH4 production? This would be another way of getting at your hypothesis about the effects of sulfate on methanogenesis. **Response**: Yes, if we log-transform methane production rates from the brackish and freshwater wetland comparison and run correlation tests on both porewater sulfate and total sulfate (porewater + water column), the results are r = –0.67, P = 0.0001, and r = –0.71, P = 0.00003, respectively. Although we believe that doing separate correlations by factor are redundant with AIC model selection (although different philosophically), we are happy to include these correlations in the manuscript if reviewers and editors feel they would improve its clarity.

**Specific comment 9:** p. 4, line 20: Does this sulfate concentration indicate the sulfate concentration in the water that was added when making the slurry or does it indicate the sulfate concentration in the slurry itself? Can you also report the salinity of the water added for the slurries (or the final salinity of the slurry, whatever is consistent with the sulfate concentration)? **Response:** We have added another table to distinguish between water and sediment chemistry data, with the water column concentrations presented in Table 1 and the porewater concentrations presented in Table 2.  The salinity concentrations of the water column are reported in Table 1, but unfortunately our salinity probe is only designed for measurements of the water column in the field so we were not able to also measure the salinity of the porewater (i.e., we could only extract a few mL of porewater per sediment sample). **Please see Table 2**, where we have presented both the porewater sulfate data as well as the total amount of sulfate available in the slurry (PW + WC).

**Specific comment 10:** p. 4, "Increased organic matter simulation" section: How much organic matter did you add to each bottle? Did you characterize the organic matter (e.g., C, N, P contents? lignin content?)? Did you use aboveground or belowground tissues? Were the tissues first cut to a standard size (e.g., passed through a grinder) before added to the bottles? Details such as those should be added to this

section. **Response:** This is a good suggestion. **Please see pg. 5, lines 14-18** where we have updated this information in the manuscript.

**Specific comment 11:** p. 4, line 30: The genus is Menyanthes, not Menanythes. The same genus name is misspelled in some of the figure legends. **Response:** We greatly appreciate you catching this error. It has been changed throughout the manuscript. **Please see pg. 5, line 13, and Figure Captions 5 and 6**.

**Specific comment 12:** p. 5, lines 5 and 9, and elsewhere in the manuscript. Generally, you include a space between a number and its units (e.g., "60 mL" on line 5) but other times you don't (e.g., "250mL" on line 5). I also remember seeing some places where you said that your experiment lasted for "14d" (instead of "14 d"). When editing, check throughout the manuscript to see that you include a space between a number and its units. **Response:** We agree with you on consistency and will check all instances. In general, we did not include a space when a unit was used an adjective, as in describing a 250-mL serum bottle, which according to the style of *Biogeosciences* should be listed as a "250mL serum bottle." Nouns such as "250 mL of sediment" should indeed contain a space as you suggest. In light of the confusion, we have gone back and hyphenated all the adjectives and highlighted these instances for reference. We hope that the editor will clarify whether he prefers the lack of space or the hyphenated adjectives in light of these instances.

**Specific comment 13:** p. 5, line 12: It is a flame ionization detector, not a flame ionizing detector. **Response:** Thank you. We have made this change, **please see pg. 5, line 31.**

**Specific comment 14:** In the context of comparing between treatments within your study, it is fine to report your CH4 production rates as μmol per bottle per time. However, this really limits your ability to compare your results with those of others. Note, for example, that you would have gotten different rates (on a per bottle basis) if you had used a different volume of sediment, even if everything else was identical. At a minimum, you should tell the reader the weight of sediment in each bottle (e.g., "The 60 ml of added sediment was equivalent to 80-90 g of dry sediment."). It would be even better if you reported your rates as μmol $CH_4$ per gram sediment per time. **Response: Please see pg. 5, line 22,** where we have added in the average amount of wet sediment. We agree with you that we would have gotten different methane production rates if we had added widely variable amounts of sediments, but the coefficient of variation in the wet sediment masses was approximately 3% and therefore did not affect the major trends when we reran all the analyses with $CH_4$ production rates as nmol $CH_4$ per gram of dry sediment per day. Originally, we reported values on a per bottle basis because we were treating each bottle as its own "wetland microcosm." This is also the reason we ran analyses with the total amount of each anion per incubation converted from porewater concentrations. Nonetheless, we agree with your argument that it is important to be able to make this comparable to other studies. Therefore, we have added the line in the methods clarifying how much wet sediment was added (dry sediment mass is available in the archived data) and have completely re-done all figures, tables, and analyses with $CH_4$ production rates being reported and analyzed as nmol $CH_4$ per gram of dry sediment per day. **Please see pg. 6, line 3, Tables 3-4 and Figures 3-6.**

**Specific comment 15:** p. 6, lines 1-2, "Porewater concentrations were scaled. . ." I don't understand what this means. Are you saying that you multiplied the anion concentrations by the volume of porewater in order to determine the total amount of each anion in the bottle? Based on your nutrient/anion results (bottom of p. 7 to top of p. 8), I think this is what you did. But, why did you do this? As with the CH4 production rates (previous comment), reporting things on a per-bottle basis makes

your numbers completely dependent on the amount of sediment (and its water content) that you ran through the centrifuge. It seems more straightforward to report your nutrients using molar units (e.g., mM) because those numbers are independent of the volume of sediment that was processed and can be easily compared with other studies. **Response:** We have clarified how and why we converted porewater concentrations to the total amount per g of dry sediment (we removed bottle rates based on your comment 14), **please see pg. 6, lines 21-25**. Essentially, we converted the porewater concentrations to the total amount of each anion per bottle because the volume of porewater extracted from the sediments varied widely among samples (~47% CV). In light of your suggestion of being able to make comparisons between studies, we have now reported both the amount of each anion in the porewater per g of dry sediment as well as the traditional porewater concentrations **in Table 2**.

**Specific comment 16:** p. 7, line 4, "total sulfate and nitrate" Is this fourth factor the sum of sulfate and nitrate? If so, why did you add them together? I recognize that both sulfate and nitrate are electron acceptors. Adding them together makes the implicit assumption that one mole of nitrate is "the same" as one mole of sulfate. However, the thermodynamics (i.e., energy yield) and stoichiometry (i.e., moles of sulfate or nitrate reduced per mol of carbon oxidized) differ for sulfate reduction and nitrate reduction. So, in terms of competing with methanogens for electron donors, one mole of nitrate is not equivalent to one mole of sulfate. **Response:** Thank you for this insightful comment. Originally, we combined sulfate and nitrate as an overall representation of the presence of alternative electron acceptors and in order to minimize the number of parameters in model selection. Although we agree with you that the thermodynamics and stoichiometry varies per reaction, we thought that trying to account for this would be negligible in light of the fact that nitrate was such a small component of this factor (~ 4.5%) in comparison to total sulfate (~ 95.5 %). In light of how negligible nitrate is and because of the stoichiometric concerns you voice here, we have decided to remove nitrate from analyses and instead just focus on total sulfate availability and therefore redid all analyses to account for this. **Please see pg. 7, line 31 and pg. 8, line 1.**

**Specific comment 17:** p. 7, line 7: What is delta i? **Response: Please see pg. 8, lines 1-13** where we have updated the manuscript to reflect more details about AIC model selection.

**Specific comment 18:** GLMs: I am approaching this manuscript as someone who is interested in the questions you are addressing but is getting lost when trying to understand and interpret the GLMs. Admittedly, this is because I have not ever received formal (or informal) training in GLMs. I am not expecting the revised manuscript or your "Response to reviewers" document to provide a tutorial in how GLMs work, but I hope that you will be able to make some modifications to the manuscript text that will make it easier for someone who isn't familiar with GLMs (such as myself) to follow the analyses that you did. **Response: Please see pg. 8, lines 1-13,** where we have updated the manuscript to reflect more details about AIC model selection. **Specific comment 18a**: In a GLM, what is a parameter estimate (e.g., as mentioned on p. 7, line 7)? I am familiar with multiple linear regression analyses, where each explanatory variable has its own parameter (or, "slope"). But, multiple linear regressions use continuous explanatory variables. In contrast, you have some nominal variables (e.g., ecosystem type, time period, macrophyte species). I can't begin to translate from my experience with multiple linear regressions to guess how you would come up with a parameter estimate for a nominal variable, or what such a parameter would even mean. **Response:** For categorical variables, a GLM essentially adjusts the intercept. For example, in considering ecosystem type, both freshwater and brackish wetlands would have their own intercept. **Specific comment 18b:** How do you estimate the importance of different

variables? And, what does "relative importance" mean? I first thought that relative importance would be where everything is expressed as a fraction of the importance of the most important variable (so, the relative importance of the most important variable would be 1 and everything else would have a lower relative importance). But, that must not be the case since none of the variables for the GLM from the organic matter addition experiment have a relative importance of 1 (p. 9, lines 10-11). **Response:** Great question, and we have included more details about this in the manuscript. **Please see pg. 8, lines 11-13.** Also, we have included the updated Akaike weights after model averaging in **Tables 3 and 4** so that the reader can mentally calculate the relative importance in their head. For example, in Table 3, there were three models included in model averaging with a $\Delta_i > 4$, and sulfate is a factor included in 2 of the 3 models whose $\omega_{i\,(MA)}$ were 0.61 and 0.26, so its relative importance is 0.87. **Specific comment 18c**: Given Figure 6, it is apparent that you can use GLMs to generate predictive equations. Would it be worthwhile to include your best predictive equations in the manuscript for the reader to see? **Response:** We will be glad to report this should the editor feel this would be helpful. Although since these models are fairly specific to this study and these ecosystems, we would not necessarily recommend that others try to extrapolate methane production rates from them. **Specific comment 18d**: I have no idea how to interpret the "Akaike weights" numbers in Tables 2 and 3. **Response: Please see pg. 8, lines 3-8. Specific comment 18e:** What is a null model? **Response:** A null model includes an intercept only and we have updated this in the methods for clarification, **please see pg. 8, line 9, and Tables 3 and 4.**

**Specific comment 19:** p. 7, Statistical Analyses: What are you using as your level of statistical significance – 0.01? 0.05? 0.10? **Response:** Great question, and we have updated our methods to reflect this. **Please pg. 8, lines 16-17.**

**Specific comment 20:** p. 7-8, Water column and porewater chemistry results: A student of mine did an experiment where he measured porewater concentrations of 130 µM sulfate, 5 µM nitrate, and 4 µM acetate. How do those numbers compare with yours? I don't actually want you to make that comparison but I want to make the point (again) that it is impossible for the reader to make that kind of comparison. I only know that you processed "~50 ml" of sediment but I have no idea of the water content of that sediment. Therefore, I cannot convert your values of the stock of sulfate or nitrate or acetate in a ~50 ml chunk of soil to a concentration. **Response:** Again, you are absolutely right that it is important for others to be able to compare their ecosystems to ours in terms of chemistry, and therefore porewater concentrations have been added to **Table 2**.

**Specific comment 21:** p. 8, line 9 and Tables 2 & 3, "total sulfate/nitrate" Earlier, I thought that you calculated sulfate + nitrate. Did you actually use the ratio of these anions in your analyses? What is the rationale for doing that? **Response:** No, we did not use a ratio, but rather we originally added sulfate and nitrate together. However, we have removed nitrate from analyses based on your comment 16.

**Specific comment 22:** p. 9, lines 18-20 and p. 13, lines 4-5: I am unclear how you concluded that it is important to consider the interactions of multiple global change mechanisms. I totally agree with that idea, but do not see how your results demonstrate the importance of studying interactions. After all, you only looked at individual factors, not at interactions. If you had done an experiment where you manipulated salinity and organic matter availability individually and together and had found that the interacting factors gave results that were unexpected based on single factor experiments, then you would have support for the idea that it is important to consider the interactions of multiple global change mechanisms. **Response:** Good point, as our study did not directly assess the interactions of

simulated sea-level rise and increased organic matter availability. However, in assessing the differences in methane production between freshwater and brackish wetlands as well as in our sea-level rise simulation, we included both redox conditions (i.e., total sulfate availability) and a component of organic matter (i.e., acetate availability) as factors in our models. Both redox conditions and acetate availability were important factors for both modeling efforts, but interestingly the influence of each factor for the freshwater/brackish comparison was in the opposite direction for the organic matter simulation. This suggests that environmental context matters, i.e., higher sulfate levels generally lead to lower methane production, but methanogens in areas with more alternative electron acceptors might be more likely to respond to increased organic matter availability. This is why we believe that other studies should not only look into these factors in different ecosystems, but also examine how they interact. **Please see pg. 11, 3-6.**

**Specific comment 23:** p. 9, line 22-26, "Many studies. . ." Weston et al. (2014; Biogeochemistry. 120:163- 189) is an example of a study that measured CH4 fluxes along a salinity gradient *and* measured rates of methanogenesis. Neubauer et al. (2013; Biogeosciences. 10:8171- 8183) reported CH4 fluxes and CH4 production for a wetland that experienced >3 years of experimental saltwater intrusion. **Response:** We thank you for drawing our attention to these sources. **Please see pg. 11, lines 12-16.**

**Specific comment 24:** p. 9, line 27, "directly linked lower CH4 production to higher sulfate and nitrate concentrations" This is odd phrasing since you saw higher CH4 production where you had higher nitrate; your sentence suggests the opposite pattern. I think the confusion here is related to the way you did the GLMs with sulfate+nitrate (or perhaps sulfate/nitrate) as a model factor. You chemically analyzed these anions separately, but then combined them in some way for the statistical analyses. As noted in earlier comments, I do not understand how/why you combined these anions for statistical analysis. **Response:** Our results did show lower methane production when sulfate availability was high (**please see Table 3**), and we removed nitrate based on your comment 15. Methane production and sulfate availability were inversely related (please see our response to your comment 8). We attempted to clarify the wording, **please see pg. 11, lines 14-16.**

**Specific comment 25:** In your Statistical Analysis section (p. 7), you say that you used "general linear models." In the legends for Tables 2 and 3, you say that you used "general linearized models." A quick Google search for the exact phrase "general linearized models" did not reveal any statistics-related results in the first two pages of search results (I didn't look any deeper than that). A search for the same term (without the quotes) suggested Wikipedia pages for "general linear model" and "generalized linear model" as the top two search results. These are not the same thing. So, this comment is a long-winded request that you clarify whether you used "general linear models," "generalized linear models," or "general linearized models" (whatever they are) and to make sure that you use the correct terminology throughout your manuscript. **Response:** We appreciate you catching this, it is most common to just refer to them as GLMs, but the analyses used were actually named "generalized linear models" and **pg. 7, line 28,** has been changed accordingly as well as in **Tables 3 and 4**.

**Specific comment 26:** p. 10, lines 11-15: You cannot directly compare your CH4 production rates with those reported by Hines et al. (2008). Most importantly, Hines used 50 ml of slurry with 1 part soil to 3 parts total slurry volume; in each of your bottles, you used 120 ml of slurry that was 1 part wet sediment to 1 part water. All else being equal, we would expect higher CH4 production in your study simply

because you had more sediment in your bottles. Expressing the rates per gram of soil/sediment, as I suggested earlier, would go a long way toward making your results comparable with those from other studies. [Helpfully, Hines et al. reported the typical weight of dry soil per milliliter of their slurry so you can get a rough idea of what their rates would be if expressed per gram of soil. Your data repository file lists sediment weights (wet or dry?) for each bottle.] **Response:** Originally, we had wanted to present Hines' data in bottle rates so that they could be comparable to the rates we originally report in the manuscript, but we agree with your point that rates should be compared based on g of sediment. **Please see pg. 12, lines 18-25**, where we have updated these calculations and extensively re-worked this paragraph.

**Specific comment 27:** p. 10, last paragraph: Ultimately, this paragraph is unsatisfying. After seeing the huge June vs. August difference in rates of CH4 production, I was really hoping that you would be able to provide some strong insight into the cause(s) of that difference. I guess you are limited by data availability. Still, I wonder if others working in similar systems have reported order-of-magnitude changes in rates over such a short time period. I don't know anything about your system except what is in the manuscript but I'm wondering if the pattern could be related to the timing of soil thaw in the early growing season or perhaps the phenology of plant growth. Finally, I'll note that the measured concentration of acetate reflects the balance between rates of acetate production and acetate consumption. So, if higher acetate production in August was balanced by higher acetoclastic methanogenesis, you would see high rates of CH4 production without correspondingly high acetate concentrations.  **Response:** Yes, you are correct that we are limited by data availability, but wish we had been able to explain these intriguing seasonal differences.  However, in the hope of making this paragraph more satisfying, we have fleshed out our hypothesis about why these seasonal differences might occur and it indeed pertains to plant phenology!  Also, you are correct that porewater acetate availability is a balance between acetate production and consumption.  **Please see pg. 13, lines 8-18.**

**Specific comment 28:** p. 5, line 20 and p. 11, line 8: Change detectible to detectable. **Response:** Done, **please see pg. 6, line 10 and pg. 14, line 17.**

**Specific comment 29:** p. 12, lines 5-8: The possible role of antimicrobial compounds is an interesting hypothesis and you presented some information to support it. However, I did not see where you tested for the effects of litter quality (e.g., C:N:P, percent lignin, lipid content) on CH4 production rates. Without having run those analyses (either the chemical analyses or the statistical analyses), why are you discounting the possible influence of those factors that have previously been shown to be important? **Response:** We have formalized our analyses of the Tiegs et al. (2013) stoichiometry for the macrophytes used in this study and can therefore rule out C:N:P. **Please see revised manuscript pg. 5, lines 14-18, pg. 8, lines 28-29, pg. 10, lines 21-22, and pg. 15, lines 2-9.** Although we cannot rule out percent lignin or lipid content, it is our hypothesis that antimicrobial compounds were responsible for why methanogens responded differently to macrophyte species based on the literature we found about their antimicrobial properties. Even if we had measured percent lignin or lipid content, it would be difficult to conclusively rule out that some other quality measure contributed to the differences we observed without extensively characterized the quality of these macrophyte species. Again, we acknowledge that antimicrobial compounds hypothesis is untested, but this study does provide some anecdotal evidence suggesting that it might be worthy of further examination. **Please see pg. 15, line 24.**

**Specific comment 30:** p. 12, line 13, "Fewer studies have examined. . ." There have been studies looking at CH4 emissions when wetland plants are grown in an elevated CO2 environment. Although there are important differences between CH4 emissions and CH4 production, the elevated CO2 studies generally find that CH4 emissions increase with elevated CO2, with this increase often being attributed to higher plant production (see, for example, Vann and Megonigal 2003. Biogeochemistry. 63:117-134). **Response:** We appreciate you pointing us toward some interesting studies. **Please find our discussion updated on pg. 15, lines 29-31, and pg. 16, lines 1-6.**

**Specific comment 31:** Table 1: The legend says that you made 4-10 spot measurements per site. Given that, why aren't there any standard deviations or other estimates of error/variability for pond depth, temperature, pH, and salinity? **Response:** In light of the fact that these variables did not vary all that much, we originally did not report the error in order to avoid the table from becoming too cluttered. However, we are happy to report them, and please find standard deviations for all these parameters **in Table 1.**

**Specific comments 32 and 33:** Data repository: I took a look at the data that you made available at the knd.informatics.org site and had a question about your CH4 production rates on the "All CH4 data" Excel worksheet. In column R, you reported "areal CH4 production (umol/m2/d)." How did you determine areal rates? What does an areal rate even mean in the context of a soil slurry in a bottle? Is the "area" the same as the cross-sectional area of the bottle? If so, that seems meaningless to me since a cylindrical bottle is going to have the same cross-sectional area whether the bottle is 1 4 full, 1 2 full, or 3 4 full but, C10 presumably, would have different rates of CH4 production due to the different amounts of soil in the bottle. Given that you didn't report the areal rates in your manuscript, this whole things would be a question of curiosity. . .except that you used these areal rates to calculate the per-bottle rates (column S of the Excel file) that *are* reported in your manuscript. So, I need to know more about these areal rates before I can judge the validity of the per-bottle rates. I do not understand the formula that you used to go from areal rates (umol/m2/d) to per-bottle rates (umol/d): per-bottle rate = areal rate * 0.2 * sediment volume in liters. In order for the units to work out, the 0.2 factor must have units of m2/L. Those are odd units. Where does 0.2 m2/L come from and what does that conversion factor represent? **Response:** We apologize for any confusion caused by not removing the areal rates from the data repository, and we have updated those data included in the repository accordingly, **please see https://knb.ecoinformatics.org/#view/doi:10.5063/F1028PF8**. We have plans to use areal $CH_4$ production rates as a parameter for another manuscript, one that involves a process-based model with a methane budget for each pond. In order to use the methane production data as an input we needed to come up with a conversion factor of sediment volume to area, which is what the 0.2 $m^2$ per L represents. That conversion factor has nothing to do with this study though; it is just that we simply back-calculated the bottle rates in this particular spreadsheet from the areal rates. Just to be clear about our process: To get bottle rates, the GC concentrations are in ppm that are regressed against incubation duration in days (since we have measurements from multiple time points), which we then convert to $\mu$mol $L^{-1}$ $d^{-1}$ which we then multiply by headspace volume. To determine areal rates, we take the GC concentrations in ppm that are regressed against incubation duration in days, which we then convert to $\mu$mol $L^{-1}$ $d^{-1}$ which we then multiply by headspace volume and divide by sediment volume, and lastly we multiply by the areal conversion factor (1 L/0.2 $m^2$). This is why in the spreadsheet online we took the areal rates, divided by that areal conversion factor (or multiplied the values by 0.2 $m^2$/L), and then multiplied that value by sediment volume in mL and divided that by 1000 to convert to L.

Again, this was only a back calculation since we did not have the raw µmol L$^{-1}$ d$^{-1}$ data in this particular spreadsheet.

Respectfully,

Carmella Vizza, Will West, Stuart Jones, Julia Hart, and Gary Lamberti

---

## Author Comment (AC5) · 6 Oct 2016

[revised manuscript text omitted]

---

## Author Response (AR2)

Dear Dr. Ardón and Anonymous Reviewer #4,

We very much appreciate the time and effort you put into reviewing this manuscript as well as the other two reviews and our extensive responses to them. We will respond to your comments in a point-by-point fashion below, and we have attached the revised manuscript as a supplement with the changes suggested by Dr. Ardón marked in red text and those suggested by Reviewer 4 highlighted in yellow. Again, we appreciate the feedback and recognize that your efforts have resulted in an improved manuscript.

Respectfully,

Carmella Vizza, Will West, Stuart Jones, Julia Hart, and Gary Lamberti

**RESPONSE TO REVIEWER 3, DR. MARCELO ARDÓN:**

I think this is an interesting study. The authors did a good job responding to the two previous reviewers. I have some minor comments regarding parts that could be clarified or worded differently.

**Comment 1:** Page 1 Line 20 of the abstract- I think the comma should be moved to after the word added. **Response:** We appreciate you catching this error. We have corrected it on **pg. 1, line 20.**

**Comment 2:** Page 2 line 10-12- this sentence is too long. Break it up into 2 sentences, and check your comma placements. I don't think you need a comma after the word fertilization. **Response:** We agree that we could make this sentence easier to read by breaking it up into two, more clearly constructed sentences. Please see **pg. 2, lines 6-9.**

**Comment 3:** Page 2- line 25- I would change the word "signal" to something like "lead to" **Response:** Thank you for this suggestion. We have made this change on **pg. 2, line 19.**

**Comment 4:** Page 3- lines 2-3- I would remove the phrase "which may be influenced by rising global CO2 concentrations and temperatures". You already spent two paragraph explaining these mechanisms. **Response:** We have removed this phrase on **pg. 2, lines 26-27.**

**Comment 5:** Page 4- 16- the brackish tidal wetlands seem to vary more than the freshwater wetlands. It seems like your brackish wetlands span a large salinity gradient, even overlapping the freshwater wetlands. Why combine them? **Response:** Originally, we chose not to combine freshwater systems in presenting Table 1 because we had more extensively characterized them, and as wetland ponds, they have more clearly delineated boundaries. In contrast, brackish tidal wetland sites were located within one large continuous area. However, based on your comments and those of reviewer 4, we have decided to add in all 10 brackish sites in both **Table 1 and 2**. Nevertheless, we should make it clear that brackish tidal wetlands sites along the salinity gradient were not combined for analyses. Please see **pg. 4, lines 11-12 and 19-20**, where we have attempted to make this clearer.

**Comment 6:** Page 4 line 28 – "We then we"- please correct **Response:** We appreciate you catching this typo. We have

corrected it on **pg. 4, line 28**.

**Comment 7:** Page 4 lines 28-30- I am confused by the number of samples in your comparisons. You sampled 9 separate locations in the non-tidal freshwater water wetlands and 5 locations in your brackish wetlands. But then you did 5 incubations per location. So is that n=45 for the non-tidal freshwater wetlands, and n=25 for the brackish wetlands? Please clarify. **Response:** We have clarified this in **pg. 4, lines 24-30**.

**Comment 8:** Page 5- line 15- can you give us a quick summary of the differences in the quality of these species? Is the Tiegs data from the same time you did your experiment? **Response:** We have added a summary of the quality differences in **pg. 5, lines 19-21**. The Tiegs data was collected a few years earlier than our experiment, but in the same study area.

**Comment 9:** Page 8 line 29- where the %C, %N, %P, C:N, and C:P measured in this experiment, or are they from the Tiegs et al. 2013 paper? **Response:** These data were from the Tiegs et al. 2013 paper. We have clarified this on **pg. 8, line 30**.

**Comment 10:** Page 9 lines 4-5- It is not clear to me how I can assess that ecosystem differences are bigger than temporal differences from Tables 1 and 2. You don't give any measure of temporal variability for any of the wetlands in the tables. The variation in brackish wetlands could be spatial or temporal. And as I pointed out above, it seems strange to me to combine the brackish wetlands when the ranges are bigger than the non-tidal freshwater wetlands, which you say you kept separate due to spatial differences. **Response:** Brackish wetlands were not combined in statistical analyses, and we have added all 10 brackish sites in both **Tables 1 and 2**. Tables 1 and 2 contain both spatial and temporal variation, but you are correct that one cannot deduce from those tables alone whether ecosystem differences are bigger than temporal differences. Instead we specifically break out ecosystem and temporal variation in the statistics that follow this opening sentence. We will attempt to rephrase it to clear up any confusion. Please see **pg. 9, lines 6-7**.

**Comment 11:** Page 11 lines 5 and 6- what do you mean by "the interaction of global change mechanisms"? Acetate and sulfate availability are not "global change mechanisms". Please explain. **Response:** We have attempted to clarify this by specifically listing global change mechanisms that could affect acetate and sulfate, please see **pg. 11, lines 9-14**.

**Comment 12:** Page 11 line 16- again I get distracted by how you use the word "signals". Leads to or indicates? I am not quite sure what you mean in this sentence. **Response:** Thank you for this word choice suggestion, please see **pg. 11, line 24** where we use the indicate instead of signal.

**Comment 13:** Page 11 line 30- add a comma after organisms. **Response:** Thank you for this suggestion, we have added a comma on **pg. 12, line 8**.

**Comment 14:** Page 12 lines 1-2- calcium co-precipitating with P does not necessarily lead to ammonium release. But increased salinity does lead to ammonium release through cation exchange. **Response:** Yes, we agree with you and have changed the sentence to better reflect that calcium co-precipitation with P does not necessarily have a causal relationship with ammonium release. Please see **pg. 12, line 10.**

**Comment 15:** Page 13 lines 14-16- Can you provide a citation for these patterns? I don't think you can infer all these patterns from your 2 time sampling. **Response:** We can only provide a general citation for how plants and their phenology can affect ecosystem process. In addition to adding this general citation, we have changed the wording to make it clear that the seasonal trajectory we are proposing, while it fits with our data, is hypothetical. We added in this hypothesis in the first revision submitted to BGS due to the fact that another reviewer felt our explanation for the seasonal patterns was dissatisfying. Please see **pg. 13, lines 22-25**, where we have attempted to incorporate your comments, while still offering a more satisfying explanation for the other reviewer. We will defer to the editor here as to whether we have struck the right balance between the two sets of reviewer comments about this discussion paragraph.

**Comment 16:** Page 13 line 30 130-320 is not two orders of magnitude larger than 5. It is one order of magnitude. **Response:** Thank you for this suggestion. The data are between 1-2 orders of magnitude larger than 5, but not 2 exactly. We have changed the wording on **pg. 14, line 9**.

**Comment 17:** Page 16 line 25- why CO2 fertilization? I would remove that one. Or change it to say increased organic matter to be consistent with your Fig 1. Increased organic matter can be caused from longer growing season, CO2 fertilization, and increased breakdown of soil organic matter. **Response:** Thank you, we have changed it be more consistent with Fig. 1. Please see **pg. 17, line 5**.

**RESPONSE TO REVIEWER 4:**

As the third reviewer of this manuscript I had an opportunity to read the previous reviews and responses. The authors addressed each of the previous comments in detail and made changes to the manuscript, with the result that the manuscript that I received has clearly improved. My comments are of two varieties, reactions to the authors responses to previous comments and new comments from my own review. I am only commenting on the responses that I had some objection to, so let me state at the beginning that the authors did a good job of addressing the many other comments that I do not mention.

**Comment on Previous Responses 1:** Dr. Scranton objected to the authors description of their seawater addition experiments as a sea level rise manipulation. I agree with Dr. Scranton that this term in inappropriate because it conjures up a large number of interacting processes and feedbacks that were not manipulated or observed in this study. Changing the term to "biogeochemical sea level rise" brings it closer to the reality, but even this term is misleading. I suggest "seawater addition experiment" or "short-term seawater addition experiment" to efficiently communicate what the authors did. The connection between the seawater addition experiment presented here and the far more complicated issue of sea level rise will be immediately obvious to most readers, particularly in the context of the introduction to the paper. This change should be made throughout the full paper. **Response:** Thank you for this suggestion. We have made this change throughout the entire manuscript whenever we refer to what we formerly called the "biogeochemical effects of sea-level rise simulation." We still

refer to sea-level rise as a global change mechanism in both the introduction and discussion, but only as it applies to broader scale patterns.

**Comment on Previous Responses 2:** I object to the term "greenhouse gas compensation", but for somewhat different reasons than Dr. Scranton. My reason is that the current greenhouse gas balance of any ecosystem is irrelevant to the issue of

5   climate change because that balance (regardless of closeness to the compensation point) is part of the greenhouse gas "baseline". It does not matter if the system is close to or far from the compensation point provided that balance is about the same now as it was in the past when climate was more stable. In the context of this paper the authors are trying to make a different point, which is that factors such as temperature and elevated $CO_2$ cause this balance to change from the baseline. Please drop the "compensation" idea and emphasize the idea of change from the baseline. The most direct way of

10   communicating this idea is to use the term "radiative forcing". **Response:** We understand your objection and appreciate the clarification. We have removed mention of compensation point and instead refer to change from the "wetland greenhouse emissions baseline." Please see **pg. 2, lines 4-9**.

**Comment on Previous Responses 3:**  Dr. Scranton commented on the amount of text devoted to elevated carbon dioxide and warming. I am happy that these issues are discussed in the paper, but I agree that it creates an expectation in the

15   introduction that $CO_2$ and temperature were manipulated in the study. This problem can be avoided by generalizing the context of the present study from elevated $CO_2$ and temperature to include any factor that changes carbon uptake, carbon loss as $CO_2$, or carbon loss as $CH_4$. Elevated $CO_2$ can then be introduced as one example of a factor that can change plant growth rates, and temperature as another example. For example, something like this: "Any factor that changes the availability of electron donors (i.e. organic carbon) or electron acceptors (e.g. sulfate supply) has the potential to change the

20   greenhouse gas balance of an ecosystem, thereby exacerbating or mitigating radiative forcing of climate. Factors that are known to change organic carbon availability include elevated $CO_2$ and temperature, which affect both plant physiology and potentially growing season length. Factors that can alter the supply of specific electron acceptors include sea level rise (i.e. sulfate supply) and agricultural pollution (i.e. nitrate supply). We investigated the sensitivity of $CH_4$ production to changes in the supplies of organic carbon (electron donor) and sulfate (electron acceptor) in wetland soils. Our objective was to gain

25   mechanistic insights on a subset of factors that regulate $CH_4$ emissions in wetland systems that will transition from non-tidal and tidal freshwater to brackish with climate change and sea level rise." Again, please discuss your manipulations as "saltwater additions" and limit the mention of sea level rise except to give some occasional context. **Response:** We appreciate your suggestions and have revised our second paragraph in light of this comment. Please see **pg. 2, lines 11-25**.

**Comment on Previous Responses 4:**  Dr. Scranton has a comment in reference to P2, Line 16 of the original submission

30   where he points out that a difference in sulfate concentration does not translate into a difference in redox conditions. I agree and would add that "redox conditions" is mostly used incorrectly in the paper. I suggest that the term "redox" be dropped

almost entirely because there are no redox data in the paper and it is not possible to infer redox state from these data. Instead use the term "electron acceptor availability". **Response:** You are correct in that we did not directly measure redox potential. However, it is not uncommon in the literature to infer some information about redox conditions from the presence of certain chemical species. Nevertheless, we believe that "electron acceptor availability" is a more precise term, and we have made this change throughout the rest of the manuscript.

**Comment on Previous Responses 5:** Please refine your edits in response to Dr. Neubauers comment about the word "generally" in describing the CH4 production response to litter addition. I think his point was that the total absence of such a response in two species is the more important result. Please edit to read something like "Organic matter addition consistently stimulated CH4 production rates for just two of the four species used as amendments. This indicates that the consequences of changes in plant production on CH4 production will be highly species dependent." **Response:** Thank you for clarifying this. We have made a change to better reflect this on **pg. 11, lines 7-9**.

**Comment on Previous Responses 6:** I understand the authors point that non-tidal sites are easily delineated from one another but the tidal sites are not, but I cannot understand why this must translate into the structure of tables 1 and 2. Because the brackish sites are used as independent replicates the reader needs to know how they differed from one another with respect to the characteristics in tables 1 and 2. Please list each brackish site separately. **Response:** There were a total of ten incubations (5 sites along a salinity gradient x 2 time periods) conducted at the brackish tidal wetlands, and sediment/water was sampled from one continuous wetlands complex. Because brackish tidal wetlands were characterized less than the freshwater wetlands, we originally grouped them together in a summary form in Tables 1 and 2. However, we did analyze each brackish site separately, and in light of that and your suggestion, we have added the data for each site in **Tables 1 and 2.**

**Comment on Previous Responses 7:** Specific Response 22. Although the response answers the question the way it was phrased, I believe the point of the comment was that the authors make too big a leap from the result that acetate and sulfate correlations behaved differently in the two wetland types to "multiple global change mechanisms". Please make a more modest statement such as "This indicates that we do not have a sufficient mechanistic understanding of how changes in electron donor and acceptors will interact to ultimately influence methane production." **Response:** Thank you again for the clarification. Please **pg. 11, lines 9-14,** where we have made this change.

**New Comment 1:** P3, L1. I respectfully disagree that we need to understand the effects of sea level rise on CH4 production to forecast the global CH4 budget. Sea level rise is an issue that affects wetlands that are already tidal or will be tidal in the future, and wetland carbon budgets like those of Bridgham et al. (2006 in Wetlands) show that CH4 emissions from such wetlands are a small part of the methane coming from wetlands globally. Perhaps something like "To accurately forecast the effects of sea level rise on coastal wetland greenhouse gas budgets requires a process-level understanding of responses to

potential changes in electron donor and acceptor availability." **Response**: Thank you for this suggestion. Please see **pg. 2, lines 26-27**, where we have made this change.

**New Comment 2:** P16, L8. The most likely explanation for the negative correlation between CH4 production and porewater acetate is that the added porewater lowered pH. This is a very common artifact in incubation studies that can only be dismissed with data on the pH of the incubation water. It can happen in two ways: (1) is if the added porewater is lower in pH than the incubation water or (2) if the porewater stimulates microbial respiration, causing CO2 to accumulate in the jar and acidify the incubation water. The correlation between CH4 and acetate could indicate that porewater with more acetate also had lower pH; this is possible because acetate is a weak acid. Alternatively, porewater with more acetate could have acidified the sample faster due to more CO2 production. This comment has the potential to affect other parts of the discussion as well. **Response:** Although we acknowledge that pH can affect $CH_4$ production, we respectfully disagree with the reviewer on this point. We did not add any additional porewater to the incubation; porewater was extracted from a subsample of the sediment prior to incubation. Therefore, porewater measurements represented what was naturally in the sediment. Nevertheless, we did add water from the overlying water column in the incubation (please see **Table 1** for chemistry information), and while it is possible that this altered the pH of the incubation, we were not able to detect any acetate in the water column. Therefore, we find no reason to suspect that the incubation pH was correlated with the acetate concentration of the porewater, which is also the total concentration in the incubation before adding organic matter. If we had also found a negative correlation between $CH_4$ production and porewater acetate in the brackish/freshwater wetland comparison, then the pH explanation might be an alternative explanation. However, we found that porewater acetate had a positive relationship with $CH_4$ production in incubations where no organic matter was added. It was not until we added organic matter that we actually observed this negative relationship between the change in $CH_4$ production and acetate, which is why we think the amount of electron donors for methanogens became saturated. Additionally, we believe that sulfate is more likely to affect the pH of the incubation solution than acetate because the total sulfate concentrations in the incubation were much higher than the acetate concentrations and because a hydrogen sulfate ion is a stronger acid than protonated acetate.

**New Comment 3:** P2, L20. Delete "of choice" **Response:** Done, please see **pg. 2, line 14**.

**New Comment 4:** P2, L24. It is incorrect to say that "carbon" (or carbon dioxide, which is what I think the authors meant) is not an energetically favorable electron acceptor because this can only be determined by calculating the energy yield of full redox couples. You can say that "CO2 reduction is not an energetically favorable electron accepting pathway compared to other pathways." **Response:** We agree with you that generally one should not consider the electron acceptor without the donor. Please see **pg. 2, line 17**, where we have attempted to correct this. Also, carbon in the form of a methyl group ($CH_3$)

or carbon dioxide acts as an electron acceptor in acetoclastic or hydrogenotrophic methanogenesis, respectively. Please see **pg. 2, line 18**, where we have attempted to clarify this.

**New Comment 5:** P2, L25. Replace "signal" with "indicate" **Response:** Done, please see **pg. 2, line 19**.

**New Comment 5:** P2, L30. Edit to "Both electron donor and electron acceptor availability will therefore play an important role in determining the effects of global change on CH4 production." **Response:** Done, please see **pg. 2, lines 24-25**.

**New Comments 6 & 7:** P3, L18. Edit to "periodically inundated tidal brackish wetlands". P4, L4. Tidal wetlands are not typically inundated at the lowest tides, almost by definition. Does this refer to mean tide? Mean low tide? **Response:** These tidal wetlands are at a confluence of a river mouth and the Gulf of Alaska, so do they do tend to be fully inundated or water-logged even at mean low tide. We have attempted to make this clearer, please see **pg. 3, line 29, and pg. 4, lines 1-3**.

**New Comment 8:** P8, L19. Edit to "…were affected by being incubated anaerobically with tidal…" **Response:** Done, please see **pg. 8, line 20.**

**New Comment 9:** P9, L17. A number of reviewer comments boil down to distinguishing between cause and effect in the language of the paper. This is one example where the word "influenced" should be replaced by "correlated with" or "associated with". You cannot say that acetate influenced CH4 production. **Response:** Done, please see **pg. 9, lines 18-19**.

**New Comment 10:** P10, L5. Edit to "Incubating non-tidal freshwater wetland soils with brackish water…" **Response:** Done, please see **pg. 10, line 9**.

**New Comment 11:** P12, L22. Change "coupled" to "related" because coupled implies you separated cause and effect. **Response:** Done, please see **pg. 12, line 30**.

**New Comment 12:** P14, L10. Please use a more precise word or phrase in place of "primed" **Response:** Done, please see **pg. 14, lines 19-20**.

Respectfully,

Carmella Vizza, Will West, Stuart Jones, Julia Hart, and Gary Lamberti

[revised manuscript text omitted]

---

## Author Response (AR3)

Dear Dr. Abril,

We very much appreciate the time and effort you put into reviewing this manuscript and its many iterations. As the editor, we greatly appreciate your service and your decision to now publish the paper as it is.

Respectfully,

Carmella Vizza, Will West, Stuart Jones, Julia Hart, and Gary Lamberti